# Hyperbolic Neural Operator

**Jieyuan Pei** [1] **Zhuoxuan Li** [2] **Wei Li** [1] **Haobo Zhang** [1] **Jiawei Jiang** [1] **Jianwei Zheng** [1 3]

## Abstract

Neural operators learn solution operators for parametric PDE families, mapping coefficients, forcing fields, or geometric inputs to full solution fields and thereby accelerating scientific computation. Transformer-based architectures offer strong flexibility on irregular domains, but dense dot-product attention often allocates pairwise scoring uniformly across token pairs, neglecting that far-field interactions in many discretized PDE kernels are numerically compressible. To address this mismatch, we draw inspiration from classical fast solvers that exploit hierarchical near–far organization. We further observe that embedding such tree-structured hierarchies in Euclidean space incurs inherent distortion, whereas hyperbolic space naturally accommodates exponential branching. Consequently, we propose **Hyperbolic Neural Operator (HNO)**, which leverages intrinsic hyperbolic geometry to instantiate a continuous Gibbs kernel based on stabilized geodesic distances on the Lorentz hyperboloid. This design imposes a geometric inductive bias for learnable multi-scale near–far routing within a unified attention mechanism. Empirically, HNO achieves the lowest error among the evaluated methods on six PDE benchmarks and two large-scale unstructured CFD tasks, reducing the mean relative $\ell_2$ error by up to 40% in the best evaluated setting. Code is available in the GitHub repository.

## 1. Introduction

Solving partial differential equations (PDEs) is a cornerstone of scientific and engineering computation, with broad applications spanning electromagnetic interactions with materials and engineered structures, aerodynamic design of vehicles and aircraft, as well as earth system and climate modeling (Tang et al., 2025). In practice, PDEs are typically discretized into meshes and then resolved by numerical approaches (Bonito et al., 2024); these simulations often take several hours or even days for complex structures. Recently, neural operators (NO) have emerged as popular surrogates for learning solution operators of parametric PDE families (Wu et al., 2024; Rahman et al., 2024). Given data generated by numerical solvers, NOs map PDE inputs such as coefficients, forcing fields, and geometric descriptors to full solution fields, enabling orders-of-magnitude faster inference (Kovachki et al., 2023), even on out-of-distribution problems at scale (Luo et al., 2025; Hu et al., 2025).

Generally, neural operators follow the workflow of "*lift-process-project*". The input function is first lifted into a latent representation, followed by iterative processing to model interactions, and finally projected back to the output function space (Kovachki et al., 2023). Currently, the scholarly pursuit to innovate NOs bifurcates into two branches: spectral-based and transformer-based paradigms. Spectral methods map inputs to representations in Fourier (Liu & Yu, 2025), Wavelet (Hu et al., 2025), or Radon spaces (Lu et al., 2025). These coefficient-space updates yield efficient global coupling on regular grids; however, extending the same mechanism to irregular geometries typically requires additional deformation or quadrature machinery, leading to a structural bottleneck. In contrast, transformer-based operators offer strong flexibility for irregular meshes and complex domains. In practice, various tokenization schemes are elaborated, lifting functions into physics-guided slices (Luo et al., 2025), spatio-temporal tokens (Holzschuh et al., 2025), or geometry-aware representations in arbitrary domains (Wen et al., 2025). However, a fundamental efficiency gap arises from a mismatch with physical reality, where nearby interactions require fine-grained resolution, whereas well-separated contributions can often be summarized with few coefficients (Greengard & Rokhlin, 1987). Here and throughout, *compressibility* refers to numerical or low-rank compressibility of far-field interactions, not fluid incompressibility. Crucially, dense dot-product attention overlooks this distinction, allocating pairwise scoring uniformly across token pairs regardless of spatial separation,

[1]College of Computer Science and Technology, Zhejiang University of Technology, Hangzhou, China [2]Tongji University, Shanghai, China [3]Zhejiang Key Laboratory of Visual Information Intelligent Processing, Zhejiang University of Technology, Hangzhou, China. Correspondence to: Jianwei Zheng <zjw@zjut.edu.cn>.

*Proceedings of the 43rd International Conference on Machine Learning*, Seoul, South Korea. PMLR 306, 2026. Copyright 2026 by the author(s).

thereby rendering the computation prohibitive as the number of tokens grows.

Classical fast solvers (Greengard & Rokhlin, 1987) exploit this knowledge, managing near-field and far-field interactions through recursive spatial partitioning, typically structured as a multilevel tree. Similarly, hierarchical matrix techniques (Hackbusch, 2015) formalize this structure by computing interactions within neighboring leaf clusters directly, while approximating well-separated cluster pairs—those satisfying an admissibility condition—via low-rank summaries, as illustrated in Fig. 1B. However, embedding tree-structured metrics in Euclidean space suffers from inherent distortion due to a fundamental geometric mismatch: Euclidean volume grows polynomially, whereas tree structures expand exponentially (Nickel & Kiela, 2017). In this work, we turn to hyperbolic space, which exhibits exponential volume growth relative to its radius and admits low-distortion embeddings of tree-like metrics in low dimension (Sarkar, 2011). By contrast, Euclidean embeddings of expanding tree metrics require substantially larger dimension or incur higher distortion (Sala et al., 2018). Rather than direct embedding of physical coordinates, we leverage intrinsic hyperbolic geometry to modulate the interaction kernel. On that basis, a representational bias toward near-field and far-field distinctions naturally emerges from learned geodesic distances. By further marrying the geometric flexibility of transformers with the hierarchical efficiency of classical solvers, we propose **Hyperbolic Neural Operator (HNO)**. Instead of treating attention as discrete token mixing, HNO instantiates a continuous Gibbs kernel defined by negative stabilized geodesic distances on the Lorentz hyperboloid. HNO is inspired by the near–far organization of fast solvers, but it does not implement multipole expansions, explicit hierarchical trees, or error-controlled truncation; the contribution is a learnable analogue of near–far routing inside the neural operator. Our contributions are as follows.

- We propose the Hyperbolic Neural Operator (HNO), whose core component is a stabilized Gibbs kernel based on Lorentz geodesic distance. This design lets scale-separation biases be learned directly within a continuum-first framework.

- We provide a theoretical interpretation linking hyperbolic geometry to the near–far decomposition in numerical solvers. The analysis highlights how radius–angle coupling enables efficient routing and establishes stability and discretization-consistency results for the conditioned hyperbolic attention operator under stated assumptions.

- HNO achieves the lowest error among the compared baselines on six PDE benchmarks and two challenging unstructured-mesh CFD tasks. Empirical analysis

confirms that HNO learns to balance local precision with global aggregation, while reducing the interaction cost from $O(N^2)$ to $O(NM + M^2)$ via an $M$-token interaction core.

## 2. Related Work

### 2.1. Deep Learning Solvers for PDEs

Deep learning paradigm for PDEs mainly covers instance-specific solvers and operator learning across parametric families. Physics-informed neural networks enforce residuals and boundary conditions in the continuous domain and are commonly trained per instance (Raissi et al., 2019; Karniadakis et al., 2021). Operator learning amortizes across instances by learning maps between function spaces, including DeepONet with branch and trunk networks (Lu et al., 2019), kernel integral neural operators (Kovachki et al., 2023), and transformation-based parameterizations such as FNO (Li et al., 2021) and RNO (Lu et al., 2025).

Transformers improve long-range coupling and scalability for operator learning. Representatively, Transolver introduces Physics Attention that partitions the domain into learnable slices and attends to physics-aware tokens on general geometries (Wu et al., 2024). Transolver++ further scales to million-point geometries via highly parallel execution and a local adaptation for industrial settings (Luo et al., 2025). Recent pretrained or universal neural PDE solvers, including OmniArch, DPOT, PDE-Transformer, and Unisolver, study cross-family pretraining, autoregressive operator pretraining, or PDE-conditioned zero/few-shot generalization (Chen et al., 2025; Hao et al., 2024; Holzschuh et al., 2025; Zhou et al., 2025). These works are complementary to our setting: HNO focuses on per-family supervised operator design and the geometry of learned routing rather than broad cross-family pretraining. For arbitrary domains, GAOT combines geometry embeddings with transformer processing and multiscale encoder-decoder designs to balance accuracy and efficiency (Wen et al., 2025). However, dense dot-product attention in these architectures often allocates pairwise scoring uniformly across token pairs, failing to exploit the compressibility of far-field physical interactions.

### 2.2. Numerical Solvers for PDEs

Traditional numerical solvers discretize PDEs through local schemes, including finite differences on structured grids, finite volume methods (LeVeque, 2002), and finite element alternatives (Brenner & Scott, 2008), with well-developed stability and convergence theory. In nonlocal formulations and integral-equation settings, discretization produces dense kernel matrices. For asymptotically smooth kernels, far-field blocks are often numerically low-rank. Fast multipole methods exploit this property to accelerate kernel summation and

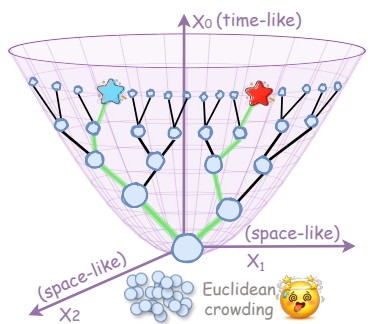

**A. Physical Response**
(Near-far Decomposition)

**B. Hierarchical Tree**
(Logical Structure)

**C. Hyperbolic Geometry**
(Lorentz model, schematic)

*Figure 1.* **A. Near–far decomposition.** Near-field interactions require fine resolution, while far-field regions are summarized coarsely. **B. Hierarchical tree.** Recursive summarization induces a multilevel cluster tree, where interactions route via lowest common ancestors (LCA). **C. Hyperbolic geometry (Lorentz model, schematic).** Exponential capacity alleviates Euclidean crowding; geodesic distances provide a proxy for hierarchy-aware routing.

matrix vector products (Greengard & Rokhlin, 1987), while hierarchical matrix frameworks provide a unified algebraic view and enable fast linear algebra such as approximate factorizations and direct solutions (Hackbusch, 2015).

Kernel-independent fast multipole methods replace analytic expansions with interpolation-based compression and require only kernel evaluations (Fong & Darve, 2009). Recent work simplifies far-field compression through recursive skeletonization with streamlined hierarchical data structures (Yesypenko et al., 2025), and develops translation-invariant $\mathcal{H}^2$ constructions on unstructured quasi-uniform meshes to reduce far-field storage (Börm & Henningsen, 2025). These methods are instance specific, so changes in coefficients, geometry, or boundary conditions require rerunning the solver and often rebuilding the compression.

## 3. Why Hyperbolic Geometry?

Searching for PDE solutions can be viewed as learning a *propagation operator* that transports and aggregates information across the domain. Classical fast solvers exploit a *near–far* structure, in which nearby interactions require fine granularity, while contributions from well-separated regions can be summarized with a small number of coefficients. Fig. 1(A) visually provides this decomposition.

This compressibility stems from the fact that many PDE kernels are asymptotically smooth (analytic away from singularities). In the far-field, contributions from a compact source cluster $C$ (radius $r$) to a distant target $x$ ($\rho \gg r$ from the center of $C$) admit truncated multipole-like expansions with geometrically decaying truncation error; Appendix D.3 states the regularity assumptions and proof, while Greengard and Rokhlin (Greengard & Rokhlin, 1987) provide the classical FMM origin.

**Lemma 3.1** (Far-field compressibility). *An order-$p$ far-field expansion yields a truncation error (remainder) $R_{p+1}(x;C)$*

bounded by $|R_{p+1}(x;C)| \le c \, (r/\rho)^{p+1}$, where $c > 0$ depends on the kernel and locations. This enables far-field effects to be mediated by a small, fixed number of coefficients with controllable accuracy.

To exploit far-field compression across the whole domain, several numerical methods apply the property recursively across different scales. Repeated near-field resolution and far-field summarization induce a multilevel cluster tree over discretization elements as in Fig. 1(B). In such a hierarchy, two locations interact strongly if they share a recent common ancestor and weakly if their lowest common ancestor is high in the tree. This hierarchy is induced by the chosen clustering and discretization practices, rather than by the physical distance of the underlying PDE domain itself.

From a geometric viewpoint, recursive near–far decomposition induces a tree-structured proximity whose branching grows exponentially with depth. Euclidean space has only polynomial volume growth given any dimensions and suffers crowding when representing such expanding structure. The Hyperbolic space instead expands exponentially with radius, fitting the hierarchical growth more naturally.

**Lemma 3.2** (Euclidean crowding and hyperbolic capacity). *Let $B_{\mathbb{R}^m}(r)$ and $B_{\mathbb{H}^m}(r)$ denote radius-$r$ balls in $\mathbb{R}^m$ and in $\mathbb{H}^m$ with curvature $-1$. Their volumes satisfy*

$$\mathrm{Vol}\big(B_{\mathbb{R}^m}(r)\big) \propto r^m, \quad \mathrm{Vol}\big(B_{\mathbb{H}^m}(r)\big) \propto \exp\big((m-1)r\big),$$

*which is consistent with representing exponentially branching structure with lower distortion in fixed dimension (Nickel & Kiela, 2017; Ganea et al., 2018b).*

As sketched in Fig. 1(C), exponential volume growth alleviates Euclidean crowding and provides a geometry-aligned proxy for hierarchy-aware routing. Taken together, far-field compressibility and multilevel recursion motivate a scale-tree view of operator interactions. Such scale trees summarize far-field information through a bounded number of

coefficients at each level while retaining near-field fidelity. With bounded far-field rank, dense interactions admit subquadratic compression. This compression principle has long been validated in numerical PDE solvers. However, directly replacing dense interactions with generic linear attention can degrade accuracy (Wu et al., 2024), since uniform linearization does not respect near–far structure. Hyperbolic geometry offers a continuous surrogate for hierarchical growth and provides a geometry-aligned routing prior for near–far interactions. The more formal statements and proofs are deferred to Appendices D–G, covering cluster-tree motivation, near–far routing, Schur stability, bi-Lipschitz residual conditions, and discretization consistency.

## 4. Methodology

### 4.1. Problem Setting

Let $\mathcal{X} = L^2(D; \mathbb{R}^{d_a})$ and $\mathcal{Y} = L^2(D; \mathbb{R}^{d_u})$ be Hilbert spaces over a bounded domain $D \subset \mathbb{R}^d$. We focus on learning nonlinear operators $\mathcal{G}^\dagger : \mathcal{X} \to \mathcal{Y}$ that arise as solution maps of parametric PDEs. In our PDE benchmarks, $a \in \mathcal{X}$ typically represents an input coefficient or forcing field, and $u \in \mathcal{Y}$ is the corresponding solution field.

Given $n_{\text{train}}$ observations $\{(a_j, u_j)\}_{j=1}^{n_{\text{train}}}$ with $a_j \sim \nu$ drawn i.i.d. and $u_j \approx \mathcal{G}^\dagger(a_j)$, possibly with noise, we consider a parametric family of operators

$$\mathcal{G}_\theta : \mathcal{X} \to \mathcal{Y}, \quad \theta \in \Theta, \quad (1)$$

where $\Theta$ is a finite-dimensional parameter space and $\theta$ is learned by minimizing the expected loss

$$\min_{\theta \in \Theta} \mathbb{E}_{a \sim \nu} \big[ \ell \big( \mathcal{G}_\theta(a), \mathcal{G}^\dagger(a) \big) \big], \quad (2)$$

where $\ell$ is a loss functional mapping $\mathcal{Y} \times \mathcal{Y}$ to $\mathbb{R}$. This formulation parallels finite-dimensional supervised learning (Kovachki et al., 2023), with the key property that $\Theta$ is shared across all discretization resolutions.

Given the problem setting, a recurring challenge is *hierarchical multiscale coupling*. Recall that nearby regions exchange fine-grained information, while distant regions interact through coarser summaries. This motivates learnable near-field and far-field routing mechanisms.

### 4.2. Hyperbolic Neural Operator

To instantiate the resolution-agnostic operator $\mathcal{G}_\theta$ in Eq. (1), we adopt the kernel-integral neural operator framework (Kovachki et al., 2023). Our key departure from prior work lies in the design of the kernel geometry. We replace the conventional Euclidean dot-product similarity with a Gibbs kernel based on stabilized hyperbolic-distance logits, which naturally encodes the hierarchical near–far structure discussed in Section 3 in a geometry-aware manner.

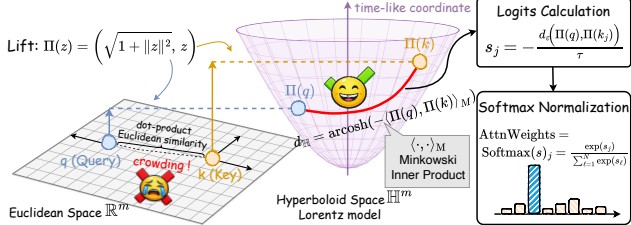

*Figure 2.* Hyperbolic-distance attention in the Lorentz model. We lift query and key vectors to the hyperboloid, compute stabilized hyperbolic distance, and convert it to attention weights with a Gibbs kernel and softmax normalization.

**Neural Operator Architecture.** We follow the standard construction comprising lifting, iterative kernel integration, and projection.

$$\mathcal{G}_\theta := \mathcal{Q} \circ v_T \circ \cdots \circ v_1 \circ \mathcal{P}, \quad (3)$$

where $\mathcal{P}$ lifts input features to a higher-dimensional latent space and $\mathcal{Q}$ projects back to the output space. Each iterative update takes the form

$$v_{t+1}(x) = \sigma_t \big( L_t v_t(x) + (\mathcal{K}_t v_t)(x) + b_t(x) \big), \quad (4)$$

where $L_t$ are pointwise linear operators, $\mathcal{K}_t$ are kernel integral operators described below, $b_t$ are learnable bias functions, and $\sigma_t$ are activation functions.

**Kernel Integral Operator.** In Eq. (4), the kernel integral operator $\mathcal{K}_t$ is the only term that couples distinct locations on the continuous domain $D$, and thus models nonlocal interactions. Following standard formulations (Kovachki et al., 2023), the kernel operator acting on a vector-valued function $v : D \to \mathbb{R}^c$ is defined as

$$(\mathcal{K}_\kappa v)(x) = \int_D \kappa(x, y; a) \, v(y) \, \mathrm{d}\mu(y), \quad (5)$$

where $\mu$ is the base measure on $D$, and $\kappa(\cdot, \cdot; a) : D \times D \to \mathbb{R}$ is a scalar kernel conditioned on the input field $a$. We share this scalar kernel across channels and use learnable value and output projections $W_V$ and $W_O$ for cross-channel interaction. Our theoretical analysis accounts for these projections through the factor $\|W_O\|_{\text{op}} \cdot \|W_V\|_{\text{op}}$.

**Hyperbolic Integral Operator.** Standard transformer-based neural operators parameterize $\kappa$ by Euclidean dot-product similarity, which assumes flat geometry and offers little primitive for hierarchical near–far routing. Motivated by Section 3, we instead instantiate $\kappa$ in Eq. (5) using a stabilized hyperbolic distance in the Lorentz model (Nickel & Kiela, 2018). Given the input field $a$ and the current hidden state $v_t$, we define query and key maps by separate pointwise projections, written as $q(x; a) := Q_\theta(v_t(x), a, x)$

and $k(y; a) := K_\theta(v_t(y), a, y)$ in $\mathbb{R}^m$. In the latent cross-attention implementation, queries are projected from latent tokens and keys from input-point features; in latent self-attention, queries and keys use separate latent projections. Unlike Euclidean dot-product attention, these projections induce a *data-adaptive interaction metric*, mapping $(v_t, a)$ into a space where geodesic proximity correlates with coupling strength, so the kernel provides a soft notion of near–far interaction rather than relying on a fixed spatial partition. The lifting to the Lorentz hyperboloid is

$$\Pi(z) := \left( \sqrt{1 + \|z\|^2}, \, z \right) \in \mathbb{H}^m. \tag{6}$$

The Lorentz model equips $\mathbb{H}^m$ with the Minkowski inner product $\langle u, v \rangle_M := -u_0 v_0 + \sum_{i=1}^m u_i v_i$, which further induces the geodesic distance $d_\mathbb{H}(u, v) = \operatorname{arcosh}(-\langle u, v \rangle_M)$. For numerical stability, we score interactions by a stabilized distance as follows.

$$d_\epsilon(u, v) := \operatorname{arcosh}\left( \max\{1 + \epsilon, \, -\langle u, v \rangle_M\}\right), \quad \epsilon > 0. \tag{7}$$

This keeps the argument bounded away from $1$; in practice, the clamping is rarely active (Appendix E.1), so geometric properties are mainly governed by the true $d_\mathbb{H}$. Since a shorter distance should yield a larger attention weight, we form stabilized hyperbolic-distance logits as negative distance with temperature $\tau > 0$.

$$s(x, y; a) := -\frac{1}{\tau} d_\epsilon\left(\Pi(q(x; a)), \Pi(k(y; a))\right). \tag{8}$$

The resulting normalized kernel is written as follows.

$$\kappa(x, y; a) := \frac{\exp(s(x, y; a))}{Z(x; a)},$$
$$Z(x; a) := \int_D \exp\left(s(x, y'; a)\right) d\mu(y'). \tag{9}$$

Including the value and output projections, the full operator becomes $\mathcal{K} = W_O \circ \mathcal{K}_\kappa \circ W_V$ and acts on values as

$$(\mathcal{K}v)(x) := W_O(\mathcal{K}_\kappa(W_V v))(x)$$
$$= W_O \int_D \kappa(x, y; a) \, W_V v(y) \, d\mu(y). \tag{10}$$

Conditioned on the query and key maps, the kernel $\kappa(\cdot, \cdot; a)$ is fixed and the map $v \mapsto \mathcal{K}v$ is linear in $v$. See Section 4.3 for the near–far routing mechanism induced by negative curvature.

**Fast-solver view.** Equations (9)–(10) formally instantiate a *conditioned integral kernel* and hence a discretized kernel matrix after Nyström quadrature, which is precisely the object compressed by fast multipole and hierarchical-matrix methods via near-/far-field structure. In HNO, the Lorentz geodesic distance induces a *data-adaptive near–far metric*

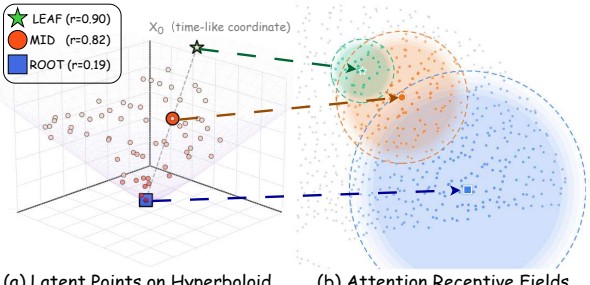

(a) Latent Points on Hyperboloid     (b) Attention Receptive Fields

*Figure 3.* Learned multi-scale attention on Elasticity. **a.** Latent tokens on the Lorentz hyperboloid. **b.** Corresponding receptive fields. Tokens closer to the origin (smaller radius $r$) exhibit broader attention, while large-radius tokens attend more locally.

in feature space rather than physical coordinates, so interaction strength is organized by learned geometric proximity, yielding routing behavior analogous to near-field coupling versus far-field summaries. This analogy is architectural and statistical: HNO does not build an explicit FMM tree, compute multipole expansions, or provide an error-controlled truncation rule.

We approximate the integral operator with Nyström quadrature on nodes $\{(x_j, w_j)\}_{j=1}^N$. Let $v_j := v(x_j)$ and set $s_{ij} := s(x_i, x_j; a)$. The discrete weights are computed by a weighted softmax

$$\alpha_{ij} = \frac{w_j \exp(s_{ij})}{\sum_{\ell=1}^N w_\ell \exp(s_{i\ell})}. \tag{11}$$

The discrete operator then reads

$$(\mathcal{K}_N v)_i := W_O \sum_{j=1}^N \alpha_{ij} \, W_V v_j. \tag{12}$$

For scalability at large $N$, we avoid forming the dense $N \times N$ node-to-node score matrix by introducing a compression map that aggregates node features into $M \ll N$ representative tokens. On grids and meshes, these are obtained via patch tokens; on point clouds, we learn a set of latent inducing tokens. The hyperbolic metric shapes the non-uniform routing within this compressed interaction pathway, paralleling hierarchical solvers that capture well-separated interactions through low-rank surrogates rather than exhaustive pairwise coupling.

In the analysis, we condition on the network weights and treat $\epsilon > 0$ and $\tau \geq \tau_{\min}$ as fixed hyperparameters; constants depend on the resulting feature-map Lipschitz bounds $L_q, L_k$. Practical discretization details, including proofs of operator bounds, are deferred to Appendix F.2.

### 4.3. Scale Separation and Near–Far Routing

Our attention kernel is a Gibbs kernel on hyperbolic distance (Eq. (9)). Since the unnormalized weight is $\exp(-d_\epsilon/\tau)$,

smaller hyperbolic distances receive larger attention mass. For $0 < \delta < 1$, let $R_\delta := \tau \log(1/\delta)$ and consider the effective interaction set $\{y \mid d_\epsilon(\Pi(q(x;a)), \Pi(k(y;a))) \leq R_\delta\}$, beyond which the unnormalized mass is suppressed by at least a factor $\delta$. Negative curvature couples radius and angle, yielding scale separation. That is, embeddings at large radii interact with non-negligible weight only within exponentially small angular apertures, whereas those near the origin remain broadly connected. This mechanism is visualized on Elasticity in Fig. 3.

**Radius–angle coupling.** We modulate attention breadth via hyperbolic radius, with larger radii favoring locality and smaller radii enabling broad aggregation. Let $o = (1, 0, \ldots, 0)$ be the Lorentz origin and define the radius as

$$
\begin{aligned}
r(z) := d_{\mathbb{H}}\big(o, \Pi(z)\big) &= \operatorname{asinh}(\|z\|) \\
&= \log\big(\|z\| + \sqrt{1 + \|z\|^2}\big),
\end{aligned} \tag{13}
$$

which grows slowly with the Euclidean norm. Hence radii are learned implicitly through feature norms, without hardcoding per-token scales.

The key mechanism is radius–angle coupling in hyperbolic distance. For positive radii, write $u = (u_0, u_{1:m})$ and $v = (v_0, v_{1:m})$ with directions $\hat{u} := u_{1:m}/\|u_{1:m}\|$ and $\hat{v} := v_{1:m}/\|v_{1:m}\|$, and define $\cos\theta(u, v) := \hat{u}^\top \hat{v}$. For $u, v \in \mathbb{H}^m$ with radii $r_u, r_v$, the hyperbolic law of cosines is

$$
\begin{aligned}
\cosh d_{\mathbb{H}}(u, v) = {}& \cosh r_u \cosh r_v \\
& - \sinh r_u \sinh r_v \cos\theta(u, v),
\end{aligned} \tag{14}
$$

so bounded distance increasingly constrains angular separation as radii grow, since the coefficient $\sinh r_u \sinh r_v$ scales exponentially in $r_u + r_v$. This is quantified as follows.

**Proposition 4.1** (Exponential angular contraction). *For any $R > 0$, there exists $C_R > 0$ such that if $r_u, r_v > 0$ and $d_{\mathbb{H}}(u, v) \leq R$, then*

$$
\theta(u, v) \leq \min\left\{\pi, C_R \exp\left(-\frac{r_u + r_v}{2}\right)\right\}. \tag{15}
$$

Under our Gibbs kernel, the unnormalized weight decays as $\exp(-d_\epsilon/\tau)$. Thus, through radius–angle coupling, large-radius regions receive non-negligible attention only within exponentially narrow cones, enabling learned separation of near-field (local) and far-field (compressed) interactions. Proof is deferred to Appendix E.2.

## 5. Experiments

We evaluate the proposed HNO on six standard PDE benchmarks and two large-scale unstructured-mesh benchmarks, covering a wide range of geometries and problem settings.

**Benchmarks and baselines.** As shown in Tab. 2, HNO is compared with 19 baselines, covering CNN surrogates, frequency-domain neural operators, and transformer-based operator learners. The suite spans point clouds, (un)structured meshes, and regular grids under 2D and spatiotemporal settings. These FNO (Li et al., 2021) and geo-FNO (Li et al., 2023a) tasks are widely used in operator learning. We further include two large-scale unstructured-mesh benchmarks with about 32k nodes per sample.

*Table 1.* Summary of used benchmarks that cover a wide range of geometries. #Mesh records the size of discretized meshes.

| Geometry | Benchmarks | #Dim | #Mesh |
|---|---|---|---|
| Point Cloud | Elasticity | 2D | 972 |
| Structured Mesh | Plasticity | 2D+Time | 3,131 |
| | Airfoil | 2D | 11,271 |
| | Pipe | 2D | 16,641 |
| Regular Grid | Navier-Stokes | 2D+Time | 4,096 |
| | Darcy | 2D | 7,225 |
| Unstruct. Mesh | ShapeNet Car | 3D | 32,186 |
| | AirfRANS | 2D | 32,000 |

**Implementation Setting.** We follow the standard benchmark protocol (Wu et al., 2024) and report the mean relative error $\ell_2$. For baselines, we use official implementations and reported hyperparameters when available. HNO PDEBench results are averaged over three runs unless otherwise noted; baseline values follow official reports or authors' implementations. All HNO experiments are run on a single NVIDIA A6000 48GB GPU. The complete training details and model configurations are provided in the Appendix I.

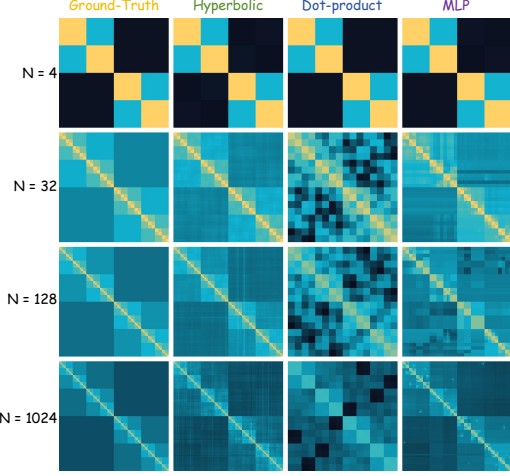

*Figure 5.* Toy multiscale tree-kernel fitting (log scale). Rows vary tree size $N$; columns show the target diffusion kernel and fitted kernels from hyperbolic, dot-product, and MLP attention.

### 5.1. Toy Multiscale Tree-Kernel Fitting

We construct a toy graph diffusion on a complete binary tree of depth $L$ with $N = 2^L$ leaves:

$$
u = Ks, \quad K_{ij} \propto \exp(-\gamma_{\text{eff}} d_{ij}), \quad \sum_{j=1}^{N} K_{ij} = 1. \tag{16}
$$

*Table 2.* Main results on six standard benchmarks are reported in mean relative $\ell_2$ error, for which lower is better. Relative improvement denotes the relative error reduction of our model over the second-best method, consistently computed as $(e_{2nd} - e_{ours})/e_{2nd}$.

| | OPERATOR | POINT CLOUD | REGULAR GRID | | STRUCTURED MESH | | |
| | | ELASTICITY | NAVIER–STOKES | DARCY | PLASTICITY | AIRFOIL | PIPE |
|---|---|---|---|---|---|---|---|
| **CLASSIC** | UNET (2015) | 0.0235 | 0.1982 | 0.0080 | 0.0051 | 0.0079 | 0.0065 |
| | RESNET (2016) | 0.0262 | 0.2753 | 0.0587 | 0.0233 | 0.0391 | 0.0120 |
| | SWIN (2021) | 0.0283 | 0.2248 | 0.0397 | 0.0170 | 0.0270 | 0.0109 |
| | DEEPONET (2019) | 0.0965 | 0.2972 | 0.0588 | 0.0135 | 0.0385 | 0.0097 |
| **FREQUENCY** | WMT (2021) | 0.0359 | 0.1541 | 0.0082 | 0.0076 | 0.0075 | 0.0077 |
| | U-FNO (2022) | 0.0239 | 0.2231 | 0.0183 | 0.0039 | 0.0269 | 0.0056 |
| | FNO (2021) | 0.0229 | 0.1556 | 0.0108 | 0.0074 | 0.0138 | 0.0067 |
| | U-NO (2022) | 0.0258 | 0.1713 | 0.0113 | 0.0034 | 0.0078 | 0.0100 |
| | F-FNO (2021) | 0.0263 | 0.2322 | 0.0077 | 0.0047 | 0.0078 | 0.0070 |
| | LSM (2023) | 0.0218 | 0.1535 | 0.0065 | 0.0025 | 0.0059 | 0.0050 |
| | RNO (2025) | 0.0078 | 0.0894 | 0.0054 | 0.0012 | 0.0054 | 0.0044 |
| **TRANSFORMER** | GALERKIN (2021) | 0.0240 | 0.1401 | 0.0084 | 0.0120 | 0.0118 | 0.0098 |
| | HT-NET (2022) | / | 0.1847 | 0.0079 | 0.0333 | 0.0065 | 0.0059 |
| | OFORMER (2022) | 0.0183 | 0.1705 | 0.0124 | 0.0017 | 0.0183 | 0.0168 |
| | GNOT (2023) | 0.0086 | 0.1380 | 0.0105 | 0.0336 | 0.0076 | 0.0047 |
| | FACTFORMER (2023B) | / | 0.1214 | 0.0109 | 0.0312 | 0.0071 | 0.0060 |
| | ONO (2024) | 0.0118 | 0.1195 | 0.0076 | 0.0048 | 0.0061 | 0.0052 |
| | TRANSOLVER (2024) | 0.0065 | 0.0892 | 0.0057 | 0.0013 | 0.0053 | 0.0046 |
| | TRANSOLVER++ (2025) | 0.0064 | 0.1010 | 0.0056 | 0.0014 | 0.0054 | 0.0042 |
| | **HNO (OURS)** | **0.0037** | **0.0676** | **0.0045** | **0.0009** | **0.0048** | **0.0027** |
| | RELATIVE IMPROVEMENT | **+42.2%** | **+24.2%** | **+16.7%** | **+25.0%** | **+9.4%** | **+35.7%** |

*Figure 4.* Qualitative error maps for HNO vs Transolver++.

Here $s$ is a signal on the leaves, $u$ is its diffused output, and $d_{ij} = d_{tree}(i, j)$ is the leaf shortest-path distance determined by LCA depth. Thus $K$ is a row-stochastic Markov diffusion kernel whose discrete tree-distance levels induce multiple interaction scales. Hyperbolic distance attention preserves this nested block structure better than dot-product and MLP controls (Fig. 5; Appendix H): the learned kernel keeps both fine diagonal bands and coarse ancestor-level partitions, whereas the Euclidean controls either over-smooth across blocks or lose the hierarchy as the tree grows. Thus the toy study bridges the motivation and the PDE experiments: it checks whether one metric can keep sharp local bands and coherent ancestor blocks before dataset-specific training choices enter. The scaling study in Appendix H confirms the same pattern from small to large trees: hyperbolic attention consistently gives lower kernel, operator, and row-wise KL errors than the dot-product control, while preserving visually coherent ancestor-level blocks. This toy problem is therefore not used as a PDE accuracy benchmark; rather, it isolates the geometric routing bias before moving to the full operator-learning tasks below.

## 5.2. Standard Benchmarks

Tab. 2 shows that HNO consistently achieves the lowest mean relative $\ell_2$ errors on six standard benchmarks. For Elasticity, nonuniform point sampling and elastic interactions make multi-scale coupling difficult for Euclidean attention; for Pipe, boundary-driven flow on structured meshes creates long-range dependencies along the channel. HNO therefore obtains its largest gains on settings where local resolution and global conditioning must coexist. On regular grids, HNO remains competitive, and Tab. 3 shows that it is substantially more memory-efficient than dot–product attention transformer baselines. Fig. 4 highlights representative error maps: HNO visibly reduces localized artifacts compared with Transolver++, matching the numerical advantages in the *Relative Improvement* row of Tab. 2. The gains are not restricted to one discretization: HNO improves the best competing error across point clouds, regular grids, structured meshes, and time-dependent settings. The suite stresses varied failure modes–irregular samples, grid fields, boundary-driven meshes, and transient dynamics–rather than repeating one geometry.

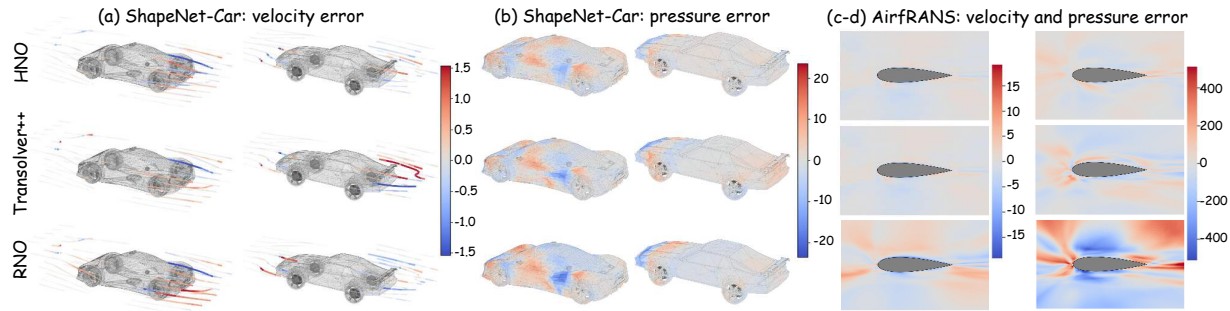

*Figure 6.* Visualization of prediction errors for velocity field (left) and surface pressure (right) on ShapeNet-Car (a-b) and AirfRANS (c-d) datasets. Rows compare HNO, Transolver++, and RNO. Blue indicates under-prediction, red indicates over-prediction, and white represents zero error.

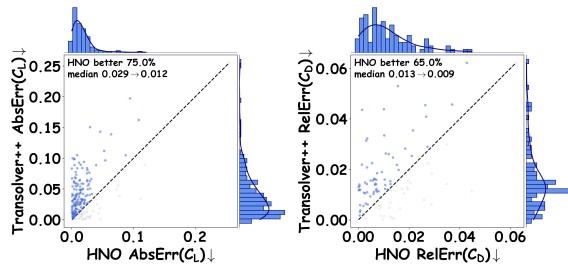

*Figure 7.* Per-sample coefficient errors: ShapeNet Car lift $C_L$ (left) and AirfRANS drag $C_D$ (right).

## 5.3. Results on Large-Scale Benchmarks

We evaluate scalability on two ∼32k-node unstructured CFD benchmarks, ShapeNet Car and AirfRANS. These settings are challenging because irregular geometries require both local boundary fidelity and long-range flow coupling. HNO keeps the Transolver++ backbone, width/depth, token budget, radius graph, and neighbor cap, changing only dot-product logits to stabilized hyperbolic-distance logits. Fig. 6–7 show cleaner residual maps and lower drag/lift errors than Transolver++ and RNO; details are in Appendix I. Because graph construction, compression, and neighborhood cap are fixed, the gains cannot be attributed to a larger receptive graph or extra engineering. They instead reflect how hyperbolic interaction geometry redistributes errors near vehicle bodies and airfoil wakes, where boundary-layer effects interact with global flow structures.

## 5.4. Ablations and Efficiency

We ablate HNO's near–far routing on Darcy (Sec. 4.3). Tab. 3 isolates two design choices under the same parameter count: the interaction metric and the temperature calibration. All ablated variants use the same data split, optimizer schedule, discretization, and backbone configuration, so the comparison changes only the routing rule while holding the training protocol fixed. Replacing hyperbolic distance with Euclidean logits raises error by 10.5%, showing that the gain comes from the geometry used to form the kernel logits rather than from a generic Gibbs-style kernel. Fixing

*Table 3.* Darcy ablations and efficiency. Top: error/params; bottom: memory/runtime.

| *Accuracy under matched HNO ablations* | | |
|---|---|---|
| METHOD | REL. $\ell_2$ ↓ | PARAMS↓ |
| HNO (OURS) | **0.00446** 0.0% | **0.82M** 0.0% |
| —HYPERBOLIC (EUCLID) | 0.00493 +10.5% | **0.82M** +0.0% |
| —TEMPERATURE (FIXED $\tau$) | 0.00464 +4.0% | **0.82M** +0.0% |
| TRANSOLVER | 0.00567 +27.1% | 2.83M +246.3% |
| TRANSOLVER++ | 0.00560 +25.6% | 2.84M +246.3% |
| RNO | 0.00535 +19.7% | 2.83M +245.1% |

| *Runtime and memory under the same benchmark protocol* | | | |
|---|---|---|---|
| METHOD | VRAM (GB)↓ | TRAIN (H)↓ | INFER (MS/B)↓ |
| HNO (OURS) | 0.227 0.0% | 0.73 0.0% | 4.47 0.0% |
| —HYPERBOLIC (EUCLID) | 0.227 +0.0% | 0.67 -8.2% | **4.16** -6.9% |
| —TEMPERATURE (FIXED $\tau$) | **0.207** -8.8% | **0.57** -21.9% | 4.49 +0.4% |
| TRANSOLVER | 2.18 +860% | 2.76 +278% | 26.10 +484% |
| TRANSOLVER++ | 6.07 +2574% | 3.12 +327% | 74.77 +1572% |
| RNO | 2.87 +1164% | 5.26 +621% | 28.40 +535% |

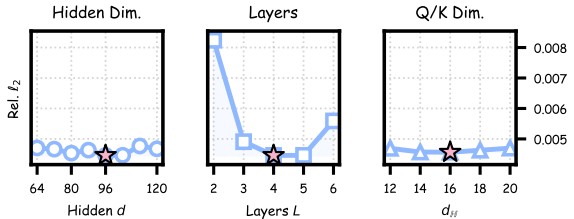

*Figure 8.* Parameter sensitivity on Darcy for hidden dimension $d$, depth $L$, and Q/K dimension $d_{\mathbb{H}}$.

$\tau$ raises error by 4.0%, indicating that adaptive scale calibration provides an additional but secondary benefit. The runtime panel rules out a cost-driven explanation: HNO stays close to its ablations in efficiency while remaining substantially lighter than transformer baselines, indicating that accuracy is not traded for extra computation. Fig. 8 further shows that the selected setting is not a narrow optimum, with stable performance around moderate width/depth choices and a compact hyperbolic Q/K dimension. Together, these controls identify hyperbolic interaction geometry, rather than raw capacity, extra computation, or implementation bookkeeping, as the active ingredient.

*Table 4.* Matched Euclidean routing controls. Errors are in $10^{-3}$; $RF_\Delta$ is RF P90–P10 on Elasticity.

| Accuracy under matched kernels ($\times 10^{-3}$) | | | | |
|---|---|---|---|---|
| Method | Darcy ↓ | Elas. ↓ | Airf. Vol. ↓ | Airf. Surf. ↓ |
| HNO | **4.455** | **3.688** | **12.009** | **4.674** |
| EucDist/Dot | 4.928 | 4.100 | 16.683 | 6.428 |
| TempGate | 4.656 | 5.422 | 71.175 | 8.524 |
| RBFGate | 4.768 | 15.020 | 133.168 | 12.560 |
| Hierarchy induced by each scale coordinate | | | | |
| Method | Scale | $\rho_{RF}$ | $RF_\Delta$ ↑ | Mean RF ↑ |
| HNO | radius | -0.638 | **0.275** | **0.331** |
| EucDist | q-norm | 0.082 | 0.182 | 0.199 |
| TempGate | $\tau$ | 0.638 | 0.197 | 0.215 |
| RBFGate | $\sigma$ | 0.646 | 0.241 | 0.192 |

## 5.5. Additional Analysis

**Geometry Versus Euclidean Routing Controls.** The matched controls in Tab. 4 test whether HNO's gains come from hyperbolic geometry itself or from giving attention a learnable notion of scale. EucDist/Dot replaces hyperbolic distance with Euclidean dot-product-style routing, while TempGate and RBFGate add explicit Euclidean scale variables through a learned temperature or radial-basis width. They can still learn to make some queries broader and others more local, as reflected by their nontrivial $|\rho_{RF}|$ values.

HNO nevertheless keeps the best accuracy on the two PDEBench datasets and both AirfRANS metrics, while attaining the largest mean RF and RF spread on Elasticity. This means that adding a scalar scale gate to Euclidean attention is not enough to recover the same behavior. The results support the specific role of hyperbolic radius: it acts as a learned scale coordinate, while exponential volume growth gives the model more room to organize multiple global-to-local interaction ranges (Lemma D.1).

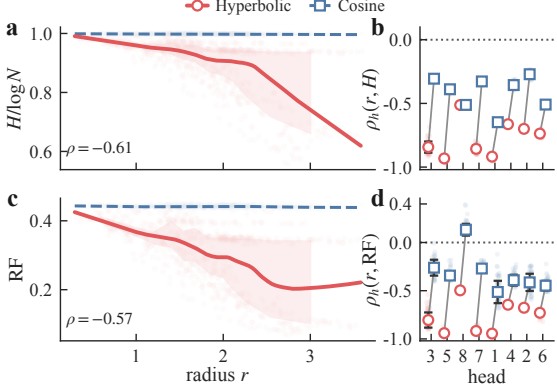

*Figure 9.* **Mechanism verification on Elasticity.** We relate hyperbolic query radius $r$ to normalized attention entropy $H/\log N$ and physical receptive-field span RF. Larger radius indicates more local attention, giving lower entropy and smaller RF. Panels (a,c) show overall Spearman correlations across tokens, while panels (b,d) show per-head correlations $\rho_h$, confirming that the radius–locality trend holds across attention heads.

**Mechanism Verification.** Sec. 4.3 predicts radius-driven scale separation under the Gibbs kernel: larger radii induce local attention, while smaller radii remain more global. On Elasticity, each query radius $r_i$ (Eq. (13)) is compared with normalized attention entropy and attention span $RF_i := \mathbb{E}_{j \sim \alpha_i} \|x_j - b_i\|_2$. Because the diagnostic is post hoc and the cosine baseline keeps query/key directions while removing distance-radius coupling, it tests whether hyperbolic coordinates organize attention beyond ordinary angular similarity in the learned operator.

Fig. 9 shows consistent negative correlations between radius, entropy, and RF, while the cosine-similarity control is much weaker. The head-wise trends support the same near–far mechanism: learned radius separates broad global aggregators from local specialists. The effect appears at the head level rather than only after pooling, indicating multiple operating scales rather than a single locality knob.

*Table 5.* Causal radius interventions on Darcy.

| Method | Rel. $\ell_2$ ↓ |
|---|---|
| HNO (baseline) | **0.00446** |
| Fixed norm (Q/K) | 0.03269 |
| Shuffled norm (Q/K) | 0.03328 |

**Causal Radius Interventions.** We intervene on Q/K norms at inference while preserving angular directions and the value pathway. Fixing the norm collapses all tokens to a common radius; shuffling preserves the marginal norm distribution but breaks its assignment to individual samples and positions. Both interventions sharply degrade Darcy accuracy (Tab. 5), supporting radius as a functional control variable for near–far routing. Since angular directions, value features, and under shuffling the norm histogram remain available, the loss points to the sample- and token-specific radial hierarchy rather than generic feature destruction.

## 6. Conclusion

HNO uses stabilized hyperbolic-distance logits and learned radii to organize neural-operator attention into hierarchical near–far interactions. This design lets the model assign different tokens and heads to different interaction ranges, so local geometric details and long-range dependencies can be handled within one attention mechanism. Across six standard PDE benchmarks and two large-scale CFD tasks, HNO achieves the lowest error among the evaluated methods while remaining parameter- and memory-efficient. The toy tree-kernel study isolates the geometric bias, the Euclidean routing controls show that scalar scale gates alone do not explain the gains, and the causal-radius interventions verify that the learned radii are functionally important. Together, these results support hyperbolic organization as a practical mechanism for neural operators on multiscale and irregular scientific domains.

## Acknowledgements

This work was supported in part by the Natural Science Foundation of Zhejiang Province under Grant Nos. LMS25F020006 and LZ24F030012, the China Postdoctoral Science Foundation under Grant No. 2024M762911, and the National Natural Science Foundation of China under Grant Nos. 62506337 and 62276232. We thank Prof. Yan Wang at Tsinghua AIR for providing computational support.

## Impact Statement

This work develops a supervised neural-operator architecture for accelerating simulations of parametric PDEs. Potential benefits include faster surrogate modeling for scientific and engineering design, especially on irregular or large-scale meshes. The method does not enforce physical constraints such as conservation laws by construction, so deployment in safety-critical simulation workflows should include validation against solvers and domain-specific checks. The experiments use scientific benchmark data and do not involve human subjects or personal data; we do not identify societal harms beyond general risks from inaccurate surrogate predictions if used without verification.

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

# Supplemental Material

## Contents

## Notation (Quick Reference)

| Symbol | Meaning |
| --- | --- |
| $D \subset \mathbb{R}^d$ | Spatial domain; $\mu$ is the base measure on $D$; $\mu(D)$ its total mass. |
| $\nu, \mathbb{E}_{a \sim \nu}$ | Data distribution over inputs and the associated expectation in the operator-learning objective. |
| $a(\cdot), u(\cdot)$ | Input field and target solution field; $\mathcal{G}^\dagger : \mathcal{X} \to \mathcal{Y}$ is the ground-truth operator. |
| $v_t(\cdot)$ | Hidden state at layer $t$ in the neural operator recursion; $v(\cdot)$ is a generic function argument of an integral operator. |
| $K_a(x, y), \mathcal{K}_a$ | Abstract (data-dependent) kernel and associated integral operator in Motivation: $(\mathcal{K}_a v)(x) = \int_D K_a(x, y) \, v(y) \, d\mu(y)$; later parameterized by $\kappa(x, y; a)$. |
| $\alpha \in \mathbb{N}^d$ | Multi-index used in the far-field/Taylor expansion (e.g., $\partial_y^\alpha$ and moment $M_\alpha(C)$). |
| $\rho, \eta$ | Far-field separation distance and separation ratio $\eta = r/\rho$ (with $r$ the source-cluster radius) in the near–far expansion analysis. |
| $q(\cdot; a), k(\cdot; a)$ | Query/key feature maps (Euclidean, in $\mathbb{R}^m$) produced from input $a$. |
| $\Pi(\cdot)$ | Lorentz (hyperboloid) lift: $\Pi(z) = (\sqrt{1 + \|z\|^2}, z) \in \mathbb{H}^m$. |
| $\langle \cdot, \cdot \rangle_M$ | Minkowski inner product; induces hyperbolic distance $d_\mathbb{H}$. |
| $p_x, m_y$ | Lifted hyperbolic query/key points: $p_x = \Pi(q(x; a))$, $m_y = \Pi(k(y; a))$. |
| $r(z), r_u, r_v$ | Hyperbolic radii. For $z \in \mathbb{R}^m$, $r(z) = d_\mathbb{H}(o, \Pi(z)) = \mathrm{asinh}(\|z\|)$. For $u, v \in \mathbb{H}^m$, $r_u = d_\mathbb{H}(o, u)$ and $r_v = d_\mathbb{H}(o, v)$, with $o = (1, 0, \ldots, 0)$. |
| $\theta(u, v)$ | For $u, v \in \mathbb{H}^m$ with spatial parts $\bar{u} = (u_1, \ldots, u_m)$ and $\bar{v} = (v_1, \ldots, v_m)$, define $\theta(u, v) = \arccos\big(\langle \bar{u}, \bar{v} \rangle / (\|\bar{u}\| \|\bar{v}\|)\big) \in [0, \pi]$. |
| $d_\epsilon(\cdot, \cdot), \tau$ | Stabilized distance and temperature in logits; $\epsilon > 0$ clamps arcosh near 1. |
| $\kappa(x, y; a), Z(x; a)$ | Normalized kernel and normalizer in Eq. (9). When $a$ is fixed, we write $\kappa(x, y)$ and $Z(x)$ for brevity. |
| $\mathcal{K}_\kappa, \mathcal{K}$ | Scalar kernel operator and full operator with projections $W_V, W_O$. |
| $\rho, \rho_h$ | Spearman rank correlations used in mechanism analysis: $\rho$ is computed over all heads×tokens, while $\rho_h$ is computed within each head. |
| $\{(x_j, w_j)\}_{j=1}^N, h$ | Nyström nodes and quadrature weights; $h$ is mesh size used in discretization bounds. |
| $P_N$ | Sampling operator: $(P_N f)_j = f(x_j)$. |
| $\mathcal{L}, \lambda_k$ | Graph Laplacian (toy tree) and its eigenvalues in the discrete smoothing operator. |
| $c_0, \tau_{\min}, R$ | Lower bound on $Z$, temperature lower bound, and radius bound used in Schur-type and Lipschitz estimates. |

## A. Hyperbolic Background

### A.1. Hyperbolic Geometry Preliminaries (Lorentz Model and Conventions)

Hyperbolic geometry is a standard continuous surrogate for hierarchical/tree-like structure: trees can be embedded into hyperbolic space with low distortion, and the exponential volume growth matches branching combinatorics (Sarkar, 2011; Nickel & Kiela, 2017; Sala et al., 2018). We summarize the Lorentz-model conventions used in this paper.

Let $\mathbb{R}^{m+1}$ be equipped with the Minkowski bilinear form $\langle x, y \rangle_M := -x_0 y_0 + \sum_{i=1}^m x_i y_i$. Define the hyperboloid model of curvature $-1$:

$$\mathbb{H}^m = \{x \in \mathbb{R}^{m+1} : \langle x, x \rangle_M = -1, \ x_0 > 0\}.$$

For $x \in \mathbb{H}^m$, the tangent space is $T_x \mathbb{H}^m = \{v \in \mathbb{R}^{m+1} : \langle v, x \rangle_M = 0\}$, and the Riemannian metric is $g_x(u, v) = \langle u, v \rangle_M$ on $T_x \mathbb{H}^m$ (positive definite).

**Hyperboloid Lift.** For $z \in \mathbb{R}^m$, define the hyperboloid lift

$$\Pi(z) := \big(\sqrt{1 + \|z\|^2}, \, z\big) \in \mathbb{R}^{m+1}. \tag{17}$$

One verifies that $\langle \Pi(z), \Pi(z) \rangle_M = -(1 + \|z\|^2) + \|z\|^2 = -1$ and $\Pi(z)_0 = \sqrt{1 + \|z\|^2} > 0$, so $\Pi(z) \in \mathbb{H}^m$. This closed-form map is used to lift Euclidean feature vectors to the hyperboloid when constructing the hyperbolic attention kernel.

**Proposition A.1** (Distance formula). *For any $x, y \in \mathbb{H}^m$, one has $-\langle x, y \rangle_M \geq 1$ and*

$$d_\mathbb{H}(x, y) = \mathrm{arcosh}\big(-\langle x, y \rangle_M\big).$$

*Proof.* Let $o = (1, 0, \ldots, 0) \in \mathbb{H}^m$. It is standard that the proper orthochronous Lorentz group $SO^+(1, m)$ acts transitively on $\mathbb{H}^m$ by isometries and preserves $\langle \cdot, \cdot \rangle_M$. Hence there exists an isometry $g \in SO^+(1, m)$ such that $gx = o$. Then

$$d_{\mathbb{H}}(x, y) = d_{\mathbb{H}}(gx, gy) = d_{\mathbb{H}}(o, gy), \qquad \langle x, y \rangle_M = \langle gx, gy \rangle_M = \langle o, gy \rangle_M.$$

By another isometry acting as a spatial rotation on $(x_1, \ldots, x_m)$, we may assume $gy = (y_0', y_1', 0, \ldots, 0)$ with $y_0' > 0$ and $-(y_0')^2 + (y_1')^2 = -1$. Thus there exists a unique $r \geq 0$ such that $(y_0', y_1') = (\cosh r, \sinh r)$.

Consider the curve $\gamma : [0, r] \to \mathbb{H}^m$ defined by $\gamma(t) = (\cosh t, \sinh t, 0, \ldots, 0)$. A direct computation gives $\langle \gamma'(t), \gamma'(t) \rangle_M = 1$, so $\gamma$ is unit-speed. Moreover, $\gamma$ lies in the intersection of $\mathbb{H}^m$ with the 2D plane spanned by $o$ and $gy$, and such intersections are geodesics in the hyperboloid model; hence $\gamma$ is the unique minimizing geodesic from $o$ to $gy$ and

$$d_{\mathbb{H}}(o, gy) = \text{Length}(\gamma) = \int_0^r \|\gamma'(t)\| \, dt = r.$$

Finally,

$$-\langle o, gy \rangle_M = -\langle (1, 0, \ldots, 0), (y_0', y_1', 0, \ldots, 0) \rangle_M = y_0' = \cosh r = \cosh(d_{\mathbb{H}}(o, gy)).$$

Applying $\text{arcosh}$ and using invariance under $g$ yields the claimed formula, and also implies $-\langle x, y \rangle_M \geq 1$. $\qquad\square$

**Coordinate Convention in Fig. 1C.** The coordinate $x_0$ is the time-like Minkowski coordinate imposed by $\langle x, x \rangle_M = -1$; it is not physical time. Coordinates $x_1, \ldots, x_m$ are space-like. Geodesics appear curved in Euclidean drawings because they are plane sections of the hyperboloid.

### A.2. Hyperbolic Deep Learning Related Work

Hyperbolic geometry has become a standard tool for representation learning when data exhibit latent hierarchies, power-law degree distributions, or scale-free growth. Early work established that hyperbolic spaces can embed tree-like metrics with low distortion and high parameter efficiency, popularizing Riemannian optimization in the Poincaré ball for hierarchical embeddings (Nickel & Kiela, 2017). Subsequent work advocated the Lorentz (hyperboloid) model as a numerically stable alternative and developed practical training recipes for learning continuous hierarchies with improved stability in high dimensions (Nickel & Kiela, 2018). Beyond metric-preserving embeddings, hyperbolic order structures were introduced to model asymmetric relations such as entailment and hypernymy through cone-based constructions that align with hierarchical partial orders (Ganea et al., 2018a). These foundations motivate using hyperbolic manifolds not merely as an embedding space, but as a geometry that matches the exponential expansion patterns of semantic and relational hierarchies.

**Hyperbolic neural networks and operations.** A key step from embeddings to deep learning is defining analogues of Euclidean layers (linear maps, nonlinearities, pooling, normalization) on manifolds while controlling mapping errors and numerical issues. Hyperbolic Neural Networks (HNN) formulated core neural operations in hyperbolic space with Riemannian optimization, demonstrating that hyperbolic feature spaces can provide higher effective capacity than Euclidean ones at comparable parameter budgets (Ganea et al., 2018b). In parallel, hyperbolic attention mechanisms were proposed by replacing Euclidean dot-product similarity with hyperbolic geometry-aware compatibility functions, enabling attention to exploit the "more room" property of hyperbolic spaces as the number of objects grows (Gulcehre et al., 2018). Hyperbolic word embedding models further showed that distributional semantics can benefit from hyperbolic structure and can be connected to probabilistic representations (e.g., Gaussian embeddings) while improving unsupervised performance on similarity, analogy, and hypernymy detection (Tifrea et al., 2018). Collectively, these works establish a reusable toolbox: (i) geometry-aware distance/similarity, (ii) manifold-compatible transformations, and (iii) training procedures that keep optimization stable under negative curvature.

**Hyperbolic graph deep learning.** Graphs frequently exhibit hierarchical organization and scale-free statistics, making them a canonical domain for hyperbolic learning. Hyperbolic Graph Convolutional Networks (HGCN) provided one of the first inductive GCN formulations in hyperbolic space, deriving message passing and feature transformation in the hyperboloid model and introducing trainable curvature across layers (Chami et al., 2019). Complementarily, Hyperbolic Graph Neural Networks (HGNN) proposed architectures for learning on Riemannian manifolds via differentiable exponential and logarithmic maps, demonstrating improved performance on benchmark graphs when hyperbolic geometry better matches the underlying structural bias (Liu et al., 2019). These lines of work highlight a recurring theme: for relational data with

implicit hierarchy, the main gains often come from reducing representational distortion and enabling compact embeddings, rather than merely adding depth.

**Hyperbolic Transformers, scalable attention, and foundation models.** More recent work pushes hyperbolic geometry into Transformer-style architectures. Hypformer introduced a more complete hyperbolic Transformer built in the Lorentz model, defining missing modules (e.g., linear transformations, normalization-like operations, and hyperbolic linear attention) and demonstrating scalability on large graph benchmarks (Yang et al., 2024). At the foundation-model scale, HELM reported billion-parameter fully hyperbolic large language models and proposed a Mixture-of-Curvature Experts mechanism to better match heterogeneous curvature patterns in token representations, together with hyperbolic analogues of key LLM components such as rotary positional encoding and RMS normalization (He et al., 2025). These developments indicate an emerging direction: geometry is not only an inductive bias for hierarchy, but can also be integrated as a system-level design choice affecting stability, scaling, and modularity in modern architectures.

## B. Interaction Compression and Non-uniform Coupling

The near–far view in Section 3 treats compression as a property of the interaction operator, rather than a choice of hidden width. In a discretization with $N$ nodes, a standard attention layer computes a dense $N \times N$ matrix of pairwise scores and applies a row-wise normalization to update values, resulting in $O(N^2)$ interactions. Reducing hidden width changes representation capacity but still retains dense coupling and $O(N^2)$ pairwise interactions.

**Interaction-level compression and complexity.** HNO reduces interaction cost by routing node-to-node coupling through $M \ll N$ representative tokens, yielding $O(NM + M^2)$ instead of $O(N^2)$. This complexity reduction comes from the token bottleneck rather than the specific choice of logits. The logits mainly control non-uniform coupling and locality, while the asymptotic cost is dictated by the $N \leftrightarrow M$ routing structure. On point clouds these are learned latent tokens, while on grids and meshes they are patch tokens. Equivalently, conditioned on routing weights, the induced node-to-node interaction factors through an $M$-dimensional token space (a Nyström-like latent bottleneck).

**Latent-mediated routing form.** For point clouds, one layer can be written in a latent-bottleneck form. Let $V \in \mathbb{R}^{N \times d_v}$ denote node values and let $T \in \mathbb{R}^{M \times d_v}$ denote latent tokens. Using row-wise normalization as in Eq. (11) with stabilized-distance logits, we obtain two cross-attention routing matrices $A \in \mathbb{R}^{M \times N}$ and $C \in \mathbb{R}^{N \times M}$, and a token mixing matrix $B \in \mathbb{R}^{M \times M}$. We apply

$$T = AV, \qquad T' = BT, \qquad \tilde{V} = V + CT'. \tag{18}$$

This makes explicit that node-to-node coupling is mediated through $M$ tokens. Here $A$ is row-normalized over nodes for each token query, $C$ is row-normalized over tokens for each node query, and $B$ is row-normalized over tokens. When the attention weights $A, C, B$ are fixed, the value update is linear in $V$ and can be written as $\tilde{V} = V + C B (AV) = (I + CBA)V$. Hence the conditioned interaction operator $CBA$ has rank at most $M$. In practice, $A, C, B$ depend on logits computed from queries and keys, so the full layer is nonlinear and the rank statement is a conditional view of the interaction pathway. This requires two cross-attention passes of size $N \times M$ and one token self-attention of size $M \times M$, yielding $O(NM + M^2)$ interactions. For regular grids and structured meshes, patchification and unpatching are local linear maps with $O(N)$ cost at fixed patch size. Attention is evaluated in the patch-token space with $O(M^2)$ cost, yielding $O(N + M^2)$ per layer. On a 2D grid with a fixed $p \times p$ patch size, $M \approx N/p^2$, so the global mixing cost scales as $O((N/p^2)^2)$ while patchification remains $O(N)$. This refers to the global token-to-token mixing term per layer; local lifting and unlifting and pointwise updates remain $O(N)$. Compared to node-wise $O(N^2)$, this reduces cost when tokenization yields $M \ll N$.

**Non-uniform coupling, not uniform mixing.** The bottleneck explains why the layer is cheaper, while the hyperbolic kernel determines which interactions are emphasized. In HNO, stabilized Lorentz-distance logits define the normalized weights in $A$, $C$, and $B$, inducing strongly heterogeneous routing and controllable locality across tokens and heads. This differs from schemes whose aggregation weights are fixed or nearly uniform (e.g., constant-kernel averaging or uniform assignments), since geometry-conditioned routing can produce both local and global receptive fields within the same $N \leftrightarrow M$ bottleneck. Uniform mixing would correspond to a row-normalized constant kernel. This yields uniform averaging over all nodes, a row-stochastic diffusion on the node set. Empirically, attention in HNO is highly heterogeneous. For a query token with attention weights $\{\alpha_j\}_{j=1}^N$, define the raw entropy $H_{\text{raw}} = -\sum_{j=1}^N \alpha_j \log \alpha_j$ and the normalized entropy $H_{\text{norm}} = H_{\text{raw}}/\log N \in [0, 1]$. The perplexity is $P = \exp(H_{\text{raw}}) = N^{H_{\text{norm}}}$, which can be interpreted as an effective support size. Let $b = \sum_{j=1}^N \alpha_j x_j$ and define the span RF $= \sum_{j=1}^N \alpha_j \|x_j - b\|_2$ in physical coordinate space. On Elasticity with $N = 972$ points and $M = 96$ latent queries, computed over 100 test samples, $H_{\text{norm}}$ ranges from 0.56 to 1.00 across

heads and latents. Equivalently, the perplexity $P$ ranges from about $47$ to $967$ nodes. Moreover, hyperbolic radius controls locality. We define the radius of a lifted embedding $u \in \mathbb{H}^m$ as $r(u) = d_{\mathbb{H}}(o, u) = \text{arcosh}(u_0)$ under the Lorentz model. The Spearman correlation between hyperbolic radius and $H_{\text{norm}}$ is about $-0.61$, and the Spearman correlation between hyperbolic radius and RF is about $-0.57$. Tokens in the bottom $20\%$ of radii have average perplexity about $769$ and $\text{RF} \approx 0.38$, while tokens in the top $20\%$ have average perplexity about $386$ and $\text{RF} \approx 0.23$. Full definitions and additional statistics are provided in Appendix K.3. Additional visualizations are shown in Appendix K.2.

**Parameter efficiency (empirical).** Empirically, interaction compression is often accompanied by improved parameter efficiency. For example on Darcy, HNO attains lower error than Transolver while using $0.82$M parameters rather than $2.83$M, a reduction by about $3.45\times$.

## C. Unsuccessful Hyperbolic Variants (What Did Not Work)

We additionally evaluated multiple hyperbolic operator families aimed at realizing near–far routing and far-field compression in hyperbolic space beyond our final HNO kernel in Eq. (9). Tab. 6 reports representative Darcy runs with the same data split and optimization setup; we additionally report executed epochs since some variants were stopped early or rerun with a shorter budget, spanning manifold-valued backbones, learned hierarchical routers, and token-compression prototypes. These alternatives either underperformed HNO under comparable settings or terminated early. These results delineate the design space and are consistent with applying hyperbolic geometry to interaction weights while avoiding premature compression.

*Table 6.* Catalog of additional hyperbolic variants on Darcy (mean relative $\ell_2$ error; lower is better). We report the parameter count, the best error observed with the same data split and optimization setup, and the executed training epochs (executed/scheduled). Status reports the execution outcome: completed (ran the full schedule), early-stopped (terminated before schedule due to numerical or engineering issues), underperformed (completed but remained above the anchor HNO), or crashed. Transolver and FNO are included as non-hyperbolic references.

| Variant (what changed) | Params | Best error | Epochs | Status |
|---|---|---|---|---|
| **Anchors and references** | | | | |
| **HNO (ours)** | 0.82M | **0.00446** | 500/500 | completed |
| **Transolver** (2024) | 2.83M | 0.00565 | 500/500 | reference |
| **FNO** (2021) | 2.38M | 0.00870 | 500/500 | reference |
| **Irregular-to-grid patchification (sparse/soft patches)** | | | | |
| **H-SparsePatch**: point→grid (bilinear), sparse patchification, patch attention, grid→point sampling (routing logits use Euclidean $\ell_2$) | 1.11M | 0.00744 | 500/500 | underperformed |
| **H-SSPatch (large)**: soft assignment to $M$ learned patch centers, patch attention, decode with shared assignment weights (routing logits use Euclidean $\ell_2$) | 0.95M | 0.01235 | 320/320 | underperformed |
| **Fully manifold-valued backbones** | | | | |
| **Hyperbolic Attention**: manifold-valued attention backbone (Lorentz-valued physics-attention blocks) | 2.83M | 0.00817 | 500/500 | underperformed |
| **Hyperbolic Attention (replica)**: second run with the same backbone | 2.83M | 0.00882 | 500/500 | underperformed |
| **HELM (native)**: fully manifold-valued pipeline with Lorentz linear/attention layers | 2.88M | 0.202 | 12/500 | early-stopped |
| **HELM (AMG)**: manifold-valued backbone with soft physics pooling/unpooling | 3.21M | 0.0392 | 500/500 | underperformed |
| **Hyperbolic SSM (2D)**: Lorentz linear layers inside a state-space backbone | 0.86M | 0.151 | 44/500 | early-stopped |
| **Hyperbolic SSM (2D, 3-layer)**: deeper state-space backbone | 1.14M | 0.258 | 2/500 | early-stopped |
| **Scale-space / tree-routing operators with hyperbolic routing** | | | | |
| **Scale-Space Tree GAOT**: hyperbolic tree router over 289 patches (depth=4) + transformer processor | 21.48M | 0.0223 | 500/500 | underperformed |
| **Scale-Space Tree GAOT + CNN (v1)**: CNN spatial encoder/decoder + contrastive regularizers | 20.86M | 0.0241 | 111/500 | early-stopped |
| **Scale-Space Tree GAOT (compact)**: hyperbolic encoder/decoder + patchification + hyperbolic tree router | 5.46M | 0.0254 | 320/320 | underperformed |
| **Scale-Space Tree GAOT (compact, replica)**: second run with the same configuration | 5.46M | 0.01049 | 320/320 | underperformed |
| **Scale-Space Tree GAOT (compact, lr)**: reduced learning rate ($10^{-3} \to 5 \times 10^{-4}$) | 5.46M | 0.01126 | 320/320 | underperformed |
| **Scale-Space Tree GAOT (deeper processor)**: increased transformer depth ($4 \to 6$ layers) | 8.03M | 0.02154 | 320/320 | underperformed |
| **Scale-Space Tree GAOT (deeper processor)**: increased transformer depth ($4 \to 8$ layers) | 10.60M | 0.03274 | 320/320 | underperformed |
| **MAGNO + hyperbolic fusion**: multi-scale hyperbolic coordinate embedding + neighbor aggregation + fusion | 0.37M | – | 0/500 | crashed |
| **Other operator prototypes** | | | | |
| **Hyperbolic Slice Attention**: slice-based token aggregation with hyperbolic-distance attention | 2.84M | 0.00708 | 320/320 | underperformed |
| **H-Proto**: prototype-based token compression with hyperbolic attention | 1.49M | 0.01047 | 320/320 | underperformed |
| **H-FNO**: spectral backbone with hyperbolic routing/modulation | 1.67M | 0.00855 | 320/320 | underperformed |

**C.1. Family-Level Designs and Failure Analysis (Far-Field Compression View)**

**Connection to Our Theory (Near–Far Compression).** Section 3 frames operator learning as a near–far decomposition: near-field interactions require high-fidelity resolution, while far-field interactions can be compressed once source/target regions are sufficiently separated. This mirrors classical fast solvers (Greengard & Rokhlin, 1987; Hackbusch, 2015), where far-field blocks admit low-rank/expansion-based approximations and are compressed only for *admissible* (well-separated) cluster pairs. A typical admissibility condition between two clusters $C$ and $D$ is

$$\max\big(\mathrm{diam}(C), \mathrm{diam}(D)\big) \leq \eta\,\mathrm{dist}(C, D), \qquad \eta \in (0, 1), \tag{19}$$

and blocks that violate (19) are refined (or treated as near-field) rather than compressed. We cite (19) only as a classical criterion indicating when compression is provably safe in hierarchical solvers; HNO does not explicitly enforce such a geometric test. From this view, the dominant failure mode is *misplacing compression*: compression is safe only for well-separated interactions, and compressing before near-field content is resolved introduces an irreversible bias.

**A 2×2 Design Space (What We Observed).** We found that the variants in Tab. 6 can be organized along two axes: *(A) how compression is applied* (early $N{\to}M$ aggregation that averages values vs. token-mediated compression of the interaction pathway), and *(B) where hyperbolic geometry is applied* (only to interaction *weights*/routing logits vs. to the full *value/backbone* pipeline). HNO sits in the quadrant of *token-mediated interaction compression + hyperbolic geometry only for logits/weights*. Below, we summarize each family in terms of the design axis it violates.

**H-SparsePatch: Point→Grid Rasterization + Patch Attention.** This variant attempted to unify irregular points and regular grids by (i) rasterizing point features to a fixed $G{\times}G$ grid via bilinear scatter (producing a grid mask), (ii) patchifying the grid with a strided convolutional patch embedding, (iii) applying patch-to-patch attention, and (iv) mapping the result back to points via bilinear sampling. We include an Euclidean-logits ablation in this family (same architecture; routing logits use Euclidean $\ell_2$) to isolate the effect of rasterization/patchification itself from the effect of geometry-aware routing. **Observed symptom:** The model underperformed HNO on Darcy (Tab. 6) and on Elasticity (see additional runs below). **Why it failed:** This introduces an early discretization bottleneck before any near–far routing: bilinear scatter/sampling can alias high-frequency content and introduce boundary bias through the grid mask. Near-field details can be irreversibly distorted, so later long-range routing cannot fully recover them; this is consistent with the systematic underperformance in Tab. 6.

**H-SSPatch: Soft Spatial Patching (Learned Centers) + Patch Attention.** Instead of patchifying a grid, this variant learned $M$ patch centers in normalized coordinate space and produced soft assignment weights $w_{i\to m} \propto \exp(-\|x_i - c_m\|/\tau)$; patch tokens were aggregated as $z_m = \sum_i w_{i\to m} h_i$ and then processed by patch-to-patch attention, followed by decoding back to points with the same weights. As above, we report an Euclidean-logits ablation (same architecture; routing logits use Euclidean $\ell_2$) to isolate the soft patching effect. **Observed symptom:** The method did not match HNO on Darcy and Elasticity under comparable settings. **Why it failed:** This variant performs token compression at the first step by collapsing $N$ points into $M$ soft patches before a near-field-preserving representation is formed. Because centers (and a global temperature $\tau$) impose a fixed geometric scale without an explicit admissibility test, points from distinct local neighborhoods can be mixed across clusters in a way that is non-admissible under (19), effectively smoothing fine-scale modes.

**Fully Manifold-Valued Backbones (Hyperbolic Attention / HELM / Hyperbolic SSM (2D)).** These variants moved the *entire* hidden-state pipeline onto a Lorentz manifold: attention/SSM blocks required repeated exp/log maps, projection back to the hyperboloid, and manifold-aware linear/normalization layers. **Observed symptom:** Some runs terminated early, while completed runs consistently underperformed HNO even with higher parameter counts (Tab. 6). **Why it failed:** Our near–far argument only requires curvature to shape interaction weights (kernels); it does not require manifold-valued value pipelines. Making the full backbone manifold-valued adds heavy exp/log/projection overhead and radius-dependent gradient scaling, which can destabilize optimization without improving where compression occurs.

**Tree/Scale-Space Operators (Scale-Space Tree Variants).** These variants replaced patchified attention with explicit hierarchical routing: tokens were aggregated into multiscale representations and routed by learned tree/scale routers, with hyperbolic distances used to compute routing probabilities (sometimes with additional contrastive losses). **Observed symptom:** Despite being expressive (often $> 20$M parameters), they underperformed HNO and some configurations

terminated early (Tab. 6). **Why it failed:** Unlike FMM/$\mathcal{H}$-matrices—where only admissible (well-separated) blocks are compressed (e.g., (19))—and truncation error decays with separation—these routers had no explicit structural guarantee for what may be safely summarized. As a result, routing can summarize interactions that should remain near-field, introducing an irreversible compression error; this is consistent with the systematic underperformance in Tab. 6. On PDEBench this is further complicated because effective long-range coupling can be input-dependent rather than purely geometric.

**Other Prototypes (Slice Assignment / Prototypes / Spectral).**  Hyperbolic Slice Attention used slice-based token aggregation with hyperbolic-distance-based routing to slice centers. H-Proto compressed tokens via hyperbolic prototype assignment and then ran attention in prototype space. H-FNO replaced the backbone with FFT-based spectral blocks and used hyperbolic distances to modulate frequency weights (plus a hyperbolic router). **Observed symptom:** Hyperbolic Slice Attention was sensitive and did not match HNO under comparable settings; H-Proto and H-FNO were stable but substantially worse than HNO on Darcy. **Why it failed:** Slice/prototype assignment is an explicit token-compression step; when assignments collapse (e.g., overly diffuse or overly peaky) or are misaligned with PDE locality, they can mix nonlocal interactions and degrade near-field fidelity, consistent with Tab. 6. For spectral backbones, FNO already achieves global mixing via a basis change, so adding hyperbolic routing/modulation changes interaction geometry without matching the spectral inductive bias and typically degrades accuracy.

**Additional Runs Beyond Darcy.**  On irregular geometries (Elasticity), we also tested (i) **H-SparsePatch** (grid rasterization + patch attention) and (ii) **H-SSPatch** (learned spatial patch centers + patch attention), which achieved best rel. errors $\approx 4.22 \times 10^{-2}$ and $\approx 1.85 \times 10^{-2}$ respectively (vs. HNO at $3.69 \times 10^{-3}$). We also prototyped **H-MAGNO** (MAGNO-style neighbor aggregation + hyperbolic physics attention), but the historical run terminated early under comparable settings, so we did not pursue it further.

**Summary (Why HNO Worked While These Did Not).**  Overall, failures concentrated in two regions of the design space: *(i) early compression* (rasterization, soft patches, slice/prototype assignment), which collapses $N$ inputs into $M$ summaries before near-field structure is represented and is therefore not recoverable; and *(ii) heavy or unstable routing/backbones* (explicit tree routers without an admissibility criterion such as (19), or fully manifold-valued pipelines), which add optimization/geometry overhead without controlling whether aggregation inadvertently averages over near-field structure. HNO keeps values in a stable Euclidean pipeline and applies hyperbolic geometry only to interaction *weights*, yielding heterogeneous routing inside the token-mediated interaction core and empirically avoiding the accuracy loss observed in early geometric-averaging variants.

## D. Motivation Details: Exponential Volume and Far-Field Low Rank

### D.1. Cluster Trees in Hierarchical Solvers

The multilevel hierarchy in Fig. 1**B** is the same object built by fast multipole methods and hierarchical matrix methods: a *cluster tree* obtained by recursively partitioning the discretization. Let $\{x_i\}_{i=1}^N$ denote discretization locations for sources and targets.

**Cluster Tree Construction in FMM.**  FMM builds a rooted spatial partition tree whose nodes are *boxes* that contain subsets of the points. The root is the bounding box that contains all $\{x_i\}$. Each non-leaf box is subdivided into child boxes, and points are assigned to the unique child that contains them. The recursion stops when each leaf box contains at most a prescribed number of points. In two dimensions this subdivision is often into four child boxes and in three dimensions into eight child boxes, but the key property is the multilevel hierarchy, not a fixed branching factor.

**Index-Cluster View.**  Equivalently, each tree node corresponds to an index cluster $C \subset \{1, \ldots, N\}$ given by the points inside a box. The children of $C$ form a partition of $C$, and the leaves form a disjoint partition of $\{1, \ldots, N\}$.

**Near–Far Separation and Routing via Ancestors.**  Given a cluster tree, hierarchical methods organize interactions between two regions by scale. At fine levels, nearby clusters are handled directly to preserve near-field fidelity. At coarser levels, interactions between well-separated clusters are approximated using a small set of coefficients, exploiting the low-rank structure of far-field blocks for asymptotically smooth kernels. This underlies fast multipole methods and hierarchical matrices (Greengard & Rokhlin, 1987; Hackbusch, 2015).

A useful view is that two leaf elements interact "through" their lowest common ancestor: the level of this ancestor determines the scale at which information can be summarized without sacrificing near-field resolution. Algorithmically, this corresponds to an upward pass that aggregates child summaries to parents, interactions between well-separated clusters at an appropriate level, and a downward pass that propagates coarse summaries back to leaves.

## D.2. Exponential Volume Growth (Details for Proposition D.1)

The following standard volume formula for hyperbolic space is a known geometric result (Anderson, 2005); we include the short derivation for completeness.

**Proposition D.1** (Known hyperbolic ball volume and asymptotics). *Let $m \geq 2$, let $B_{\mathbb{H}}(r)$ be the geodesic ball of radius $r$ in $\mathbb{H}^m$, and let $\omega_{m-1} = \mathrm{vol}(\mathbb{S}^{m-1})$. Then*

$$\mathrm{vol}(B_{\mathbb{H}}(r)) = \omega_{m-1} \int_0^r \sinh^{m-1}(t)\, dt.$$

*Moreover,*

$$\lim_{r \to \infty} e^{-(m-1)r}\, \mathrm{vol}(B_{\mathbb{H}}(r)) = \frac{\omega_{m-1}}{(m-1)\, 2^{m-1}}, \quad \text{hence } \mathrm{vol}(B_{\mathbb{H}}(r)) \sim \frac{\omega_{m-1}}{(m-1)\, 2^{m-1}} e^{(m-1)r}.$$

*Proof.* In geodesic polar coordinates $(t, \theta) \in [0, \infty) \times \mathbb{S}^{m-1}$, the hyperbolic metric has the warped-product form $ds^2 = dt^2 + \sinh^2(t)\, d\Omega^2(\theta)$, so the Riemannian volume element is $d\mathrm{vol} = \sinh^{m-1}(t)\, dt\, d\Omega$. Integrating over $\theta$ and $t \in [0, r]$ gives the stated integral formula.

For the asymptotics, define $V(r) := \int_0^r \sinh^{m-1}(t)\, dt$. Then $V(r) \to \infty$ and $V'(r) = \sinh^{m-1}(r)$. By l'Hospital's rule,

$$\lim_{r \to \infty} \frac{V(r)}{e^{(m-1)r}} = \lim_{r \to \infty} \frac{V'(r)}{(m-1)e^{(m-1)r}} = \frac{1}{m-1} \lim_{r \to \infty} \left(\frac{\sinh r}{e^r}\right)^{m-1}.$$

Since $\sinh r = \frac{1}{2}(e^r - e^{-r})$, one has $\lim_{r \to \infty} \sinh r / e^r = 1/2$, hence

$$\lim_{r \to \infty} \frac{V(r)}{e^{(m-1)r}} = \frac{1}{(m-1)\, 2^{m-1}}.$$

Multiplying by $\omega_{m-1}$ yields the claimed limit for $\mathrm{vol}(B_{\mathbb{H}}(r))$ and the asymptotic equivalence. $\square$

## D.3. Detailed Derivation of the Far-Field Expansion

**Assumption D.2** (Derivative decay away from $C$). Let

$$B_{p+1}(x; C) := \sup_{z \in \overline{\mathrm{co}(C \cup \{c\})}} \max_{|\alpha| = p+1} \left| \partial_y^\alpha K_a(x, z) \right|.$$

For each input field $a$, there exists $A_{p+1}(x; a) \in (0, \infty)$ such that for every bounded measurable $C$ and every $c \in \mathbb{R}^d$,

$$B_{p+1}(x; C) \leq \frac{A_{p+1}(x; a)}{\rho^{p+1}} \qquad \text{where } \rho = \mathrm{dist}\big(x, \overline{\mathrm{co}(C \cup \{c\})}\big).$$

**Lemma D.3** (Taylor far-field moment expansion). *Assume that for fixed $x$, the map $y \mapsto K_a(x, y)$ is $(p+1)$-times continuously differentiable on a neighborhood of $\overline{\mathrm{co}(C \cup \{c\})}$ and satisfies the derivative decay condition in Assumption D.2. When the source region $C$ is well-separated from the evaluation point $x$ and the separation ratio satisfies $\eta = r/\rho < 1$, the kernel integral admits a multipole expansion:*

$$\int_C K_a(x, y) s(y)\, dy = \sum_{|\alpha| \leq p} \frac{\partial_y^\alpha K_a(x, c)}{\alpha!} M_\alpha(C) + R_{p+1}(x; C), \tag{20}$$

*where $M_\alpha(C) = \int_C (y - c)^\alpha s(y)\, dy$ are aggregated moments, $c$ is the cluster center, $r$ is the cluster radius, and $\rho = \mathrm{dist}\big(x, \overline{\mathrm{co}(C \cup \{c\})}\big)$. Moreover, the remainder satisfies $|R_{p+1}(x; C)| \leq \mathcal{C}(x, p; a)\eta^{p+1}$, where $\mathcal{C}(x, p; a)$ is derived below.*

**Setup on Measurability, Norms, and Separation.** Let $C \subset \mathbb{R}^d$ be Lebesgue measurable and bounded, let $c \in \mathbb{R}^d$ be fixed, and let $s \in L^1(C)$. Fix a norm $\|\cdot\|$ on $\mathbb{R}^d$. Let $C_{\text{eq}} \geq 1$ be such that $\|v\|_\infty \leq C_{\text{eq}}\|v\|$ for all $v \in \mathbb{R}^d$. Such a constant exists by norm equivalence. We define

$$r = \sup_{y \in C} \|y - c\| < \infty, \qquad \rho = \text{dist}\big(x, \overline{\text{co}(C \cup \{c\})}\big) > 0, \qquad \eta = r/\rho < 1.$$

If $c \in C$, which holds for typical choices of region centers, then $r \leq \text{diam}(C)$ and hence $\eta \leq \text{diam}(C)/\rho$. More generally $r \leq \text{diam}(C) + \text{dist}(c, C)$.

Assume that for fixed $x$, the map $y \mapsto K_a(x, y)$ is $(p + 1)$-times continuously differentiable on an open convex set $U \subset \mathbb{R}^d$ such that

$$\overline{\text{co}(C \cup \{c\})} \subset U.$$

In particular, the segment $[c, y] = \{c + t(y - c) : t \in [0, 1]\}$ lies in $U$ for each $y \in C$. This ensures that multivariate Taylor's theorem with integral remainder along $[c, y]$ applies and that the derivative envelope $B_{p+1}(x; C)$ defined below is finite.

**Taylor Expansion with Integral Remainder.** For a multi-index $\alpha \in \mathbb{N}^d$, write $|\alpha| = \sum_i \alpha_i$, $\alpha! = \prod_i \alpha_i!$, $(y - c)^\alpha = \prod_i (y_i - c_i)^{\alpha_i}$, and $\partial_y^\alpha = \prod_i \partial_{y_i}^{\alpha_i}$. For each $y \in C$, Taylor's theorem around $c$ yields

$$K_a(x, y) = \sum_{|\alpha| \leq p} \frac{1}{\alpha!} \partial_y^\alpha K_a(x, c)(y - c)^\alpha + \sum_{|\alpha| = p+1} \frac{p+1}{\alpha!}(y - c)^\alpha \int_0^1 (1 - t)^p \partial_y^\alpha K_a\big(x, c + t(y - c)\big)\, dt. \tag{21}$$

Define the moments (well-defined since $s \in L^1(C)$ and $C$ is bounded):

$$M_\alpha(C) = \int_C (y - c)^\alpha s(y)\, dy, \qquad |\alpha| \leq p.$$

**Moment Expansion Justified by Tonelli.** Let

$$I(x; C) = \int_C K_a(x, y) s(y)\, dy.$$

Multiplying (21) by $s(y)$ and integrating over $C$, the finite sum over $|\alpha| \leq p$ is immediate.

For later use, we define the order $(p + 1)$ derivative envelope over the relevant compact set.

$$B_{p+1}(x; C) = \sup_{z \in \overline{\text{co}(C \cup \{c\})}} \max_{|\alpha| = p+1} \big|\partial_y^\alpha K_a(x, z)\big| < \infty.$$

Moreover, for $|\alpha| = p + 1$ and $(y, t) \in C \times [0, 1]$,

$$|(y - c)^\alpha| \leq \|y - c\|_\infty^{p+1} \leq C_{\text{eq}}^{p+1}\|y - c\|^{p+1}, \qquad \big|\partial_y^\alpha K_a\big(x, c + t(y - c)\big)\big| \leq B_{p+1}(x; C),$$

so the integrand is dominated by $C_{\text{eq}}^{p+1} B_{p+1}(x; C)\|y - c\|^{p+1}|s(y)|(1 - t)^p$, which is integrable since $C$ is bounded, $s \in L^1(C)$, and $\int_0^1 (1 - t)^p dt < \infty$. Hence Tonelli's theorem applies to the remainder term. Therefore

$$I(x; C) = \sum_{|\alpha| \leq p} \frac{1}{\alpha!} \partial_y^\alpha K_a(x, c) M_\alpha(C) + R_{p+1}(x; C), \tag{22}$$

where the truncation remainder admits the exact representation

$$R_{p+1}(x; C) = \sum_{|\alpha| = p+1} \frac{p+1}{\alpha!} \int_C (y - c)^\alpha s(y) \int_0^1 (1 - t)^p \partial_y^\alpha K_a\big(x, c + t(y - c)\big)\, dt\, dy. \tag{23}$$

**Deterministic Remainder Bound.** Using (23), the bound $|(y-c)^\alpha| \leq \|y-c\|_\infty^{|\alpha|} \leq C_{\text{eq}}^{|\alpha|}\|y-c\|^{|\alpha|}$, and $\int_0^1 (1-t)^p dt = \frac{1}{p+1}$,

$$
\begin{aligned}
|R_{p+1}(x;C)| &\leq \sum_{|\alpha|=p+1} \frac{p+1}{\alpha!} \int_C |(y-c)^\alpha|\,|s(y)| \int_0^1 (1-t)^p \left|\partial_y^\alpha K_a(x, c+t(y-c))\right| dt\, dy \\
&\leq \sum_{|\alpha|=p+1} \frac{p+1}{\alpha!} B_{p+1}(x;C)\, C_{\text{eq}}^{p+1} \int_C \|y-c\|^{p+1}|s(y)| \int_0^1 (1-t)^p\, dt\, dy \\
&= B_{p+1}(x;C)\, C_{\text{eq}}^{p+1} \Big( \sum_{|\alpha|=p+1} \frac{1}{\alpha!} \Big) \int_C \|y-c\|^{p+1}|s(y)|\, dy \\
&\leq B_{p+1}(x;C)\, C_{\text{eq}}^{p+1} \Big( \sum_{|\alpha|=p+1} \frac{1}{\alpha!} \Big) r^{p+1} \int_C |s(y)|\, dy.
\end{aligned}
\tag{24}
$$

The combinatorial factor can be computed *exactly*:

$$
\sum_{|\alpha|=n} \frac{1}{\alpha!} = \frac{d^n}{n!}, \qquad n \in \mathbb{N},
$$

since the coefficient of $t^n$ in $\exp(t)^d = \prod_{i=1}^d \exp(t)$ equals $\sum_{|\alpha|=n} 1/\alpha!$ and also equals $d^n/n!$. Applying this with $n = p+1$ to (24) gives the explicit bound

$$
|R_{p+1}(x;C)| \leq \frac{d^{p+1}}{(p+1)!}\, B_{p+1}(x;C)\, C_{\text{eq}}^{p+1}\, r^{p+1}\, \|s\|_{L^1(C)}.
\tag{25}
$$

**Separation-Rate Form ($\eta^{p+1}$) Under a Kernel Regularity Assumption.** The step from (25) to an $\eta^{p+1}$ rate requires the distance-to-singularity derivative control stated in Assumption D.2.

**Theorem D.4** (Far-field remainder bound). *Under the setup of Lemma D.3, with $C$ bounded, $s \in L^1(C)$, $\rho > 0$, $\eta = r/\rho < 1$, and Assumption D.2, the truncation remainder in (22) satisfies*

$$
|R_{p+1}(x;C)| \leq \mathcal{C}(x,p;a)\, \eta^{p+1},
\tag{26}
$$

*where $\mathcal{C}(x,p;a) := \frac{d^{p+1}}{(p+1)!} A_{p+1}(x;a)\, C_{\text{eq}}^{p+1} \|s\|_{L^1(C)}$ and $\eta := r/\rho$.*

*Proof.* Combining (25) with $r = \eta\rho$ and Assumption D.2 yields the result. $\qquad\square$

**Corollary D.5** (Far-field low rank: rank–accuracy tradeoff). *Let $r_p := |\{\alpha \in \mathbb{N}^d : |\alpha| \leq p\}| = \binom{p+d}{d}$. Define the truncated far-field operator (cf. Eq. (20))*

$$
(T_p s)(x) := \sum_{|\alpha| \leq p} \frac{\partial_y^\alpha K_a(x,c)}{\alpha!}\, M_\alpha(C), \qquad M_\alpha(C) = \int_C (y-c)^\alpha s(y)\, dy.
$$

*Then $T_p$ factors through the $r_p$ moments $\{M_\alpha(C)\}_{|\alpha| \leq p}$ and is therefore a rank-$r_p$ separated approximation of the far-field block. Moreover, the approximation error is controlled by Theorem D.4:*

$$
\left| \int_C K_a(x,y)s(y)\, dy - (T_p s)(x) \right| \leq \mathcal{C}(x,p;a)\, \eta^{p+1}.
$$

*In particular, for any finite set of target points $\{x_i\}$, the induced interaction matrix between $\{x_i\}$ and sources in $C$ admits a rank-$r_p$ approximation with accuracy governed by $\eta^{p+1}$.*

*Proof.* The representation is a sum of $r_p$ separable terms indexed by $|\alpha| \leq p$, so it factors through an $r_p$-dimensional moment vector. The error bound is exactly (26). $\qquad\square$

*Table 7.* Empirical clamp activation in Eq. (7) on benchmark evaluations using our best checkpoints. We report the count of pairwise distance evaluations where $-\langle u,v\rangle_M < 1+\epsilon$ (i.e., the clamping path is taken), along with the minimum observed margin $\min(-\langle u,v\rangle_M - 1)$.

| Dataset | $\epsilon$ | #pairs | Clamp count | $\min(-\langle u,v\rangle_M - 1)$ |
|---|---|---|---|---|
| Elasticity | $10^{-4}$ | $2.38 \times 10^8$ | 0 | 0.370 |
| Navier–Stokes | $10^{-6}$ | $2.10 \times 10^9$ | 0 | 0.717 |
| Darcy | $10^{-6}$ | $2.67 \times 10^8$ | 0 | 0.511 |
| Plasticity | $10^{-4}$ | $8.95 \times 10^9$ | 0 | 0.183 |
| Airfoil | $10^{-6}$ | $1.52 \times 10^{10}$ | 0 | 0.186 |
| Pipe | $10^{-6}$ | $1.09 \times 10^{10}$ | 0 | 0.567 |

*Remark* D.6 (Constant dependencies). The constant $\mathcal{C}(x,p;a)$ in Theorem D.4 depends only on $d,p$, the derivative-decay envelope $A_{p+1}(x;a)$, the norm-equivalence constant $C_{\text{eq}}$, and $\|s\|_{L^1(C)}$. No dependence on the shape of $C$ enters beyond $\eta$ and $\|s\|_{L^1(C)}$.

*Remark* D.7 (When Assumption D.2 holds). For classical PDE kernels (Laplace/Helmholtz/heat) away from their singularities, bounds of the form $\max_{|\alpha|=p+1} |\partial_y^\alpha K(x,y)| \le C_{p+1}\|x-y\|^{-(p+1)}$ hold on $\{\|x-y\| \ge \rho\}$, implying Assumption D.2 with $\rho = \text{dist}(x, \overline{\text{co}(C \cup \{c\})})$. For general data-dependent kernels $K_a$, Assumption D.2 should be treated as an explicit regularity condition. This assumption is used only to formalize the far-field motivation from classical fast solvers; our subsequent stability (Appendices E and F.2) and discretization (Appendix G) analyses do not rely on it.

# E. Scale Separation and Near–Far Routing

## E.1. Stabilized distance $d_\epsilon$ (properties used in analysis)

*Remark* E.1 ($d_\epsilon$ is not a metric). The clamping in Eq. (7) can break the triangle inequality, so $d_\epsilon$ is generally not a metric on $\mathbb{H}^m$. In this paper, we only use $d_\epsilon$ as a stabilized similarity scale for logits, together with the monotonicity and Lipschitz properties below. Empirically, the clamp is never activated on our benchmark evaluations (Tab. 7), so the behavior is governed by the true hyperbolic distance $d_\mathbb{H}$. In other words, the clamp is a training-time numerical safeguard (to avoid the singular derivative of arcosh at 1), and it is inactive at inference for our best checkpoints.

**Lemma E.2** (Monotonicity of $d_\epsilon$ and logits). *Let $g(t) = \text{arcosh}(\max\{t,1+\epsilon\})$ with $\epsilon > 0$. Then $g$ is nondecreasing, hence $d_\epsilon(u,v) = g(-\langle u,v\rangle_M)$ is nondecreasing in $-\langle u,v\rangle_M$. Consequently, for any fixed $\tau > 0$, the logit $s = -d_\epsilon/\tau$ is nonincreasing in $-\langle u,v\rangle_M$.*

*Proof.* The map $t \mapsto \max\{t,1+\epsilon\}$ is nondecreasing, and arcosh is nondecreasing on $[1,\infty)$, so their composition $g$ is nondecreasing. The remaining statements follow by composition with $-\langle u,v\rangle_M$. $\square$

**Lemma E.3** (Global Lipschitz bound induced by clamping). *Let $g(t) = \text{arcosh}(\max\{t,1+\epsilon\})$ with $\epsilon > 0$. Then $g$ is globally Lipschitz on $\mathbb{R}$ with*

$$\text{Lip}(g) \le \frac{1}{\sqrt{(1+\epsilon)^2 - 1}} = \frac{1}{\sqrt{\epsilon(2+\epsilon)}}.$$

*In particular, for any $u_1, u_2, v \in \mathbb{H}^m$,*

$$\left|d_\epsilon(u_1,v) - d_\epsilon(u_2,v)\right| \le \frac{1}{\sqrt{\epsilon(2+\epsilon)}}\left|\langle u_1 - u_2, v\rangle_M\right| \le \frac{\|v\|_2}{\sqrt{\epsilon(2+\epsilon)}}\|u_1 - u_2\|_2,$$

*and similarly in the second argument. On bounded subsets of $\mathbb{H}^m$, this yields a uniform Euclidean Lipschitz constant and is the only place where the factor $1/\sqrt{\epsilon(2+\epsilon)}$ enters; Lemma G.1 translates this into Lipschitzness of the kernel in physical coordinates.*

*Proof.* For $t > 1$, $\frac{d}{dt}\text{arcosh}(t) = \frac{1}{\sqrt{t^2-1}}$. Since $\max\{t,1+\epsilon\} \ge 1+\epsilon$, the derivative is bounded by $1/\sqrt{(1+\epsilon)^2-1}$ wherever it exists, and $g$ is constant on $(-\infty, 1+\epsilon)$. Hence $g$ is globally Lipschitz with the stated constant. The final inequalities use $|\langle a,b\rangle_M| \le \|a\|_2\|b\|_2$. $\square$

### E.2. Proof of Proposition 4.1

On the Lorentz hyperboloid, write $u = (u_0, \bar{u})$ and $v = (v_0, \bar{v})$. For $u \in \mathbb{H}^m$, we have $u_0 = \cosh r_u$ and $\|\bar{u}\| = \sinh r_u$, where $r_u = d_{\mathbb{H}}(o, u)$, and similarly for $v$. Assume $r_u, r_v > 0$ and let $\theta = \theta(u, v)$ satisfy $\cos \theta = \langle \bar{u}, \bar{v} \rangle / (\|\bar{u}\| \|\bar{v}\|)$.

The hyperbolic law of cosines gives

$$\cosh d_{\mathbb{H}}(u, v) = \cosh r_u \cosh r_v - \sinh r_u \sinh r_v \cos \theta = \cosh(r_u - r_v) + 2 \sinh r_u \sinh r_v \sin^2 \frac{\theta}{2}.$$

Assume $d_{\mathbb{H}}(u, v) \leq R$. Then

$$2 \sinh r_u \sinh r_v \sin^2 \frac{\theta}{2} \leq \cosh R - \cosh(r_u - r_v) \leq \cosh R.$$

For $r \geq 1$, $\sinh r \geq \frac{1}{2}(e^r - 1) \geq \frac{1}{2}e^{r-1}$, hence for $r_u, r_v \geq 1$,

$$\sinh r_u \sinh r_v \geq \tfrac{1}{4} e^{r_u + r_v - 2}.$$

Therefore,

$$\sin^2 \frac{\theta}{2} \leq \frac{\cosh R}{2 \sinh r_u \sinh r_v} \leq 2e^2 \cosh R \cdot e^{-(r_u + r_v)}.$$

Using $\sin t \geq 2t/\pi$ for $t \in [0, \pi/2]$ (concavity of sin) yields $\sin(\theta/2) \geq \theta/\pi$ for $\theta \in [0, \pi]$, hence

$$\theta \leq \pi \sqrt{2e^2 \cosh R} \, \exp\left(-\frac{r_u + r_v}{2}\right) =: C_R \exp\left(-\frac{r_u + r_v}{2}\right),$$

which proves the claim for $r_u, r_v \geq 1$. If instead $\min\{r_u, r_v\} < 1$, then $|r_u - r_v| \leq d_{\mathbb{H}}(u, v) \leq R$ implies $r_u + r_v \leq R + 2$, so $\exp(-(r_u + r_v)/2) \geq \exp(-(R + 2)/2)$. Since $\theta \in [0, \pi]$ and $C_R \exp(-(R + 2)/2) \geq \pi$ (using $\cosh R \geq e^R/2$), the same bound holds for all radii.

### E.3. From weight thresholds to angular cones

**Corollary E.4** (Thresholded attention concentrates in a hyperbolic ball and an angular cone)**.** *Fix a query point $u \in \mathbb{H}^m$, a measurable key map $y \mapsto v(y) \in \mathbb{H}^m$, and $\tau > 0$. Let $d_{\min} := \inf_{y \in D} d_\epsilon(u, v(y))$ and define the (relative) $\delta$-threshold set for $\delta \in (0, 1)$,*

$$\mathcal{N}_\delta(u) := \left\{ y \in D : \exp\left(-d_\epsilon(u, v(y))/\tau\right) \geq \delta \exp\left(-d_{\min}/\tau\right) \right\}.$$

*Equivalently, if $\kappa_u(y) \propto \exp(-d_\epsilon(u, v(y))/\tau)$ denotes the normalized attention weight for fixed $u$, then $\mathcal{N}_\delta(u) = \{y : \kappa_u(y) \geq \delta \sup_{y'} \kappa_u(y')\}$. Then $\mathcal{N}_\delta(u) \subseteq \{y \in D : d_\epsilon(u, v(y)) \leq d_{\min} + \tau \log(1/\delta)\}$. Moreover, since $d_{\mathbb{H}}(u, v) \leq d_\epsilon(u, v)$, one also has*

$$\mathcal{N}_\delta(u) \subseteq \left\{ y \in D : d_{\mathbb{H}}(u, v(y)) \leq d_{\min} + \tau \log(1/\delta) \right\},$$

*and Proposition 4.1 implies that points in $\mathcal{N}_\delta(u)$ lie in an angular cone whose aperture decays as $\exp(-(r_u + r_{v(y)})/2)$, up to a constant depending on $d_{\min} + \tau \log(1/\delta)$.*

*Proof.* The first inclusion is immediate: $\exp(-d_\epsilon/\tau) \geq \delta \exp(-d_{\min}/\tau)$ is equivalent to $d_\epsilon \leq d_{\min} + \tau \log(1/\delta)$. The second inclusion uses $d_{\mathbb{H}} \leq d_\epsilon$, which follows from $\max\{1 + \epsilon, -\langle u, v \rangle_M\} \geq -\langle u, v \rangle_M$ and monotonicity of arcosh. Applying Proposition 4.1 with $R = d_{\min} + \tau \log(1/\delta)$ yields the cone bound. $\square$

## F. Operator Stability via Schur's Test

### F.1. Setup for the Hyperbolic Attention Kernel

Let $D \subset \mathbb{R}^d$ be a bounded domain with measure $\mu$. We distinguish two operator levels.

**Scalar Kernel Operator.** The scalar attention operator $\mathcal{K}_\kappa : L^2(D) \to L^2(D)$ is defined by

$$(\mathcal{K}_\kappa v)(x) = \int_D \kappa(x, y; a) \, v(y) \, d\mu(y), \tag{27}$$

where $\kappa(x, y; a)$ is the normalized hyperbolic attention kernel, a scalar.

**Full Operator with Projections.** With learnable projections $W_O, W_V$, the full operator is

$$\mathcal{K} := W_O \circ \mathcal{K}_\kappa \circ W_V.$$

All Schur bounds are first derived for $\mathcal{K}_\kappa$, then $\|\mathcal{K}\| \le \|W_O\|_{\mathrm{op}} \|W_V\|_{\mathrm{op}} \|\mathcal{K}_\kappa\|$.

**Vector-Valued and Multi-Head Extension.** The analysis in the main text focuses on a scalar kernel operator $\mathcal{K}_\kappa$ and accounts for value/output projections by operator norms. The same reasoning extends to standard multi-head attention by viewing heads as a direct sum of bounded operators.

**Lemma F.1** (Multi-head operator norm bound). *Let $H \ge 1$. For each head $h \in \{1, \ldots, H\}$, let $\mathcal{K}_\kappa^{(h)} : L^2(D) \to L^2(D)$ be a scalar-kernel operator with $\|\mathcal{K}_\kappa^{(h)}\|_{L^2 \to L^2} \le C_h$, and denote by the same symbol its componentwise extension to vector-valued functions. Let $W_V^{(h)}$ be the value projection for head $h$ and let $W_O$ be the (pointwise) output projection applied after concatenation. Define the multi-head layer on vector-valued functions by*

$$\mathcal{K}_{\mathrm{MH}}(v) := W_O \Big[ \mathcal{K}_\kappa^{(1)}(W_V^{(1)} v), \ldots, \mathcal{K}_\kappa^{(H)}(W_V^{(H)} v) \Big],$$

*where $[\cdot, \ldots, \cdot]$ denotes concatenation over heads. Then*

$$\|\mathcal{K}_{\mathrm{MH}}\|_{L^2(D;\mathbb{R}^{d_v}) \to L^2(D;\mathbb{R}^{d_v})} \le \|W_O\|_{\mathrm{op}} \Big( \sum_{h=1}^H \big( C_h \|W_V^{(h)}\|_{\mathrm{op}} \big)^2 \Big)^{1/2} \le \|W_O\|_{\mathrm{op}} \sqrt{H} \max_h \big( C_h \|W_V^{(h)}\|_{\mathrm{op}} \big).$$

*Proof.* By definition and Cauchy–Schwarz in the direct-sum Hilbert space,

$$\sum_{h=1}^H \|\mathcal{K}_\kappa^{(h)}(W_V^{(h)} v)\|_{L^2}^2 \le \sum_{h=1}^H C_h^2 \|W_V^{(h)}\|_{\mathrm{op}}^2 \|v\|_{L^2}^2.$$

Concatenation preserves the squared norm by summation over heads, and applying $W_O$ pointwise contributes a factor $\|W_O\|_{\mathrm{op}}$. $\square$

### F.2. Boundedness via Schur's Test

We show boundedness of $\mathcal{K}_\kappa$ for the continuum normalized kernel.

#### F.2.1. CONTINUUM NORMALIZED KERNEL

For the continuum normalized kernel (Eq. 9):

$$\kappa(x, y) = \frac{\exp(-d_\epsilon(\Pi(q(x)), \Pi(k(y)))/\tau)}{Z(x)}, \quad Z(x) := \int_D \exp(-d_\epsilon(\Pi(q(x)), \Pi(k(y')))/\tau) \, d\mu(y').$$

**Proposition F.2** (Continuum Boundedness). *The continuum normalized kernel satisfies:*

1. *$\kappa(x, y) \ge 0$ and $\int_D \kappa(x, y) \, d\mu(y) = 1$ for all $x$ (normalization).*

2. *If $Z(x) \ge c_0 > 0$ uniformly, then $\int_D \kappa(x, y) \, d\mu(x) \le \mu(D)/c_0$.*

*By Schur's test (Kress, 2014) with $C_1 = 1$ and $C_2 = \mu(D)/c_0$:*

$$\|\mathcal{K}_\kappa\|_{L^2 \to L^2} \le \sqrt{\mu(D)/c_0}, \qquad \|\mathcal{K}\|_{L^2 \to L^2} \le \|W_O\|_{\mathrm{op}} \|W_V\|_{\mathrm{op}} \sqrt{\mu(D)/c_0}.$$

*Proof.* Property (1) is immediate from the definition. For property (2), note that the unnormalized kernel $\tilde{\kappa}(x, y) := \exp(-d_\epsilon(\Pi(q(x)), \Pi(k(y)))/\tau)$ satisfies $\tilde{\kappa} \le 1$. Thus $\int_D \kappa(x, y) \, d\mu(x) = \int_D \tilde{\kappa}(x, y)/Z(x) \, d\mu(x) \le \mu(D)/c_0$. $\square$

**Proposition F.3** (A sufficient lower bound on the normalizer $Z(x)$). *Assume there exist $R < \infty$ and $\tau_{\min} > 0$ such that for all $x, y \in D$, $\Pi(q(x; a)), \Pi(k(y; a)) \in B_{\mathbb{H}}(o, R)$ and $\tau \geq \tau_{\min}$. Then*

$$Z(x) \geq \mu(D) \exp\left(-\frac{\max\{2R, \mathrm{arcosh}(1 + \epsilon)\}}{\tau_{\min}}\right) =: c_0 \quad \text{for all } x \in D.$$

*In particular, it suffices to bound Euclidean feature norms: if $\|q(x; a)\| \leq B_q$ and $\|k(y; a)\| \leq B_k$ for all $x, y$, then one can take $R = \max\{\mathrm{asinh}(B_q), \mathrm{asinh}(B_k)\}$.*

*Proof.* If $\Pi(q(x; a)), \Pi(k(y; a)) \in B_{\mathbb{H}}(o, R)$, then the hyperbolic triangle inequality gives $d_{\mathbb{H}}(\Pi(q(x; a)), \Pi(k(y; a))) \leq 2R$. By Proposition A.1 and monotonicity of $\mathrm{arcosh}$,

$$d_\epsilon\big(\Pi(q(x; a)), \Pi(k(y; a))\big) = \mathrm{arcosh}\big(\max\{1 + \epsilon, -\langle\Pi(q(x; a)), \Pi(k(y; a))\rangle_M\}\big) \leq \max\{2R, \mathrm{arcosh}(1 + \epsilon)\}.$$

Hence $\exp(-d_\epsilon/\tau) \geq \exp(-\max\{2R, \mathrm{arcosh}(1 + \epsilon)\}/\tau_{\min})$ for all $y$, and integrating over $D$ yields the claimed bound on $Z(x)$. Finally, $r(z) = d_{\mathbb{H}}(o, \Pi(z)) = \mathrm{asinh}(\|z\|)$ implies $\Pi(z) \in B_{\mathbb{H}}(o, \mathrm{asinh}(\|z\|))$, so a uniform Euclidean norm bound implies a uniform radius bound. $\quad\square$

*Remark* F.4 (How to enforce the assumptions in practice). The radius bound can be guaranteed by explicitly bounding $\|q\|$ and $\|k\|$ via bounded activations such as $\tanh$, explicit norm clipping, or layer normalization followed by a bounded scale. If $\tau$ is learnable, a convenient way to ensure $\tau \geq \tau_{\min}$ is to parameterize $\tau = \tau_{\min} + \mathrm{softplus}(\hat{\tau})$.

### F.3. Bi-Lipschitz Residuals and Stability

For the main text, we analyze a *fixed-kernel* setting for the operator family $\{\mathcal{K}_a\}$ introduced in Section 4.2. We treat the query and key maps as fixed for a given input field $a$. Under this conditioning, the kernel $\kappa_a(x, y)$ is fixed and the map $v \mapsto \mathcal{K}_a v$ is linear. In particular, it is $\|\mathcal{K}_a\|$-Lipschitz in any norm, and in $L^2(D)$ we have $\mathrm{Lip}(\mathcal{K}_a) = \|\mathcal{K}_a\|_{L^2 \to L^2}$. Combining this with the boundedness results from Appendix F.2 yields a sufficient uniform Lipschitz bound to invoke Proposition F.6; the resulting condition is sufficient and can be conservative.

*Remark* F.5 (Implementation (stabilization and temperature lower bound)). In our implementations, the per-head temperature is clamped to $\tau \in [0.1, 3.0]$, and the $\mathrm{arcosh}$ argument is clamped to $\geq 1 + \epsilon$ with $\epsilon \in \{10^{-6}, 10^{-4}\}$ to avoid the singular derivative at $1$. We also apply LayerNorm to the hidden state before Q/K projections. Tab. 7 confirms that the $\mathrm{arcosh}$ clamp is never activated on benchmark evaluations of our best checkpoints.

**Proposition F.6** (Bi-Lipschitz stability of residual maps). *Let $(X, \|\cdot\|)$ be a normed vector space and let $\mathcal{F} : X \to X$ be $C_{\mathcal{F}}$-Lipschitz: $\|\mathcal{F}(u) - \mathcal{F}(v)\| \leq C_{\mathcal{F}}\|u - v\|$. If $|\alpha|C_{\mathcal{F}} < 1$ and $T := I + \alpha\mathcal{F}$, then for all $u, v \in X$,*

$$(1 - |\alpha|C_{\mathcal{F}})\|u - v\| \leq \|T(u) - T(v)\| \leq (1 + |\alpha|C_{\mathcal{F}})\|u - v\|.$$

*In particular, $T$ is injective and has a Lipschitz inverse on its range.*

*Proof.* By the triangle inequality:

$$\begin{aligned}
\|T(u) - T(v)\| &= \|(u - v) + \alpha(\mathcal{F}(u) - \mathcal{F}(v))\| \\
&\leq \|u - v\| + |\alpha|\|\mathcal{F}(u) - \mathcal{F}(v)\| \leq (1 + |\alpha|C_{\mathcal{F}})\|u - v\|, \\
\|T(u) - T(v)\| &\geq \|u - v\| - |\alpha|\|\mathcal{F}(u) - \mathcal{F}(v)\| \geq (1 - |\alpha|C_{\mathcal{F}})\|u - v\|.
\end{aligned}$$

This gives the claimed bi-Lipschitz bounds. $\quad\square$

**Application to Fixed-Kernel Layers.** For each $a$, apply Proposition F.6 with $X = L^2(D)$ and $\mathcal{F} = \mathcal{K}_a$, so that $C_{\mathcal{F}} = \|\mathcal{K}_a\|_{L^2 \to L^2}$. Since $\mathcal{K}_a$ is linear, this condition is sharp up to constants and matches standard sufficient conditions for invertible residual blocks (Behrmann et al., 2019).

## G. Discretization Consistency

We analyze discretization consistency in the *fixed-kernel* setting: for each input $a$, the feature maps $q(\cdot; a)$ and $k(\cdot; a)$ (hence $\kappa(\cdot, \cdot; a)$) are treated as fixed, and the discretization error is measured relative to the induced continuum operator. We first prove that the discrete scalar-kernel operator $\mathcal{K}_{\kappa, N}$ converges to the continuum kernel operator $\mathcal{K}_\kappa$ as the mesh refines. The full layer discretization is then $\mathcal{K}_N := W_O \circ \mathcal{K}_{\kappa, N} \circ W_V$, which converges to $\mathcal{K} = W_O \circ \mathcal{K}_\kappa \circ W_V$. This is crucial for neural operators: it ensures the learned model does not depend on a particular discretization.

## G.1. Setup and Notation

Let $D \subset \mathbb{R}^d$ be a bounded domain with Lebesgue measure $\mu$. Consider a sequence of discretizations $\{x_j\}_{j=1}^N \subset D$ with associated quadrature weights $\{w_j\}_{j=1}^N$. We assume the quadrature rule satisfies the Lipschitz accuracy condition (Assumption G.3), i.e., for any Lipschitz integrand $g$:

$$\left| \int_D g(y) \, d\mu(y) - \sum_{j=1}^N w_j g(x_j) \right| \leq C_q \, h \operatorname{Lip}(g), \tag{28}$$

where $h := \max_j \sup_{x \in V_j} \|x - x_j\|$ is the mesh size (maximum cell diameter), and $V_j$ is the Voronoi cell of $x_j$. We equip vectors $u \in (\mathbb{R}^{d_v})^N$ with the weighted norm $\|u\|_{\ell_w^2}^2 := \sum_{j=1}^N w_j \|u_j\|_2^2$.

The continuum operator is:

$$(\mathcal{K}_\kappa v)(x) = \int_D \kappa(x, y) \, v(y) \, d\mu(y),$$

and we define the sampling operator $P_N$ by $(P_N v)_j := v(x_j)$. The Nyström (quadrature) discretization of the continuum kernel is the map $\mathcal{K}_{\kappa,N}^{\text{cont}} : (\mathbb{R}^{d_v})^N \to (\mathbb{R}^{d_v})^N$ given by

$$(\mathcal{K}_{\kappa,N}^{\text{cont}} u)_i := \sum_{j=1}^N w_j \, \kappa(x_i, x_j) \, u_j, \qquad u \in (\mathbb{R}^{d_v})^N.$$

The implemented discrete scalar operator $\mathcal{K}_{\kappa,N}$ uses the discretely normalized kernel $\kappa_N$ from Eq. 29 and applies it with quadrature weights,

$$(\mathcal{K}_{\kappa,N} u)_i := \sum_{j=1}^N w_j \, \kappa_N(x_i, x_j) \, u_j = \sum_{j=1}^N \alpha_{ij} \, u_j, \quad \alpha_{ij} = w_j \, \kappa_N(x_i, x_j).$$

The full layer discretization is $\mathcal{K}_N := W_O \circ \mathcal{K}_{\kappa,N} \circ W_V$.

We prove discretization consistency.

## G.2. Kernel Regularity

We first establish regularity of the hyperbolic attention kernel.

**Lemma G.1** (Lipschitzness of the fixed-kernel hyperbolic attention kernel)**.** *Let $\kappa(x,y) = \tilde{\kappa}(x,y)/Z(x)$ with $\tilde{\kappa}(x,y) = \exp(-d_\epsilon(\Pi(q(x)), \Pi(k(y)))/\tau)$ and $d_\epsilon(p,q) = \operatorname{arcosh}(\max\{-\langle p, q \rangle_M, 1 + \epsilon\})$. Assume that $q$ and $k$ are Lipschitz with constants $L_q$ and $L_k$, that $\Pi(q(D))$ and $\Pi(k(D))$ lie in a hyperbolic ball $B_{\mathbb{H}}(o, R)$, and that $Z(x) \geq c_0 > 0$ and $\tau \geq \tau_{\min} > 0$ hold uniformly. Let $L_{qk} := \max\{L_q, L_k\}$. Then for each fixed $x$, the map $y \mapsto \tilde{\kappa}(x,y)$ is globally Lipschitz on $D$ with*

$$\operatorname{Lip}_y(\tilde{\kappa}(x, \cdot)) \leq C_R \cdot \frac{L_{qk}}{\tau_{\min}} \cdot \frac{1}{\sqrt{\epsilon(2 + \epsilon)}}.$$

*Consequently, $y \mapsto \kappa(x,y)$ is globally Lipschitz on $D$ with*

$$\operatorname{Lip}_y(\kappa(x, \cdot)) \leq C_R \cdot \frac{L_{qk}}{\tau_{\min} c_0} \cdot \frac{1}{\sqrt{\epsilon(2 + \epsilon)}},$$

*and for $\epsilon > 0$ the kernel is piecewise $C^\infty$ away from the clamping boundary $\{-\langle p, q \rangle_M = 1 + \epsilon\}$ and globally Lipschitz.*

*Proof.* **Proof outline.** Define $g(t) := \operatorname{arcosh}(\max\{t, 1 + \epsilon\})$ for $\epsilon > 0$. Since $\frac{d}{dt} \operatorname{arcosh}(t) = \frac{1}{\sqrt{t^2 - 1}}$ for $t > 1$, one has

$$|g'(t)| \leq \frac{1}{\sqrt{(1 + \epsilon)^2 - 1}} = \frac{1}{\sqrt{\epsilon(2 + \epsilon)}} =: L_g,$$

and the clamping makes $g$ globally $L_g$-Lipschitz on $\mathbb{R}$.

Next, the Lorentz lift $\Pi(z) = (\sqrt{1 + \|z\|^2}, z)$ satisfies that the scalar map $z \mapsto \sqrt{1 + \|z\|^2}$ has gradient norm $\leq 1$, hence it is 1-Lipschitz. Therefore, for all $z_1, z_2 \in \mathbb{R}^m$,

$$\|\Pi(z_1) - \Pi(z_2)\|_2^2 = \left(\sqrt{1 + \|z_1\|^2} - \sqrt{1 + \|z_2\|^2}\right)^2 + \|z_1 - z_2\|_2^2 \leq 2\|z_1 - z_2\|_2^2,$$

so $\Pi$ is $L_\Pi := \sqrt{2}$-Lipschitz.

**(i) Stabilized distance is Lipschitz on bounded sets.** Recall $d_\epsilon(p, q) = g(-\langle p, q \rangle_M)$, and $|\langle a, b \rangle_M| \leq \|a\|_2 \|b\|_2$. Fix $x \in D$ and write $p_x := \Pi(q(x; a))$. If $p_x$ and $m_y := \Pi(k(y; a))$ remain in a hyperbolic ball $B_{\mathbb{H}}(o, R)$, then their Euclidean norms are uniformly bounded; for instance

$$\|w\|_2 \leq \cosh R + \sinh R \leq e^R =: B_R \quad \text{for all } w \in B_{\mathbb{H}}(o, R).$$

For any $y_1, y_2 \in D$,

$$\left|d_\epsilon(p_x, m_{y_1}) - d_\epsilon(p_x, m_{y_2})\right| \leq L_g \left|\langle p_x, m_{y_2} - m_{y_1} \rangle_M\right| \leq L_g \|p_x\|_2 \|m_{y_2} - m_{y_1}\|_2 \leq L_g B_R \|m_{y_2} - m_{y_1}\|_2.$$

Using the Lipschitzness of $k(\cdot; a)$ and $\Pi$ gives

$$\|m_{y_2} - m_{y_1}\|_2 \leq L_\Pi L_k \|y_2 - y_1\|_2,$$

and similarly with $L_q$ when varying $x$. Thus, for each fixed $x$, the map $y \mapsto d_\epsilon(\Pi(q(x; a)), \Pi(k(y; a)))$ is Lipschitz with constant $\leq L_g B_R L_\Pi L_{qk}$.

**(ii) Kernel Lipschitzness and normalization.** For $\tau \geq \tau_{\min} > 0$, the map $t \mapsto e^{-t/\tau}$ has derivative bounded by $1/\tau_{\min}$ on $[0, \infty)$, and $\tilde{\kappa}(x, y) \in (0, 1]$. Hence

$$\text{Lip}_y(\tilde{\kappa}(x, \cdot)) \leq \frac{1}{\tau_{\min}} \text{Lip}_y\left(d_\epsilon(\Pi(q(x; a)), \Pi(k(\cdot; a)))\right) \leq \frac{L_g B_R L_\Pi L_{qk}}{\tau_{\min}}.$$

Finally, $\kappa(x, y) = \tilde{\kappa}(x, y)/Z(x)$ with $Z(x) \geq c_0 > 0$ by assumption, and for fixed $x$ the denominator is constant w.r.t. $y$. Therefore

$$\text{Lip}_y(\kappa(x, \cdot)) \leq \frac{1}{c_0} \text{Lip}_y(\tilde{\kappa}(x, \cdot)) \leq \frac{L_g B_R L_\Pi L_{qk}}{\tau_{\min} c_0}.$$

This yields the stated bound (absorbing $B_R L_\Pi$ into the radius-dependent constant $C_R$) and also shows Lipschitzness for $\epsilon > 0$ under the bounded-radius assumption. $\qquad\square$

**Lemma G.2** (Feature Map Lipschitz Bound). *Assume, for a fixed input field $a$ and layer state, that the feature maps satisfy $q(\cdot; a), k(\cdot; a) \in W^{1,\infty}(D)$ with constants $L_q$ and $L_k$. This condition holds, for example, when the input fields and positional features entering the Q/K maps are Lipschitz and the finite-dimensional projection networks have bounded Lipschitz constants. With weight normalization or spectral normalization, the network contribution to these constants can be controlled.*

### G.3. Quadrature Error Analysis

For normalized kernels, the discrete operator uses softmax normalization:

$$\kappa_N(x_i, x_j) := \frac{\exp(-d_\epsilon(\Pi(q(x_i)), \Pi(k(x_j)))/\tau)}{\sum_{\ell=1}^N w_\ell \exp(-d_\epsilon(\Pi(q(x_i)), \Pi(k(x_\ell)))/\tau)}. \tag{29}$$

This differs from the continuum kernel which uses integral normalization. In our experiments, we take uniform quadrature weights $w_j = \mu(D)/N$ (equivalently $w_j = 1$ up to a global scaling), so the implemented coefficients $\alpha_{ij} = w_j \kappa_N(x_i, x_j)$ reduce to standard softmax attention weights. We keep the weighted form for general nonuniform discretizations.

**Assumption G.3** (Lipschitz quadrature accuracy). Assume $w_j \geq 0$ and $\sum_{j=1}^N w_j = \mu(D)$. There exists $C_q > 0$ such that for any Lipschitz function $g$ on $D$,

$$\left|\int_D g(y)\, d\mu(y) - \sum_{j=1}^N w_j g(x_j)\right| \leq C_q\, h\, \text{Lip}(g).$$

*Remark* G.4 (Relation to $W_1$). Assumption G.3 holds, for instance, when the normalized measures $\bar{\mu} := \mu/\mu(D)$ and $\bar{\mu}_N := \frac{1}{\mu(D)} \sum_{j=1}^N w_j \delta_{x_j}$ satisfy $W_1(\bar{\mu}, \bar{\mu}_N) \leq Ch$ for some constant $C$ independent of $N$, by the Kantorovich–Rubinstein duality (Villani, 2008). This includes quasi-uniform meshes, and can be interpreted as a mesh-quality condition in point-cloud settings with uniform weights.

**Theorem G.5** (Discretization consistency under Lipschitz kernels). *Assume Lemma G.1 and Assumption G.3. If $v$ is bounded and Lipschitz on $D$, then*

$$\|\mathcal{K}_{\kappa,N} P_N v - P_N \mathcal{K}_\kappa v\|_{\ell_w^2} \leq C\, h\, \|v\|_{W^{1,\infty}(D)},$$

*where $C$ depends on $C_q$, the uniform bound on $\kappa$, and $\sup_x \mathrm{Lip}_y(\kappa(x,\cdot))$. All norms are understood in the vector-valued sense using the Euclidean norm in the feature space.*

Consequently, the full layer discretization satisfies

$$\|\mathcal{K}_N P_N v - P_N \mathcal{K} v\|_{\ell_w^2} \leq \|W_O\|_{\mathrm{op}} \|W_V\|_{\mathrm{op}}\, C\, h\, \|v\|_{W^{1,\infty}(D)}.$$

*Proof.* We decompose the error at sample points into two terms:

$$\mathcal{K}_{\kappa,N} P_N v - P_N \mathcal{K}_\kappa v = \underbrace{\left(\mathcal{K}_{\kappa,N} P_N v - \mathcal{K}_{\kappa,N}^{\mathrm{cont}} P_N v\right)}_{\text{(I) Normalization gap}} + \underbrace{\left(\mathcal{K}_{\kappa,N}^{\mathrm{cont}} P_N v - P_N \mathcal{K}_\kappa v\right)}_{\text{(II) Quadrature}}.$$

**(I) Normalization gap.** Let $\tilde{\kappa}_i(y) := \exp(-d_\epsilon(\Pi(q(x_i)), \Pi(k(y)))/\tau)$, $Z(x_i) := \int_D \tilde{\kappa}_i(y)\, d\mu(y)$, and $Z_N(x_i) := \sum_{j=1}^N w_j \tilde{\kappa}_i(x_j)$. Then $\kappa(x_i, x_j) = \tilde{\kappa}_i(x_j)/Z(x_i)$ and $\kappa_N(x_i, x_j) = \tilde{\kappa}_i(x_j)/Z_N(x_i)$, hence

$$\left|(\mathcal{K}_{\kappa,N} P_N v)_i - (\mathcal{K}_{\kappa,N}^{\mathrm{cont}} P_N v)_i\right| \leq \|v\|_\infty \left|\frac{1}{Z_N(x_i)} - \frac{1}{Z(x_i)}\right| \sum_{j=1}^N w_j \tilde{\kappa}_i(x_j)$$

$$= \|v\|_\infty \frac{|Z(x_i) - Z_N(x_i)|}{Z(x_i)} \leq \frac{\|v\|_\infty}{c_0} |Z(x_i) - Z_N(x_i)|.$$

Applying Assumption G.3 to $y \mapsto \tilde{\kappa}_i(y)$ gives $|Z(x_i) - Z_N(x_i)| \leq C_q h\, \mathrm{Lip}(\tilde{\kappa}_i)$. By Lemma G.1, $\mathrm{Lip}(\tilde{\kappa}_i)$ is uniformly bounded (and does not incur an additional $1/c_0$ factor). Since $\sum_{j=1}^N w_j = \mu(D) > 0$ and $\tilde{\kappa}_i(x_j) > 0$, we have $Z_N(x_i) > 0$ and $\kappa_N$ is well-defined. Thus (I) is bounded by $Ch\|v\|_\infty$ for a constant $C$ independent of $N$.

**(II) Quadrature.** Fix $i$ and define $g_i(y) := \kappa(x_i, y)\, v(y)$. By Assumption G.3,

$$\left|(\mathcal{K}_{\kappa,N}^{\mathrm{cont}} P_N v)_i - (P_N \mathcal{K}_\kappa v)_i\right| = \left|\sum_{j=1}^N w_j g_i(x_j) - \int_D g_i(y)\, d\mu(y)\right| \leq C_q h\, \mathrm{Lip}(g_i).$$

Moreover, $\mathrm{Lip}(g_i) \leq \|\kappa(x_i, \cdot)\|_\infty \mathrm{Lip}(v) + \|v\|_\infty \mathrm{Lip}_y(\kappa(x_i, \cdot))$. Using Lemma G.1 to bound $\sup_i \mathrm{Lip}_y(\kappa(x_i, \cdot))$ and the uniform bound $\|\kappa\|_\infty \leq 1/c_0$, summing over quadrature points with weights $w_i$ and using $\sum_i w_i = \mu(D)$, yields a bound $\leq C\sqrt{\mu(D)}\, h\|v\|_{W^{1,\infty}}$ in $\ell_w^2$ for a constant $C$ independent of $N$.

Combining (I) and (II) and using $\|v\|_\infty \leq \|v\|_{W^{1,\infty}(D)}$ yields the stated estimate. □

*Remark* G.6 (Discrete normalization stability). Since $\sum_{j=1}^N w_j = \mu(D) > 0$ and $\tilde{\kappa}_i(x_j) > 0$, one always has $Z_N(x_i) > 0$. Moreover, if $h$ is small enough such that $C_q h \sup_i \mathrm{Lip}(\tilde{\kappa}_i) \leq c_0/2$, then $Z_N(x_i) \geq c_0/2$ for all $i$, which improves numerical stability of the discrete normalization.

**Corollary G.7** (Strong convergence / discretization invariance). *Under the assumptions of Theorem G.5, for any sequence of discretizations with mesh size $h \to 0$ one has, for each $v \in W^{1,\infty}(D; \mathbb{R}^{d_v})$,*

$$\|\mathcal{K}_{\kappa,N} P_N v - P_N \mathcal{K}_\kappa v\|_{\ell_w^2} \to 0, \qquad \|\mathcal{K}_N P_N v - P_N \mathcal{K} v\|_{\ell_w^2} \to 0.$$

*Equivalently, the difference operators converge strongly and have $O(h)$ error as maps $W^{1,\infty}(D; \mathbb{R}^{d_v}) \to \ell_w^2$, matching the continuum-referenced discretization consistency commonly required in neural operators (Kovachki et al., 2023).*

*Proof.* Immediate from Theorem G.5. □

G.3.1. PRACTICAL IMPLICATIONS

*Remark* G.8 (Convergence Rate). For typical meshes with $N \sim h^{-d}$ points, Theorem G.5 implies:

$$\|\mathcal{K}_{\kappa,N} P_N v - P_N \mathcal{K}_\kappa v\|_{\ell_w^2} = O(N^{-1/d}).$$

*Remark* G.9 (Stability of Hyperbolic Distance). The arcosh function has a singularity at $z = 1$ (when $d_{\mathbb{H}} = 0$). In our implementation, we clamp the argument to $z \geq 1 + \epsilon$ with a small $\epsilon$ (typically $10^{-6}$ or $10^{-4}$ depending on the variant), which:

1. Ensures the derivative $1/\sqrt{z^2 - 1}$ is bounded by $O(1/\sqrt{\epsilon})$.

2. Introduces a bias of order $O(\sqrt{\epsilon})$ in the distance, which is negligible.

This numerical stabilization does not affect the convergence rate.

*Remark* G.10 (Temperature Dependence). The Lipschitz constant in Lemma G.1 depends on $1/\tau$. Lower temperatures $\tau \to 0$ sharpen the attention but increase the kernel's Lipschitz constant, potentially degrading the discretization error constant. In our implementations, we clamp the learned temperatures to $\tau \in [0.1, 3.0]$ for numerical stability.

## H. Toy Multiscale Tree-Kernel Fitting: Quantitative Scaling

We report quantitative errors for the toy tree-kernel fitting experiment summarized in the main text.

**Setup.** This toy is a controlled *graph-domain* operator-learning problem (binary tree), not a physical PDE benchmark. We use complete binary trees of depth $L \in \{5, 6, 7, 8, 9, 10\}$ with $N = 2^L$ leaves. Let $d_{\text{tree}}(i, j)$ denote the shortest-path distance between leaves $i$ and $j$. Equivalently, if $\ell(i, j) \in \{0, \ldots, L\}$ is the depth of the lowest common ancestor (LCA) of leaves $i, j$, then $d_{\text{tree}}(i, j) = 2(L - \ell(i, j))$, which induces $L+1$ interaction scales by LCA depth. We define the ground-truth row-stochastic kernel

$$K_{\text{true}}(i, j) \propto \exp\big(-\gamma_{\text{eff}} \, d_{\text{tree}}(i, j)\big), \qquad \gamma_{\text{eff}} = \gamma/L,$$

with $\gamma = 1$. We fit $K_{\text{pred}}$ from learned leaf embeddings using either dot-product or hyperbolic-distance attention (both with embedding dimension $d=4$ and learnable temperature $\tau$). For hyperbolic attention, we use a Poincaré-ball distance with learnable curvature $c$ (bounded by a sigmoid parameterization for stability). We optimize a mixed objective combining row-wise kernel divergence $\text{KL}(K_{\text{true}} \| K_{\text{pred}})$ and an operator regression loss on random sources $s \sim \mathcal{N}(0, I)$, using AdamW (lr $10^{-2}$, weight decay $10^{-4}$) for 2000 steps with batch size 64, and keep the optimization budget fixed across $N$.

**Metrics.** We report the relative kernel error $\|K_{\text{pred}} - K_{\text{true}}\|_F / \|K_{\text{true}}\|_F$, the relative operator error $\mathbb{E}_s\big[\|K_{\text{pred}} s - K_{\text{true}} s\|_2 / \|K_{\text{true}} s\|_2\big]$ with $s \sim \mathcal{N}(0, I)$, and the row-wise KL divergence $\text{KL}(K_{\text{true}} \| K_{\text{pred}})$.

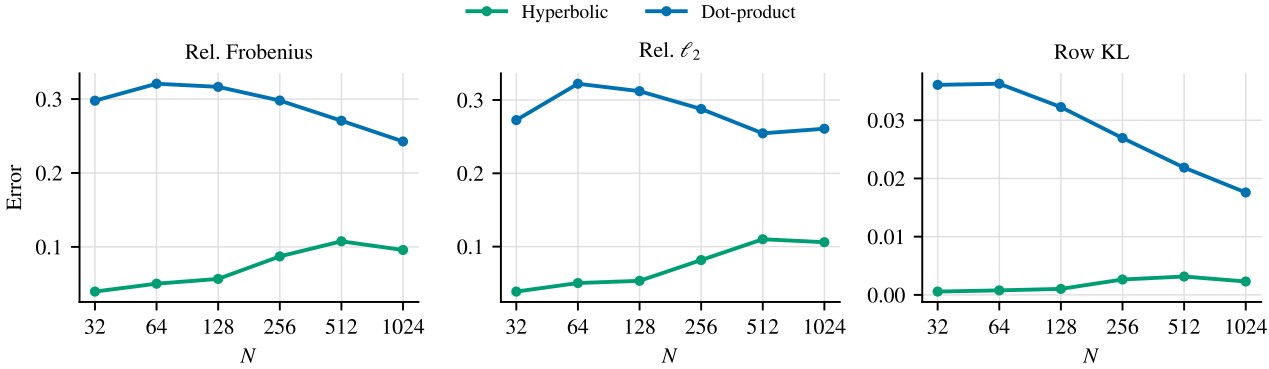

*Figure 10.* Toy tree-kernel fitting: quantitative scaling with leaf count $N$.

*Table 8.* Numerical values for Fig. 10.

| $N$ | REL. FROBENIUS | | REL. $\ell_2$ | | ROW KL | |
|---|---|---|---|---|---|---|
| | HYP | DOT | HYP | DOT | HYP | DOT |
| 32 | 0.040 | 0.298 | 0.039 | 0.273 | 0.0006 | 0.0361 |
| 64 | 0.050 | 0.321 | 0.050 | 0.322 | 0.0008 | 0.0363 |
| 128 | 0.057 | 0.317 | 0.053 | 0.312 | 0.0010 | 0.0323 |
| 256 | 0.087 | 0.298 | 0.082 | 0.288 | 0.0026 | 0.0269 |
| 512 | 0.108 | 0.271 | 0.110 | 0.255 | 0.0032 | 0.0219 |
| 1024 | 0.096 | 0.243 | 0.106 | 0.261 | 0.0023 | 0.0176 |

**PDE Viewpoint and Properties.** Although this experiment is phrased as kernel fitting, the mapping $u = K_{\text{tree}}s$ can be viewed as a discrete nonlocal diffusion step on a hierarchical domain (the leaves of a binary tree). Since $K_{\text{tree}}$ is nonnegative and row-stochastic, each output entry is a convex combination of inputs, so it obeys a discrete maximum principle:

$$\min_j s_j \leq (K_{\text{tree}}s)_i \leq \max_j s_j, \qquad \|K_{\text{tree}}s\|_\infty \leq \|s\|_\infty, \qquad K_{\text{tree}}\mathbf{1} = \mathbf{1}.$$

This makes the operator stable and mass-preserving on constant modes, consistent with diffusion-type PDE solution operators. Our exponential distance kernel $K_{ij} \propto \exp(-\gamma_{\text{eff}}d_{\text{tree}}(i, j))$ mirrors the off-diagonal decay typical of Green's/heat kernels of elliptic operators on tree-like geometries. Scaling $\gamma_{\text{eff}} = \gamma/L$ keeps the effective interaction range comparable as depth grows (since $d_{\text{tree}}$ scales linearly in $L$), avoiding trivial near-diagonal or near-uniform limits at large $N$.

**Connection to a Graph Elliptic PDE.** Distance-decaying kernels on trees also arise from diffusion/elliptic operators on graphs. For example, consider the (tree-)graph elliptic problem

$$(I + \alpha\mathcal{L} + \beta\mathcal{L}^2)u = s, \tag{30}$$

where $\mathcal{L} = D - A$ is the graph Laplacian of the full binary tree. For $\alpha, \beta \geq 0$ the operator is symmetric positive definite, so the solution exists and is unique, and it acts as a spectral low-pass filter: if $\mathcal{L}\phi_k = \lambda_k\phi_k$, then $\langle u, \phi_k \rangle = g(\lambda_k)\langle s, \phi_k \rangle$ with $g(\lambda) = 1/(1 + \alpha\lambda + \beta\lambda^2)$. The resulting smoothing couples leaves across multiple scales induced by shared ancestors, which is precisely the hierarchical structure the toy is designed to expose.

# I. Training and Implementation Details

**Model Variants.** HNO uses the same hyperbolic-distance Gibbs kernel across all datasets, with tokenization chosen to match the underlying discretization. For point clouds (Elasticity), we use a learned latent set of $M$ tokens and perform latent↔point cross-attention to encode/decode, with hyperbolic self-attention operating on the latent tokens. For regular grids and structured meshes (Navier–Stokes, Darcy, Airfoil, Pipe), we tokenize the field into non-overlapping 2D patches via a strided convolutional embedding; attention is computed between patch tokens (with a 2D relative position bias), and predictions are mapped back to the original resolution with an unpatching operator. For the spatiotemporal Plasticity setting, we keep the same patch tokenization and hyperbolic attention, but use a stronger local feature extractor to better preserve fine-scale dynamics.

**Computational Complexity.** Let $N$ be the number of input discretization nodes and let $M$ be the number of tokens processed by the interaction core. Dense attention over all nodes incurs $O(N^2)$ pairwise interactions per layer. For point clouds, we use a latent set of $M$ tokens with cross-attention for encoding and decoding, which costs $O(NM)$, and hyperbolic self-attention on the latent set, which costs $O(M^2)$. This yields an overall interaction cost $O(NM + M^2)$ per layer. For regular grids and structured meshes, patchification produces $M$ patch tokens and self-attention runs in the patch space with cost $O(M^2)$, where $M \ll N$ at fixed patch size. Replacing dot-product logits by stabilized hyperbolic-distance logits does not change these asymptotic costs, since computing Minkowski inner products and arcosh is constant-time per token pair.

**Training Protocol.** Unless otherwise noted, we train PDEBench benchmarks for 500 epochs; Elasticity uses 300 epochs and Airfoil uses 800 epochs. For PDEBench, we follow the standard split of 1000 training and 200 test samples and report mean relative $\ell_2$ error (averaged over three runs). We use AdamW with weight decay $10^{-5}$ and apply gradient clipping

*Table 9.* Best hyperparameters for HNO on PDEBench. Here $d$ is hidden width, $L$ is depth, $H$ is the number of heads, and $d_\mathbb{H}$ is the hyperbolic Q/K dimension. Tokenization is either a learned latent set ($M$ tokens) or patchification with patch size ps.

| DATASET | TOKENS | $d$ | $L$ | $H$ | $d_\mathbb{H}$ | LR | EPOCHS | BATCH | CLIP | SCHED. |
|---|---|---|---|---|---|---|---|---|---|---|
| ELASTICITY | $M$=96 | 384 | 6 | 8 | 16 | $5\times10^{-4}$ | 300 | 4 | – | COSINE |
| NAVIER–STOKES | ps=4 | 132 | 4 | 4 | 16 | $1\times10^{-3}$ | 500 | 8 | 0.5 | ONECYCLE |
| DARCY | ps=5 | 96 | 4 | 4 | 16 | $1\times10^{-3}$ | 500 | 4 | 0.1 | ONECYCLE |
| PLASTICITY | ps=3 | 128 | 5 | 8 | 16 | $8\times10^{-4}$ | 500 | 16 | 0.1 | ONECYCLE |
| AIRFOIL | ps=5 | 56 | 5 | 4 | 16 | $1\times10^{-3}$ | 800 | 4 | 0.5 | ONECYCLE |
| PIPE | ps=3 | 132 | 4 | 4 | 16 | $1\times10^{-3}$ | 500 | 4 | 1.0 | ONECYCLE |

*Table 10.* Top-3 hyperparameter configurations per PDEBench dataset (lower Rel. $\ell_2$ is better; best per dataset in bold).

| DATASET | TOKENS | $d$ | $L$ | $H$ | $d_\mathbb{H}$ | LR | PARAMS | REL. $\ell_2$ |
|---|---|---|---|---|---|---|---|---|
| ELASTICITY | M=96 | 384 | 6 | 8 | 16 | $5\times10^{-4}$ | 15.37M | **0.00368971** |
| ELASTICITY | M=64 | 320 | 8 | 8 | 16 | $1\times10^{-3}$ | 14.23M | 0.00374535 |
| ELASTICITY | M=64 | 256 | 10 | 8 | 16 | $1\times10^{-3}$ | 11.47M | 0.00410951 |
| NAVIER–STOKES | PS=4 | 132 | 4 | 4 | 16 | $1\times10^{-3}$ | 1.19M | **0.0675513** |
| NAVIER–STOKES | PS=4 | 128 | 4 | 4 | 16 | $1.2\times10^{-3}$ | 1.12M | 0.0679059 |
| NAVIER–STOKES | PS=4 | 128 | 4 | 4 | 16 | $1\times10^{-3}$ | 1.12M | 0.0689504 |
| DARCY | PS=5 | 96 | 4 | 4 | 16 | $1\times10^{-3}$ | 0.82M | **0.00446** |
| DARCY | PS=5 | 96 | 5 | 4 | 16 | $1\times10^{-3}$ | 0.91M | 0.00446 |
| DARCY | PS=5 | 104 | 4 | 4 | 16 | $1\times10^{-3}$ | 0.91M | 0.004462 |
| PLASTICITY | PS=3 | 128 | 5 | 8 | 16 | $8\times10^{-4}$ | 2.27M | **0.000903203** |
| PLASTICITY | PS=5 | 128 | 4 | 8 | 16 | $8\times10^{-4}$ | 2.38M | 0.00107384 |
| PLASTICITY | PS=3 | 128 | 4 | 8 | 16 | $5\times10^{-4}$ | 1.89M | 0.0011549 |
| AIRFOIL | PS=5 | 56 | 5 | 4 | 16 | $1\times10^{-3}$ | 0.36M | **0.00508543** |
| AIRFOIL | PS=5 | 96 | 6 | 4 | 16 | $5\times10^{-4}$ | 1.17M | 0.00510241 |
| AIRFOIL | PS=5 | 112 | 5 | 4 | 16 | $5\times10^{-4}$ | 1.23M | 0.00515098 |
| PIPE | PS=3 | 132 | 4 | 4 | 16 | $1\times10^{-3}$ | 5.10M | **0.00273442** |
| PIPE | PS=3 | 128 | 3 | 4 | 16 | $1\times10^{-3}$ | 3.86M | 0.00284324 |
| PIPE | PS=3 | 132 | 4 | 4 | 16 | $1.2\times10^{-3}$ | 5.10M | 0.0029343 |

**Discussion (Elasticity).** In our Elasticity sweeps, the best-performing configurations use larger hidden widths (and thus larger parameter counts), which we attribute to the harder encoding problem induced by irregular, nonuniform point sampling. This is orthogonal to overly smooth tokenizations that can suppress high-frequency modes; we believe designing geometry-aware tokenizers/encoders that better preserve fine-scale content on irregular 2D/3D point sets is a valuable direction, but we do not explore it in this work.

when needed for stability. Elasticity uses cosine learning-rate annealing, while the other PDEBench datasets use a OneCycle learning-rate schedule (stepped per batch).

**Preprocessing and Normalization.** For Darcy/Pipe/Elasticity, we standardize targets (and inputs where applicable) using dataset statistics and decode predictions back to the original scale before reporting relative $\ell_2$. For Airfoil, we do not normalize the output and (for our best model) keep the input coordinates unnormalized due to the native physical-coordinate range.

**PDEBench Benchmarks.** Tab. 9 summarizes the best hyperparameters used for HNO on the six PDEBench benchmarks.

**Hyperparameter Search Robustness (Top-3).** To demonstrate robustness and reduce concerns about configuration cherry-picking, we report the top-3 configurations (sorted by test relative $\ell_2$) from our lightweight sweep on each benchmark, using the same data split and evaluation metric as in the main results. Each row lists the tokenization parameter ($M$ or patch size ps), architecture hyperparameters ($d, L, H, d_\mathbb{H}$), learning rate, parameter count, and the corresponding test error. These rows summarize lightweight sweep configurations used to select architectures; final main-table results may use the selected configuration under the main evaluation protocol.

**Large-Scale Benchmarks.** We implement HNO on large-scale unstructured meshes by replacing dot-product logits in Transolver++ with stabilized hyperbolic-distance logits, keeping the same backbone width and depth. We use $d$=256, $L$=8, $H$=8, 64 slices, and $d_\mathbb{H}$=16, trained with Adam and a OneCycle learning-rate schedule and batch size 1. Tab. 11 summarizes the dataset-specific settings.

*Table 11.* Large-scale unstructured-mesh benchmarks: training setup.

| DATASET | #NODES | LR | EPOCHS | GRAPH | SUBSAMPLE | NOTES |
|---------|--------|-----|--------|-------|-----------|-------|
| SHAPENET CAR | 32,186 | $1 \times 10^{-3}$ | 500 | RADIUS ($r=0.2$) | – | LOSS IS VELOCITY MSE PLUS WEIGHTED SURFACE-PRESSURE MSE ($\lambda=0.5$). |
| AIRFRANS | 32,000 | $1 \times 10^{-3}$ | 500 | RADIUS ($r=0.05$) | 32K | PER-EPOCH RESAMPLING WITH MAX NEIGHBORS 64; WEIGHTED MSE (SURFACE+VOLUME). |

*Table 13.* Mixed-precision stability check on Darcy ($85 \times 85$, batch 4).

| PRECISION | MODE | STEPS | NANS |
|-----------|------|-------|------|
| FP16 | TRAIN (AMP + GRADSCALER) | 200 | NO |
| BF16 | FORWARD-ONLY | 200 | NO |

**Visualization of Large-Scale Error Maps.** For AirfRANS (2D), we load the ground-truth internal mesh and model predictions (VTK `.vtu`), compute signed errors in velocity magnitude and pressure, and render them as filled-contour maps using the original mesh connectivity (triangulating quads when needed). Color limits are set per sample using symmetric percentile clipping and shared across models for fair visual comparison; the airfoil boundary is overlaid for context. For ShapeNet Car (3D), we render a semi-transparent outer-domain surface colored by signed speed residual ($\|U_{\mathrm{pred}}\| - \|U_{\mathrm{gt}}\|$) together with a surface pressure error map ($p_{\mathrm{pred}} - p_{\mathrm{gt}}$); meshes are aligned to a consistent upright orientation across cases.

## J. Additional Experiments

### J.1. Ablations and Sensitivity

**Darcy sensitivity.** We sweep hidden dimension, the number of attention layers, and hyperbolic Q/K dimension while keeping other settings fixed (Fig. 8). Tab. 3 reports the official baseline configurations, which are not parameter-matched to HNO; for a parameter-matched efficiency microbenchmark, see Tab. 12.

*Table 12.* Parameter-matched efficiency microbenchmark on Darcy (A6000 48GB, $85 \times 85$ grid, batch 4). "Train" is extrapolated from the measured per-step time to 500 epochs with 1000 training samples.

| METHOD | PARAMS | VRAM (GB)↓ | TRAIN (H)↓ | INFER (MS/B)↓ |
|--------|--------|------------|------------|---------------|
| HNO (OURS) | 0.82M | 0.22 | 1.18 | 5.69 |
| TRANSOLVER++ (OFFICIAL) | 2.84M | 3.07 | 4.03 | 54.79 |
| TRANSOLVER++ (PARAM-MATCHED) | 0.82M | 1.53 | 1.97 | 50.84 |

**Parameter-matched efficiency on Darcy.** To isolate architectural efficiency from model size, we benchmark a parameter-matched Transolver++ configuration on Darcy by reducing hidden width and depth while keeping the slicing mechanism unchanged. We report peak allocated GPU memory during a single training step, along with per-batch inference latency. All measurements use an NVIDIA A6000 (48GB), a $85 \times 85$ grid (downsample 5), and batch size 4.

### J.2. Numerical Stability and Runtime Breakdown

**Mixed-precision stability (Darcy).** We run a short stability check on Darcy ($85 \times 85$, batch 4). FP16 uses AMP with GradScaler for 200 optimization steps and produces no NaNs. Our current PyTorch build does not support BF16 backprop through rsqrt, so we report a BF16 forward-only check.

**Runtime breakdown (Darcy).** We profile a forward pass on Darcy ($85 \times 85$, batch 4) and report a coarse operator-level time breakdown. Hyperbolic distance nonlinearity (arcosh) accounts for $< 1\%$ of self CUDA time.

### J.3. Spectral Error Analysis on Darcy

We analyze the frequency content of prediction errors on Darcy ($85 \times 85$) by computing a radial average of the relative error spectrum over 200 test samples: for each radial bin $b$, we report $\sqrt{\sum_{\|k\| \in b} |\hat{e}(k)|^2 / \sum_{\|k\| \in b} |\hat{u}(k)|^2}$, where $e = u_{\mathrm{pred}} - u_{\mathrm{gt}}$.

*Table 14.* Runtime breakdown on Darcy (85×85, batch 4). Percentages are relative to total self CUDA time from a CUDA profiler trace; latency is measured by CUDA events.

| METHOD | LATENCY (MS/B)↓ | arcosh (%) | MATMUL+EINSUM (%) | CONV (%) |
|---|---|---|---|---|
| HNO (HYPERBOLIC) | 5.87 | 0.8 | 15.7 | 25.0 |
| −HYPERBOLIC (EUCLID) | 5.18 | 0.0 | 16.7 | 22.1 |

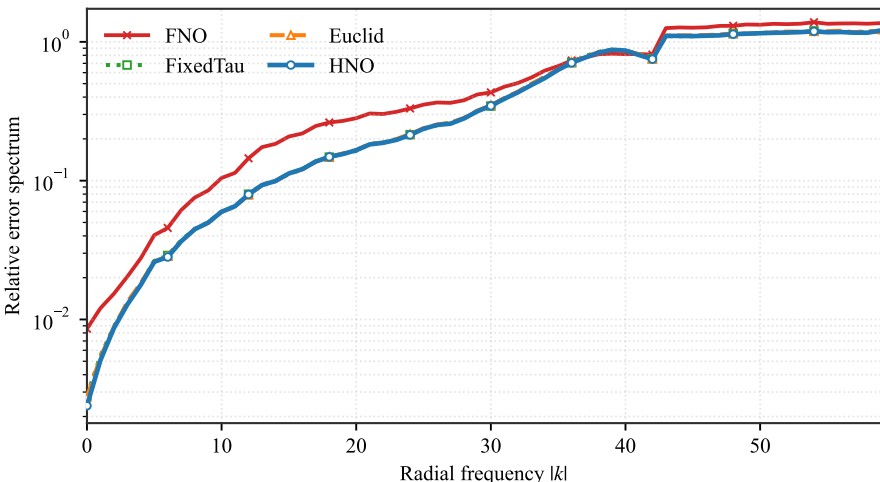

*Figure 11.* Darcy error spectrum (log scale). HNO yields uniformly lower spectral error than Euclidean-distance and fixed-temperature ablations, and does not exhibit increased high-frequency error relative to an FNO baseline.

## J.4. Causal Radius Interventions

**Motivation.** Fig. 9 shows that learned hyperbolic radii correlate with attention locality. To test whether radius is a *causal* control signal rather than a byproduct of training, we perform radius interventions that preserve the Q/K directions but disrupt their norms.

**Interventions.** Let $q_i \in \mathbb{R}^{d_{\mathbb{H}}}$ and $k_j \in \mathbb{R}^{d_{\mathbb{H}}}$ denote Euclidean features that are lifted to the Lorentz model. Write $q_i = \|q_i\|\bar{q}_i$ and $k_j = \|k_j\|\bar{k}_j$ with $\|\bar{q}_i\| = \|\bar{k}_j\| = 1$, so the radius is $r(q_i) = \mathrm{asinh}(\|q_i\|)$. We consider two interventions applied at inference time: (i) *Fixed norm:* set all $\|q_i\|$ (and all $\|k_j\|$) to their within-head mean while keeping $\bar{q}_i, \bar{k}_j$ unchanged; (ii) *Shuffled norm:* randomly permute the set of $\{\|q_i\|\}_i$ (and $\{\|k_j\|\}_j$) within each head, again keeping directions unchanged. Both interventions keep angles in feature space and the value pipeline fixed, while removing meaningful radius stratification.

**Results (Darcy).** Tab. 15 reports the mean relative $\ell_2$ error on Darcy (85×85). Both interventions substantially degrade accuracy, indicating that learned radii are important for this checkpoint's near–far routing behavior.

*Table 15.* Causal radius interventions on Darcy (85×85).

| METHOD | REL. $\ell_2$ ↓ |
|---|---|
| HNO (BASELINE) | 0.00446 |
| FIXED NORM (Q/K) | 0.03269 |
| SHUFFLED NORM (Q/K) | 0.03328 |

## J.5. Learned Kernel Decay

Assumption D.2 posits that the far-field kernel is smooth and decays away from the near field. As an empirical sanity check, we visualize distance-dependent decay in learned attention weights on Elasticity. For a representative global token (small hyperbolic radius) and a representative local token (large hyperbolic radius), we compute each token's physical barycenter $b$ and plot $\log_{10}$ attention weights against the physical distance $\|x - b\|_2$. Local tokens exhibit a much faster decay, while global tokens remain broadly connected.

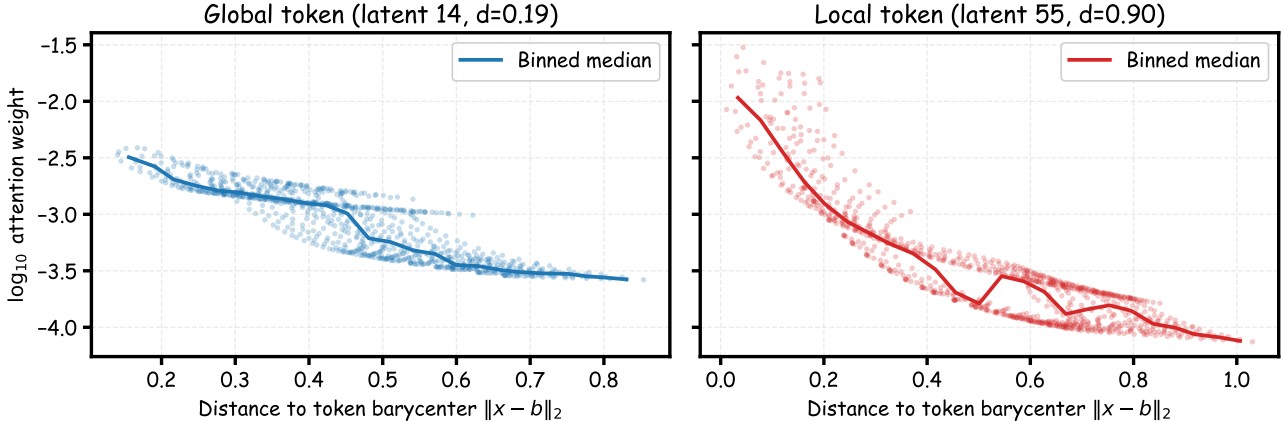

*Figure 12.* Elasticity: distance-dependent decay of learned attention weights for representative global and local tokens. The curve shows the binned median of $\log_{10}$ attention weights over distance to the token-specific barycenter.

## J.6. Discretization Robustness

**Quadrature weights in the discrete operator.** Our Nyström discretization uses nodes $\{x_j\}_{j=1}^N$ with nonnegative weights $\{w_j\}_{j=1}^N$ (Eq. (11)). On PDEBench, we use uniform weights $w_j = \mu(D)/N$ (equivalently $w_j = 1$ up to a global scaling), so the implemented coefficients reduce to standard softmax attention weights.

**Scope of the consistency result.** The discretization consistency theorem (Appendix G) assumes a Lipschitz-accurate quadrature rule (Assumption G.3), which can be interpreted as a mesh-quality condition. This covers quasi-uniform grids and well-behaved point clouds whose empirical measures converge to the underlying continuum measure in $W_1$ at rate $O(h)$. The theorem does not apply to arbitrary resampling or severe subsampling that violates this quadrature accuracy condition. Empirically, our reported benchmarks evaluate each dataset on its native discretization. To provide a concrete discretization-density sanity check, Appendix J.6.1 additionally reports an AirfRANS node subsampling scan that varies the available mesh nodes at test time. We leave robustness to aggressive remeshing schemes or distribution-shifted point sampling as future work.

### J.6.1. AIRFRANS DISCRETIZATION DENSITY SCAN

**Setup.** We evaluate Transolver-series models on AirfRANS by varying the number of nodes available at test time. For a representative test case, we uniformly subsample the mesh nodes and report results at representative node budgets $N \in \{2\text{k}, 32\text{k}, 128\text{k}\}$. We report mean relative $\ell_2$ error in physical units, peak GPU memory, and end-to-end forward latency on an RTX A6000 48GB.

*Table 16.* AirfRANS subsampling scan on a representative test case. Each entry reports mean relative $\ell_2$ / peak VRAM (GB) / forward latency (ms) at the given node budget $N$.

| $N$ | TRANSOLVER | TRANSOLVER++ | HNO |
|---|---|---|---|
| 2K | 0.509 / 0.28 / 126.0 | 0.334 / 0.29 / 44.9 | 0.165 / 0.31 / 76.4 |
| 32K | 0.478 / 3.49 / 148.9 | 0.237 / 3.75 / 171.1 | 0.106 / 3.98 / 186.0 |
| 128K | 0.469 / 13.65 / 533.6 | 0.235 / 14.72 / 535.2 | 0.105 / 15.59 / 515.7 |

## K. Additional Visualizations

### K.1. Qualitative Results on PDEBench

We provide additional qualitative visualizations for all six PDEBench datasets using the HNO best checkpoints. Each row corresponds to one test case; columns show input (when available), ground truth (GT), HNO prediction, and error (prediction minus GT).

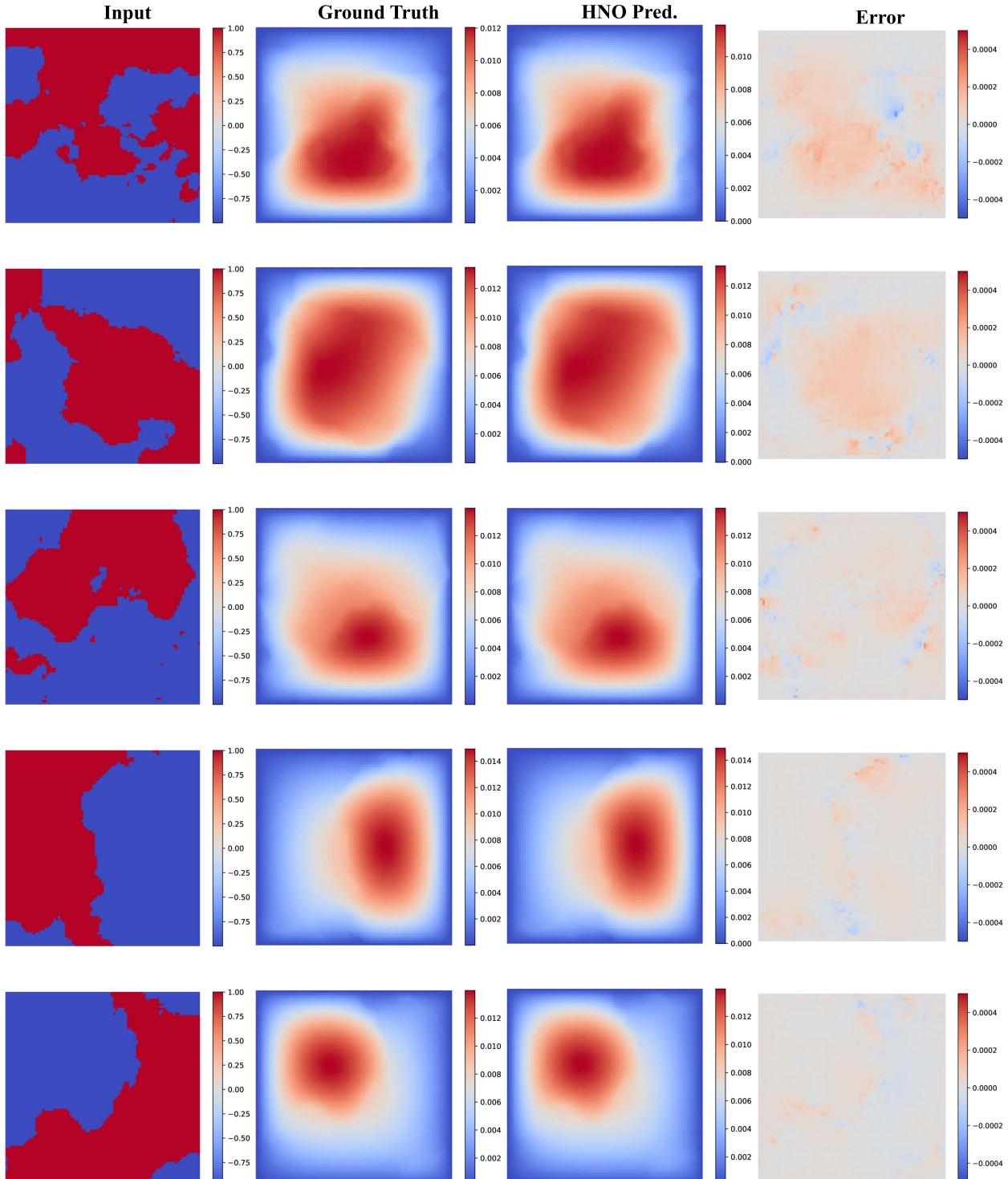

*Figure 13.* Darcy qualitative results (page 1/2). Columns show Input / GT / HNO prediction / Error (prediction minus GT).

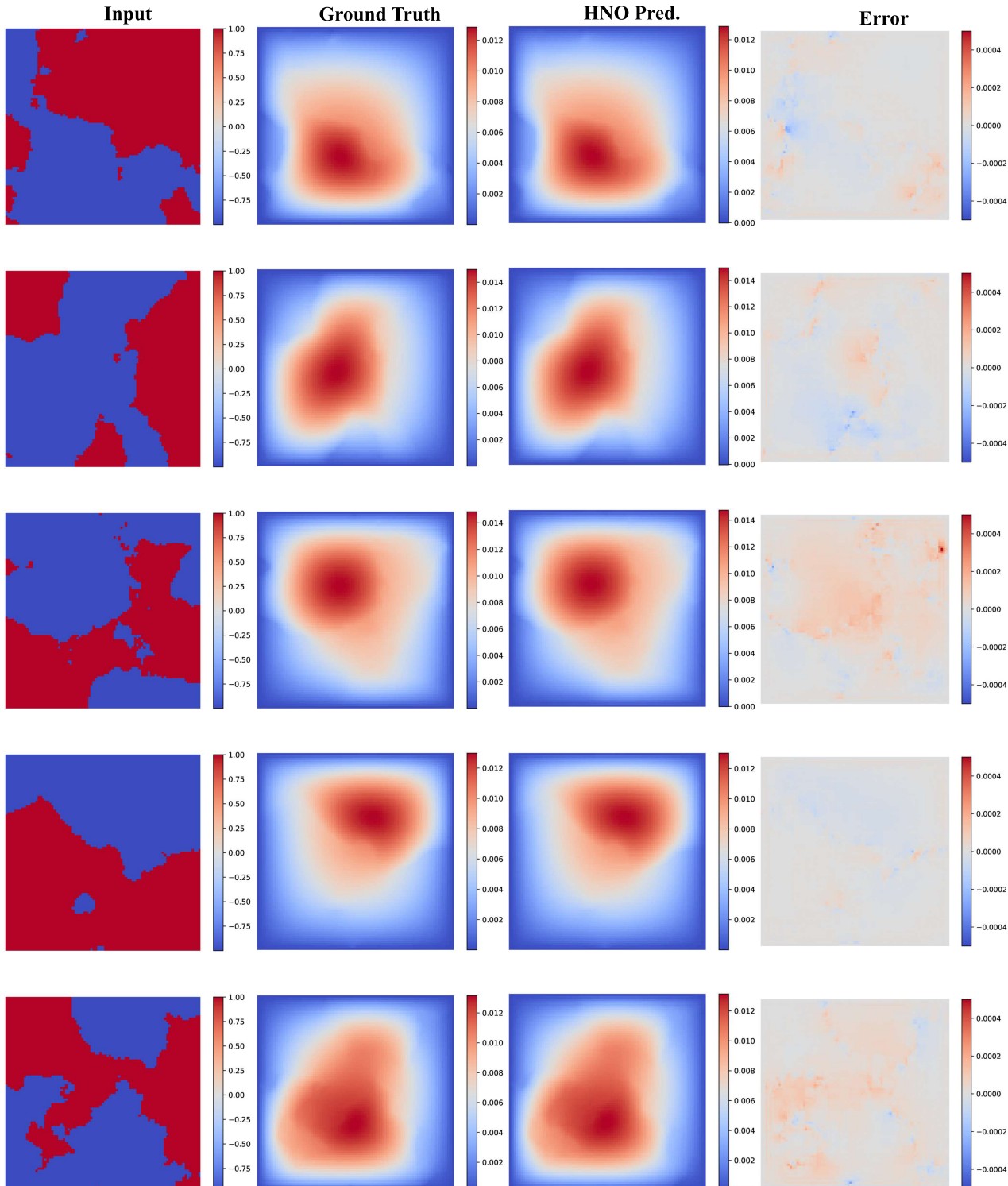

*Figure 14.* Darcy qualitative results (page 2/2). Columns show Input / GT / HNO prediction / Error (prediction minus GT).

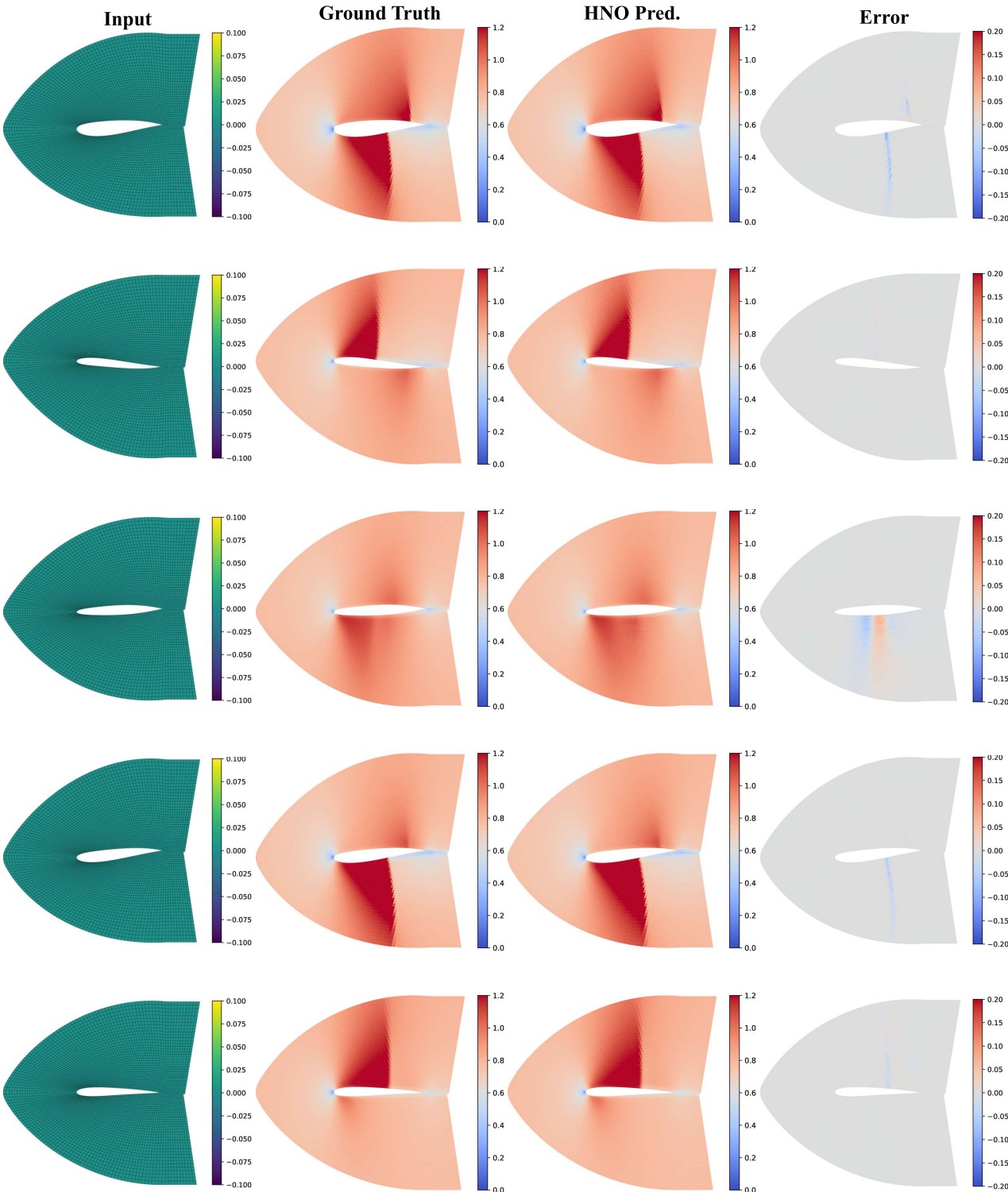

*Figure 15.* Airfoil qualitative results (page 1/2). Columns show Input / Ground Truth / HNO prediction / Error (prediction minus GT).

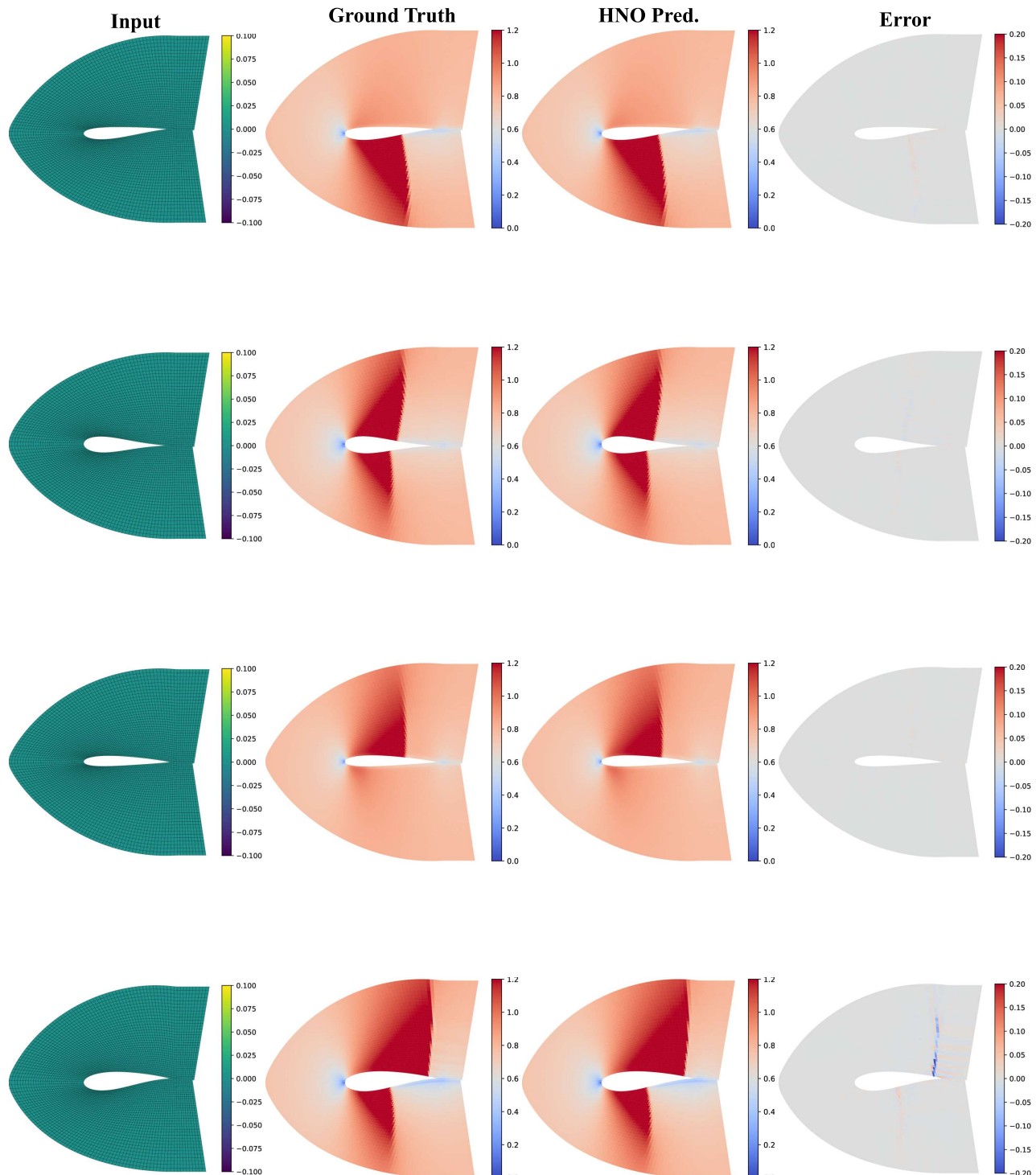

*Figure 16.* Airfoil qualitative results (page 2/2). Columns show Input / Ground Truth / HNO prediction / Error (prediction minus GT).

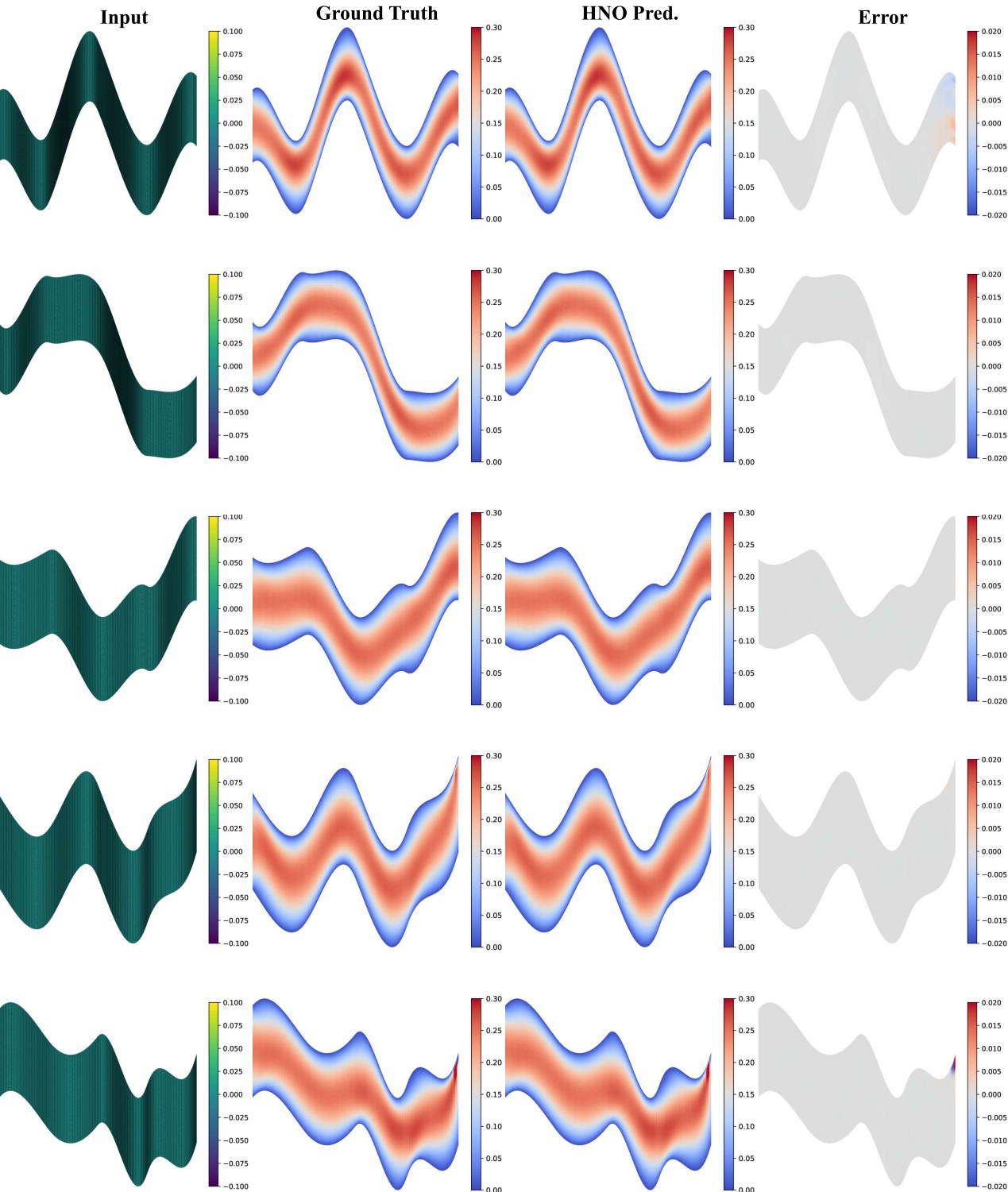

*Figure 17.* Pipe qualitative results (page 1/2). Columns show Input / Ground Truth / HNO prediction / Error (prediction minus GT).

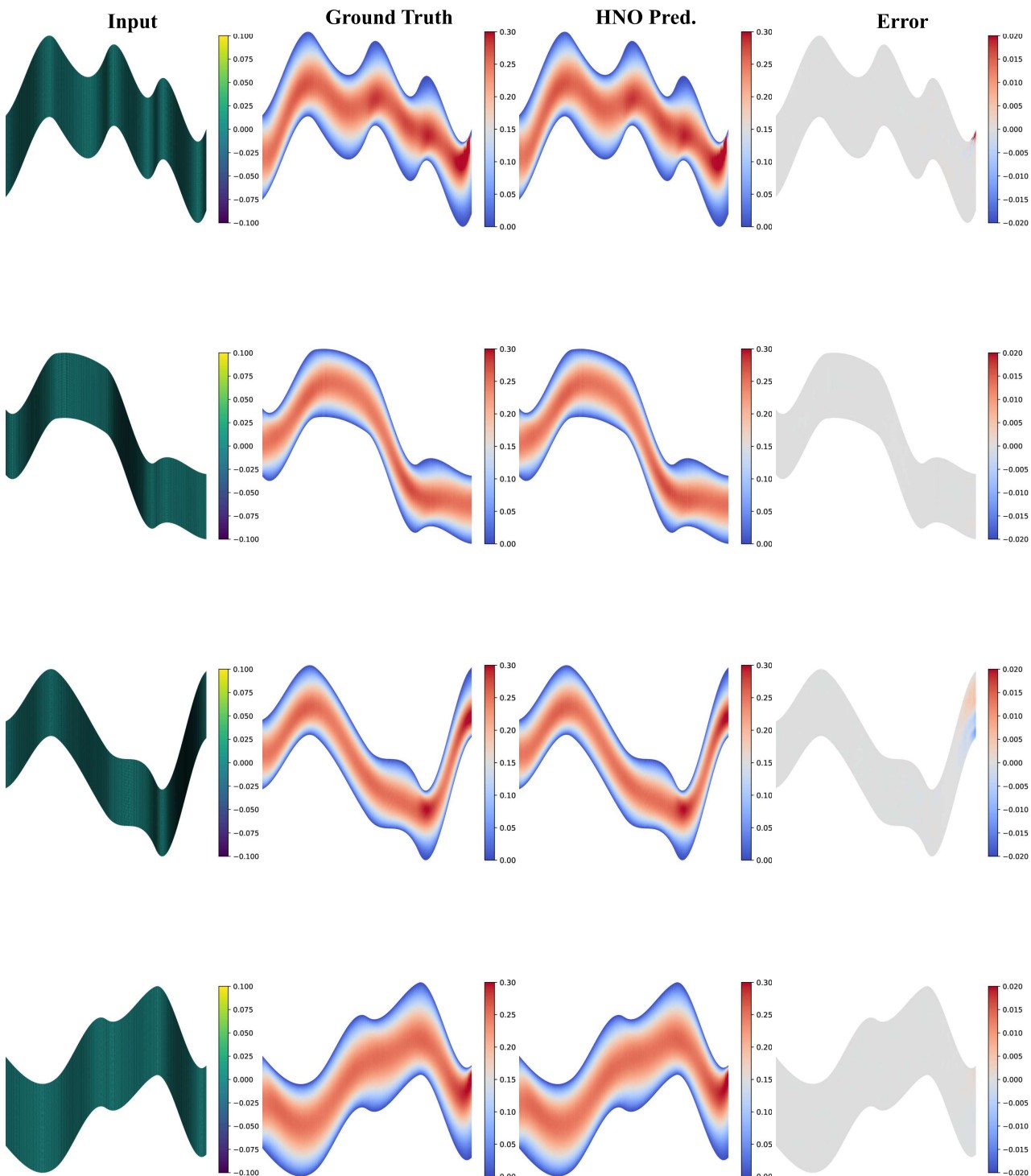

*Figure 18.* Pipe qualitative results (page 2/2). Columns show Input / Ground Truth / HNO prediction / Error (prediction minus GT).

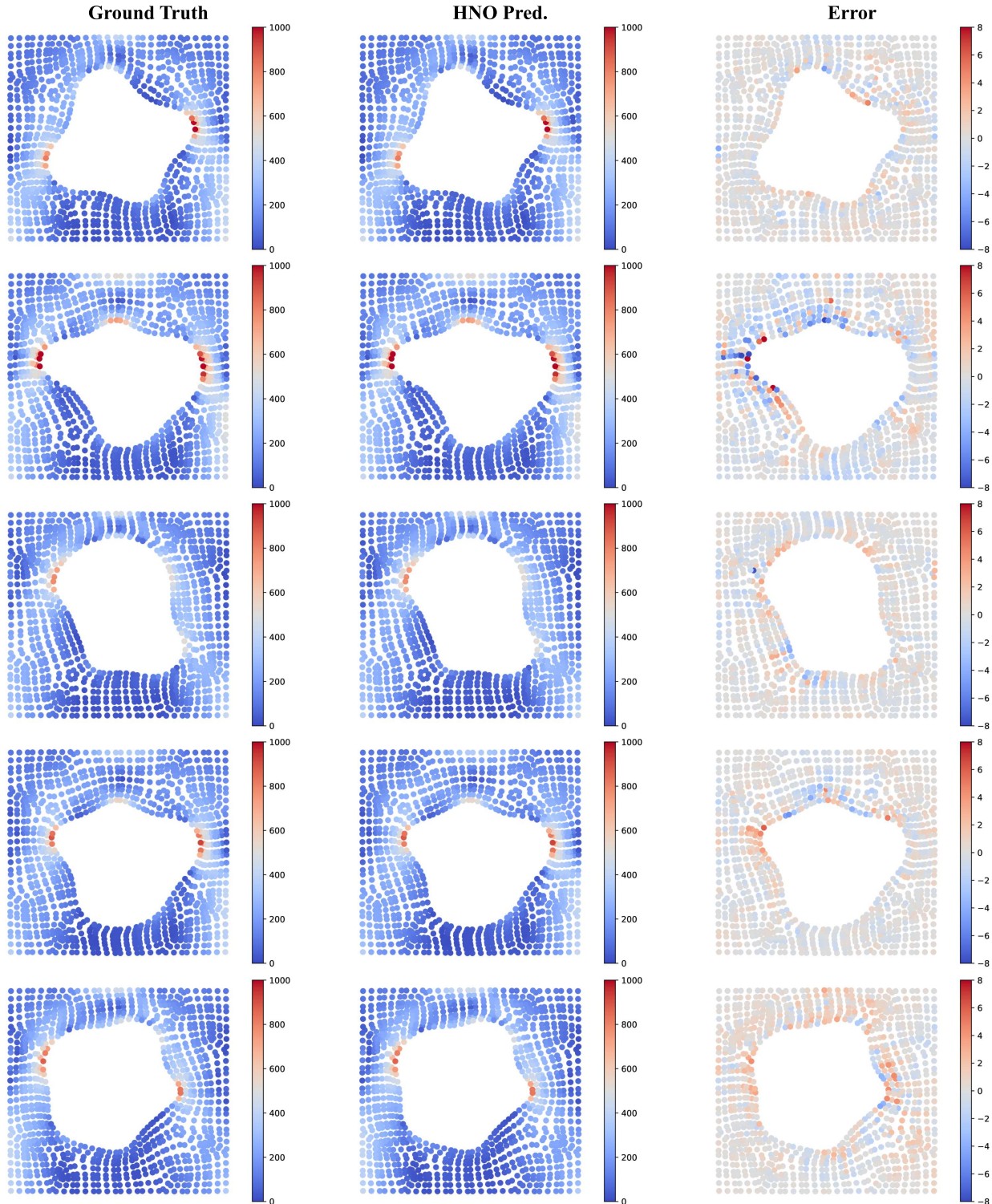

*Figure 19.* Elasticity qualitative results (page 1/2). Columns show Ground Truth / HNO prediction / Error (prediction minus GT).

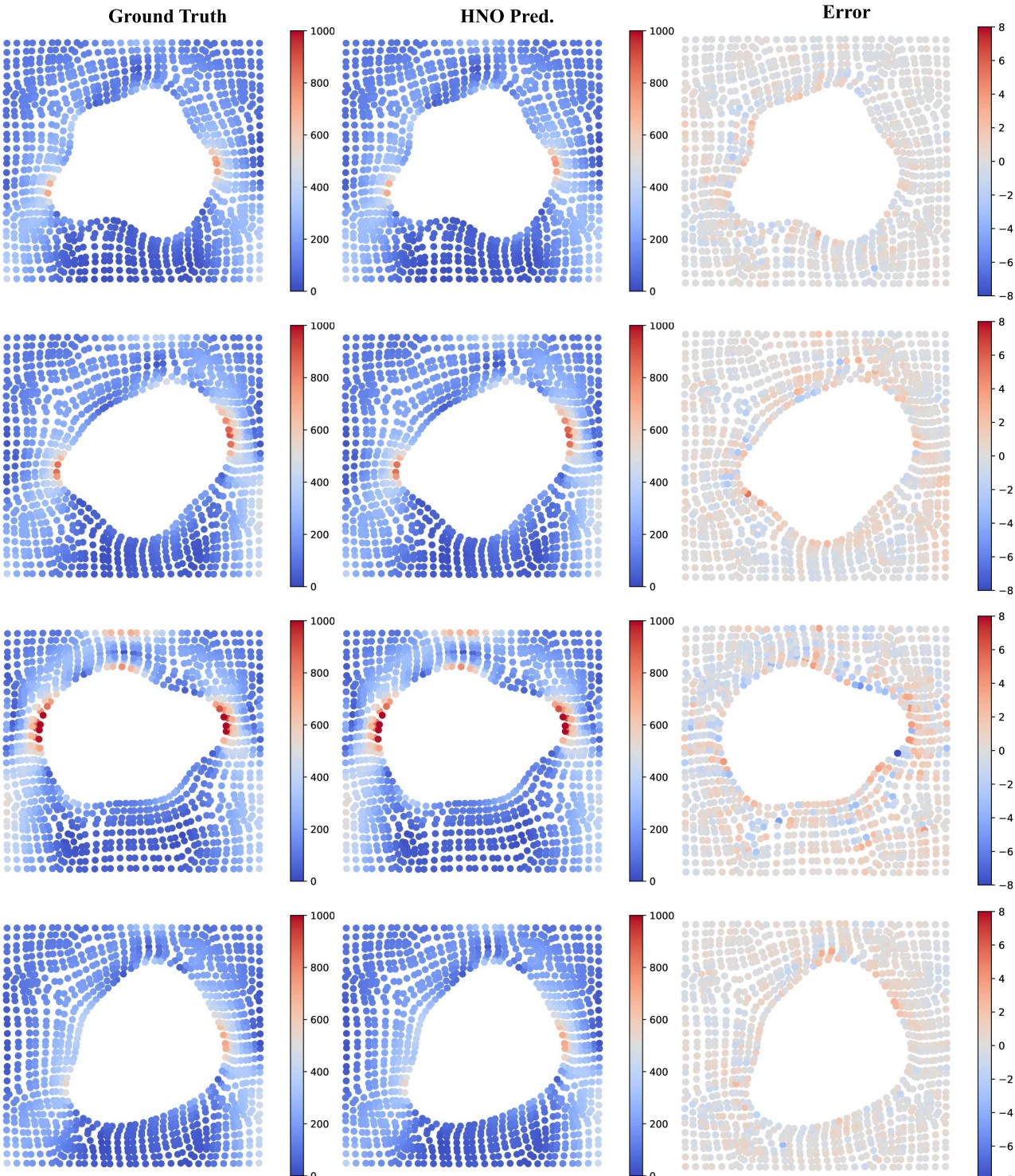

*Figure 20.* Elasticity qualitative results (page 2/2). Columns show Ground Truth / HNO prediction / Error (prediction minus GT).

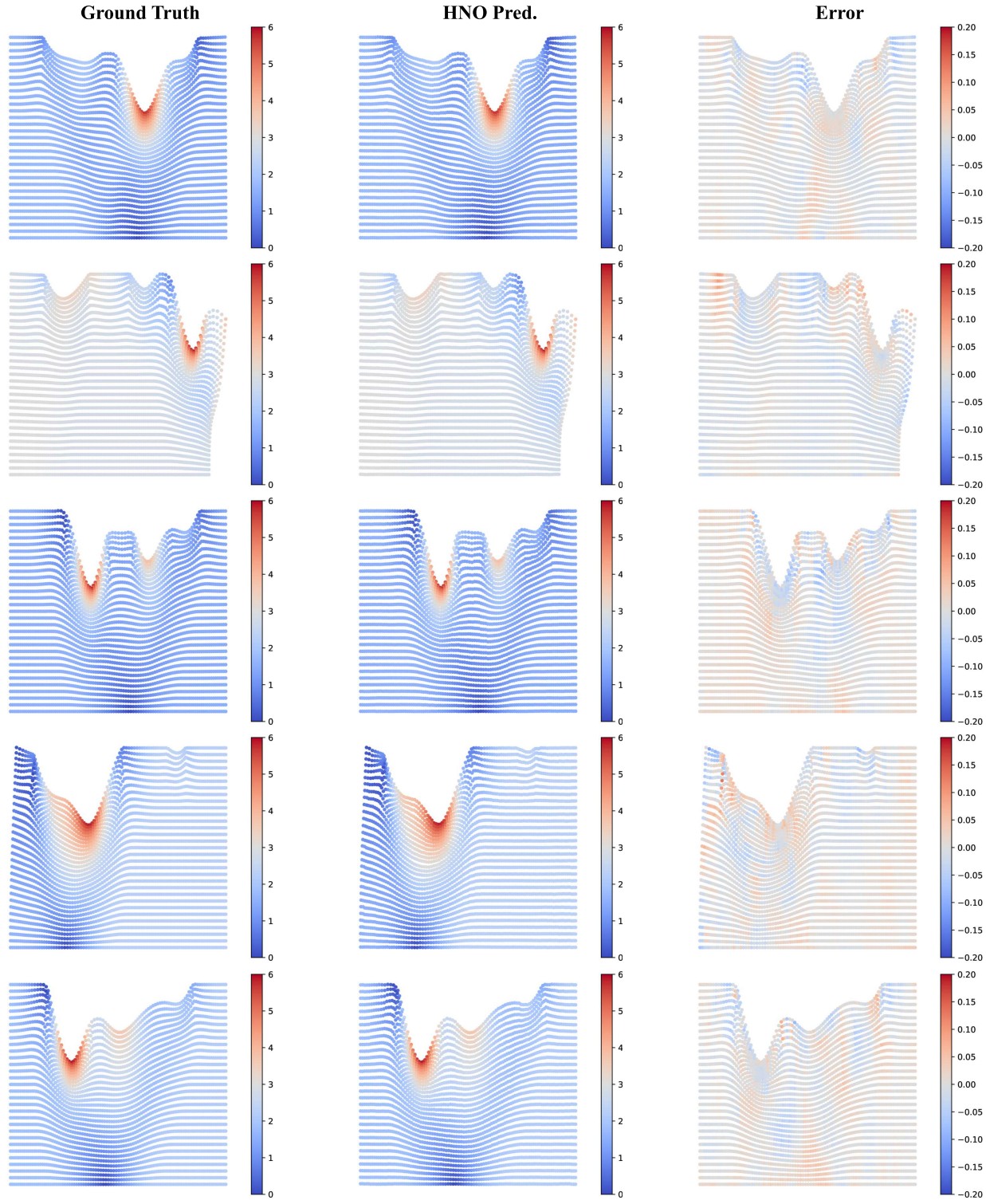

*Figure 21.* Plasticity qualitative results (page 1/1). Columns show Ground Truth / HNO prediction / Error (prediction minus GT).

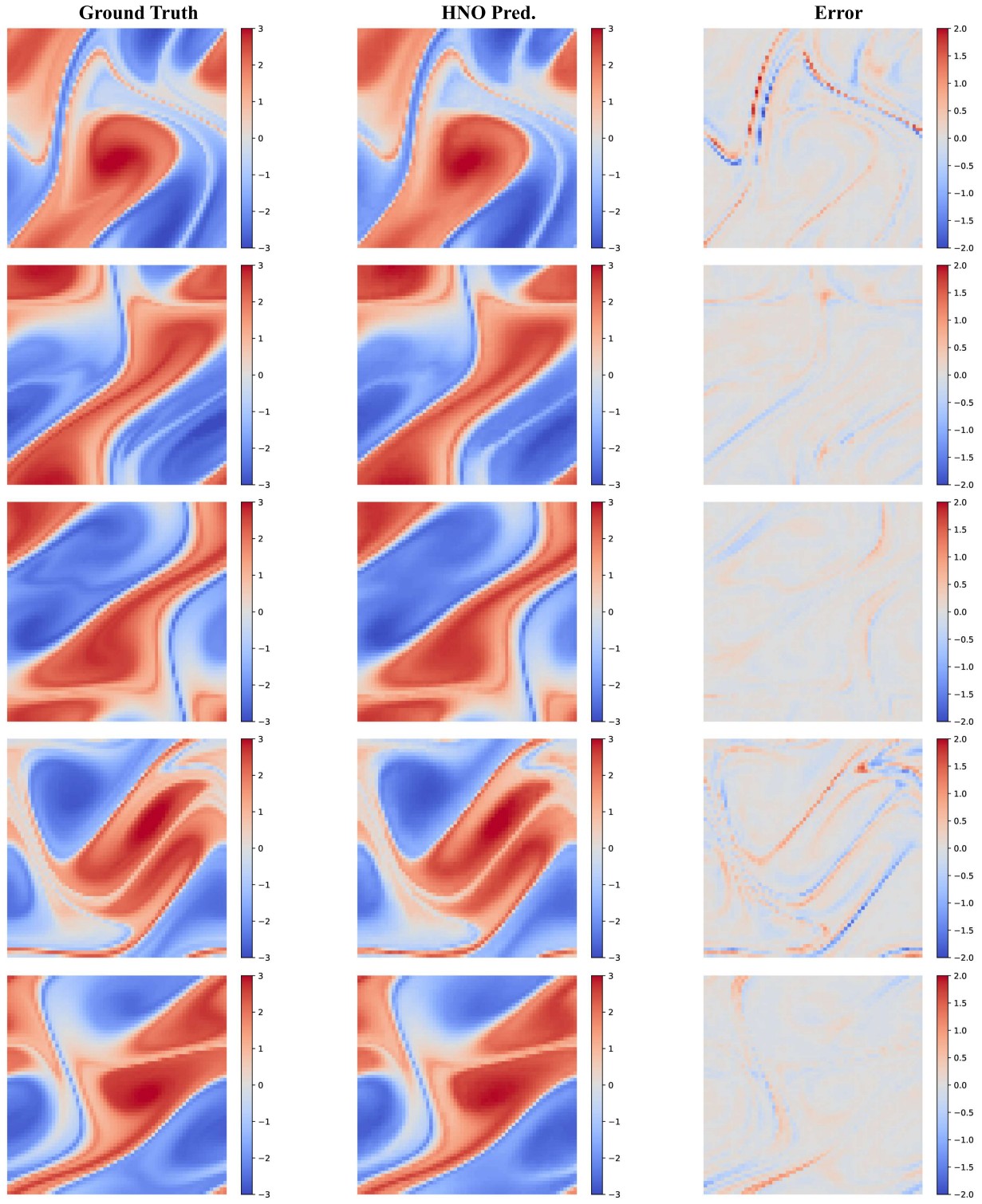

*Figure 22.* Navier–Stokes qualitative results (page 1/2). Columns show Ground Truth / HNO prediction / Error (prediction minus GT).

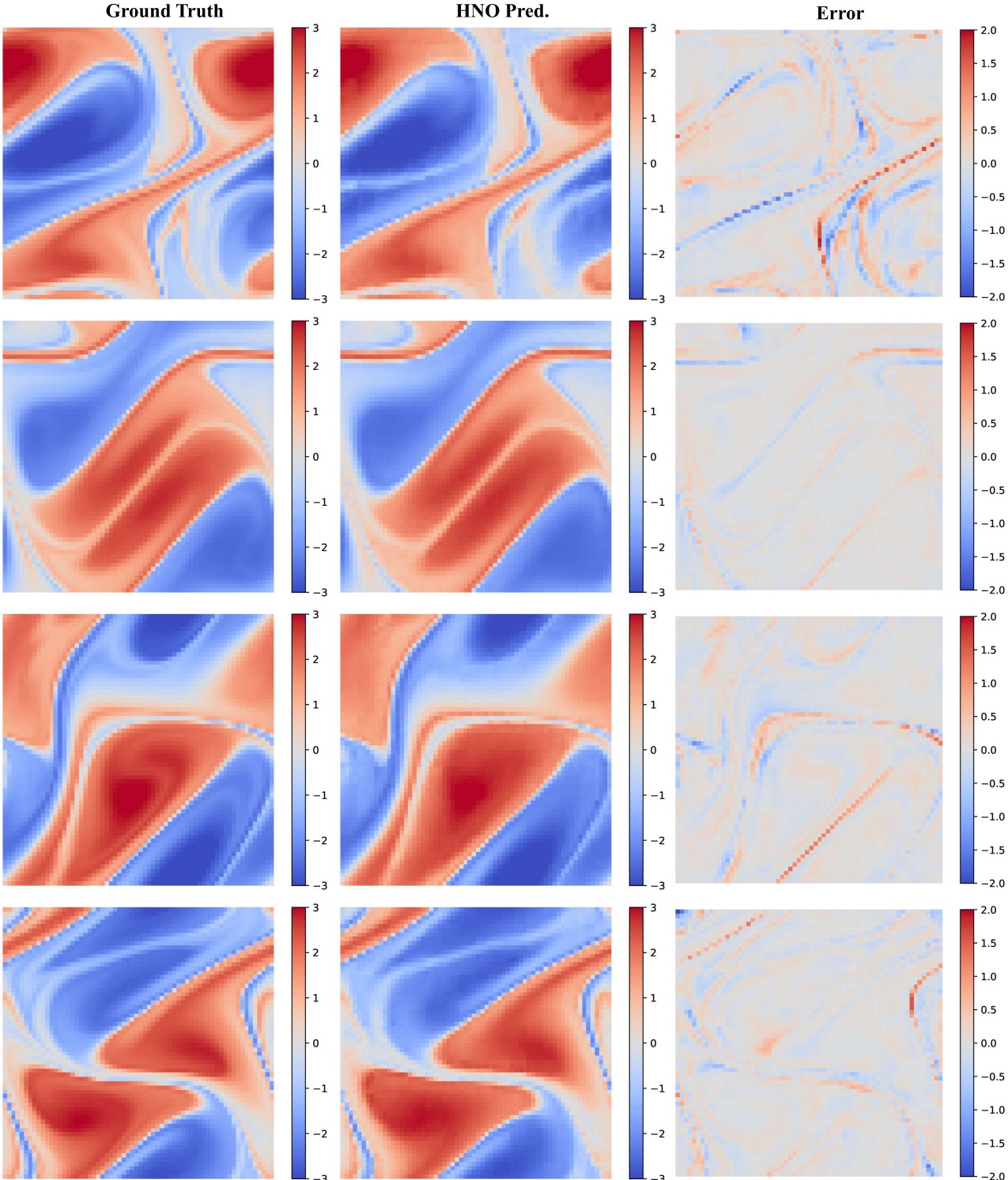

*Figure 23.* Navier–Stokes qualitative results (page 2/2). Columns show Ground Truth / HNO prediction / Error (prediction minus GT).

## K.2. Direct Attention Map Visualization

Figures 24–31 visualize representative global (small radius) and local (large radius) query tokens on Elasticity using cached attention weights. These examples support the qualitative trend that larger-radius tokens attend more sharply: their heatmaps concentrate around a tight neighborhood and their distance-decay curves drop faster. At the same time, the separation is not a hard global–local dichotomy. First, locality is a *continuous* property: tokens with intermediate radii can exhibit partially overlapping receptive fields, and even "global" tokens can be anisotropic (e.g., emphasizing particular boundary/geometry regions). Second, overlap is expected when the selected queries are spatially close or when the solution requires both near-field fidelity and weak far-field conditioning (e.g., to satisfy global constraints), which leaves nonzero long-range tails after normalization. Overall, the hyperbolic kernel induces a *learnable routing bias* rather than an explicit partition: radius interacts with angular alignment and temperature, so the effective interaction range adapts smoothly to the input and the local geometry.

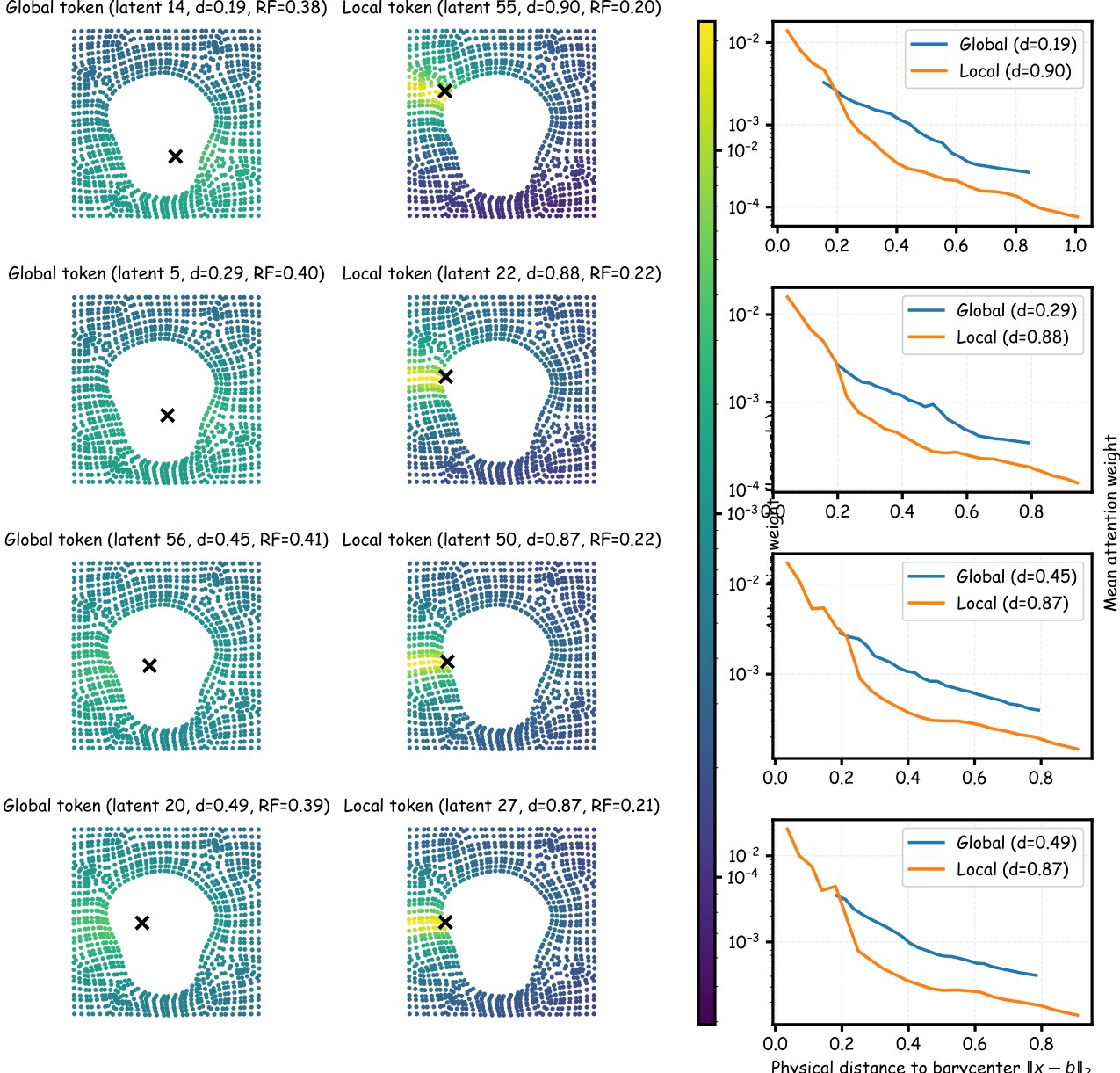

*Figure 24.* Additional attention-map examples on Elasticity, page 1. Each row shows a global token (left), a local token (middle), and their binned mean attention decay versus distance (right).

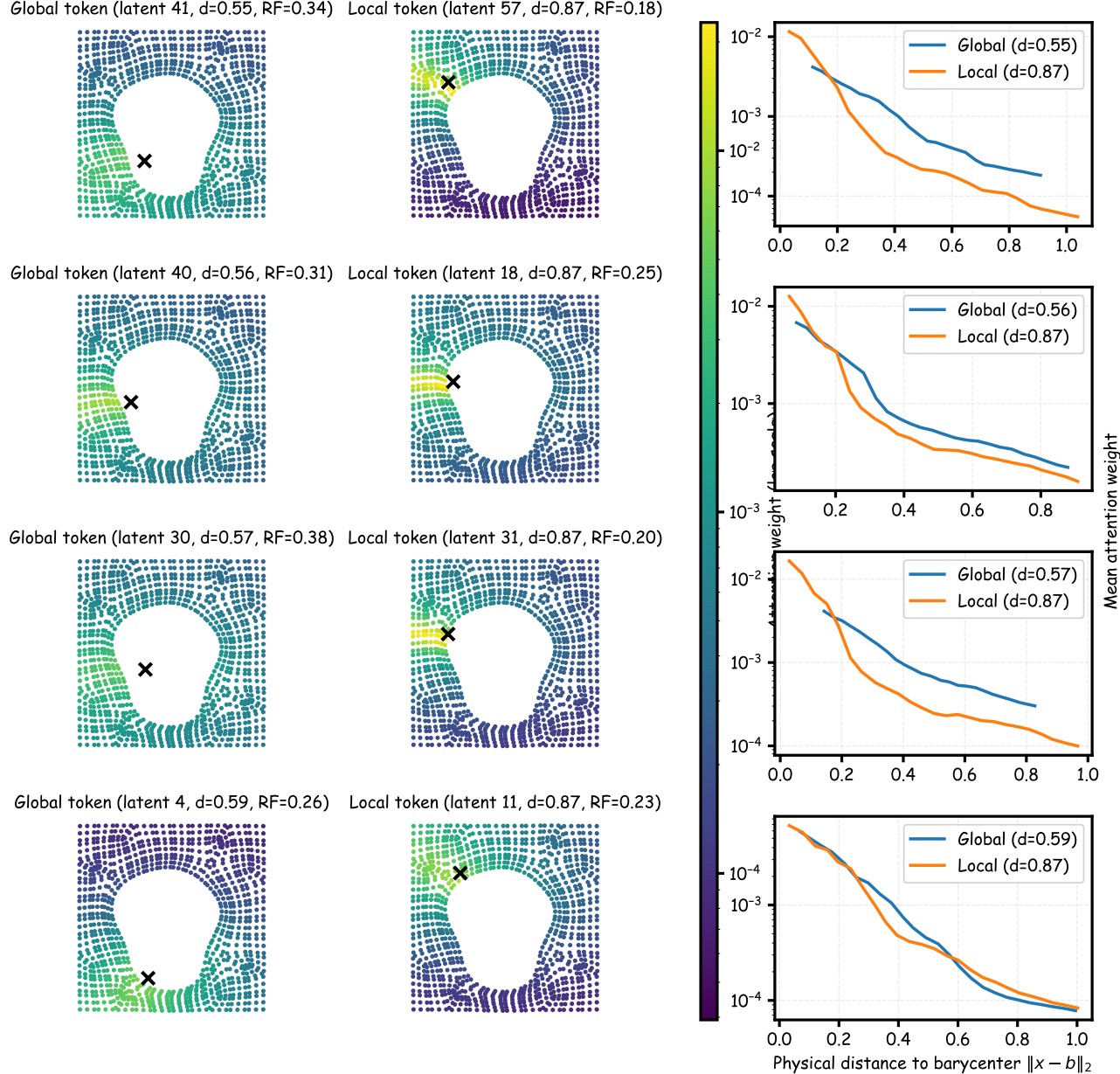

*Figure 25.* Additional attention-map examples on Elasticity, page 2.

## Elasticity attention maps: additional global vs local examples

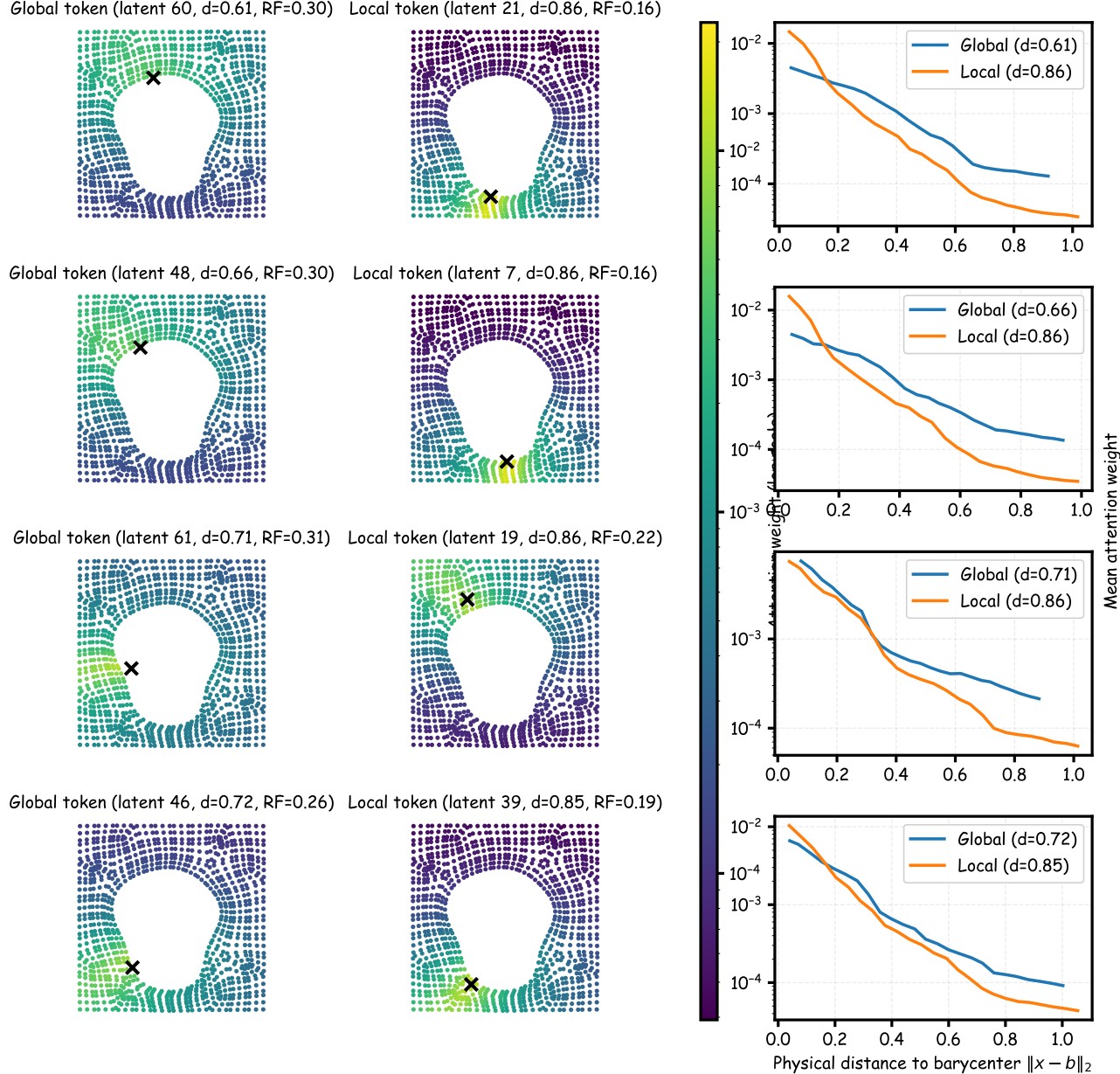

*Figure 26.* Additional attention-map examples on Elasticity, page 3.

**Elasticity attention maps: additional global vs local examples**

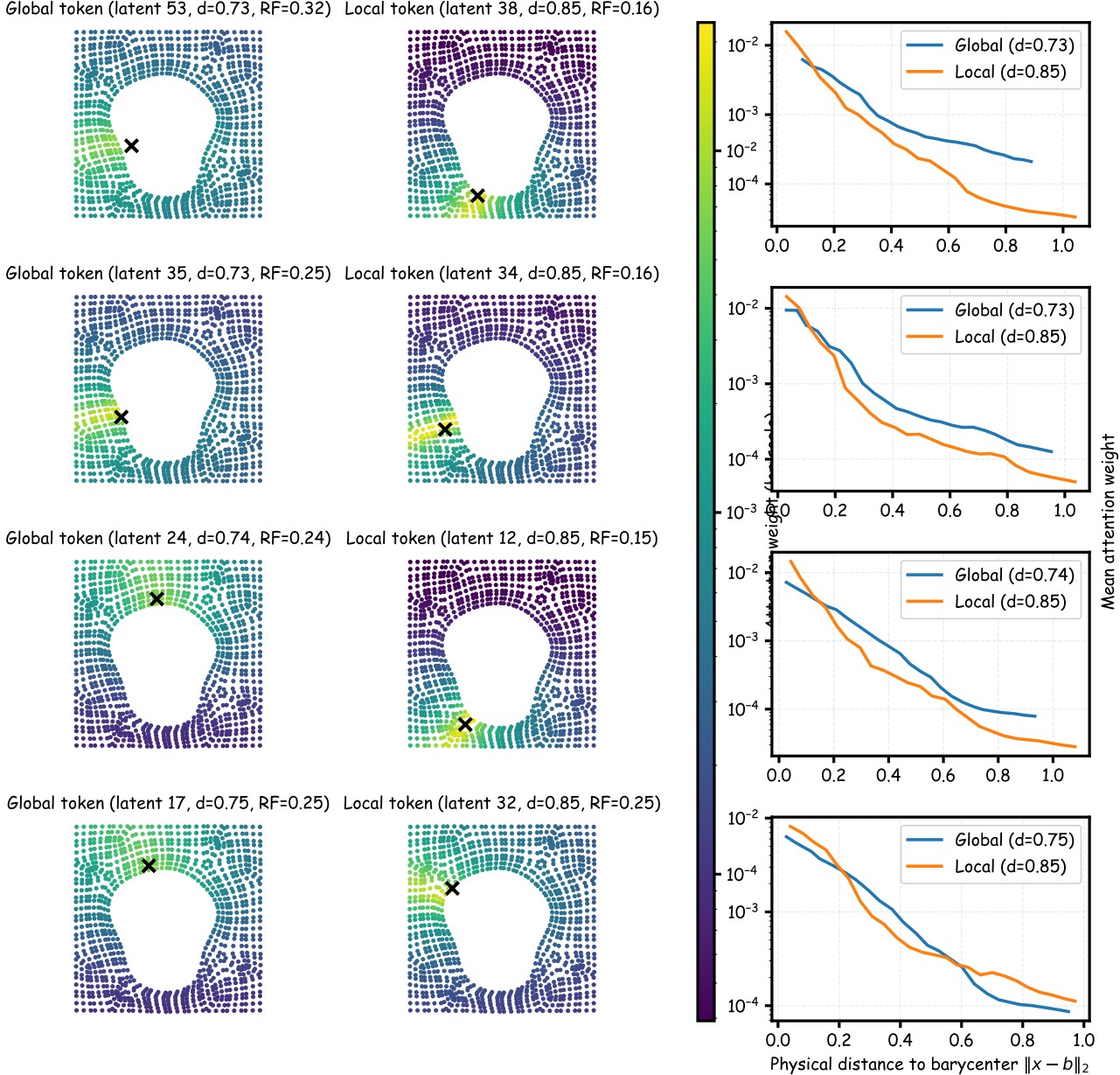

*Figure 27.* Additional attention-map examples on Elasticity, page 4.

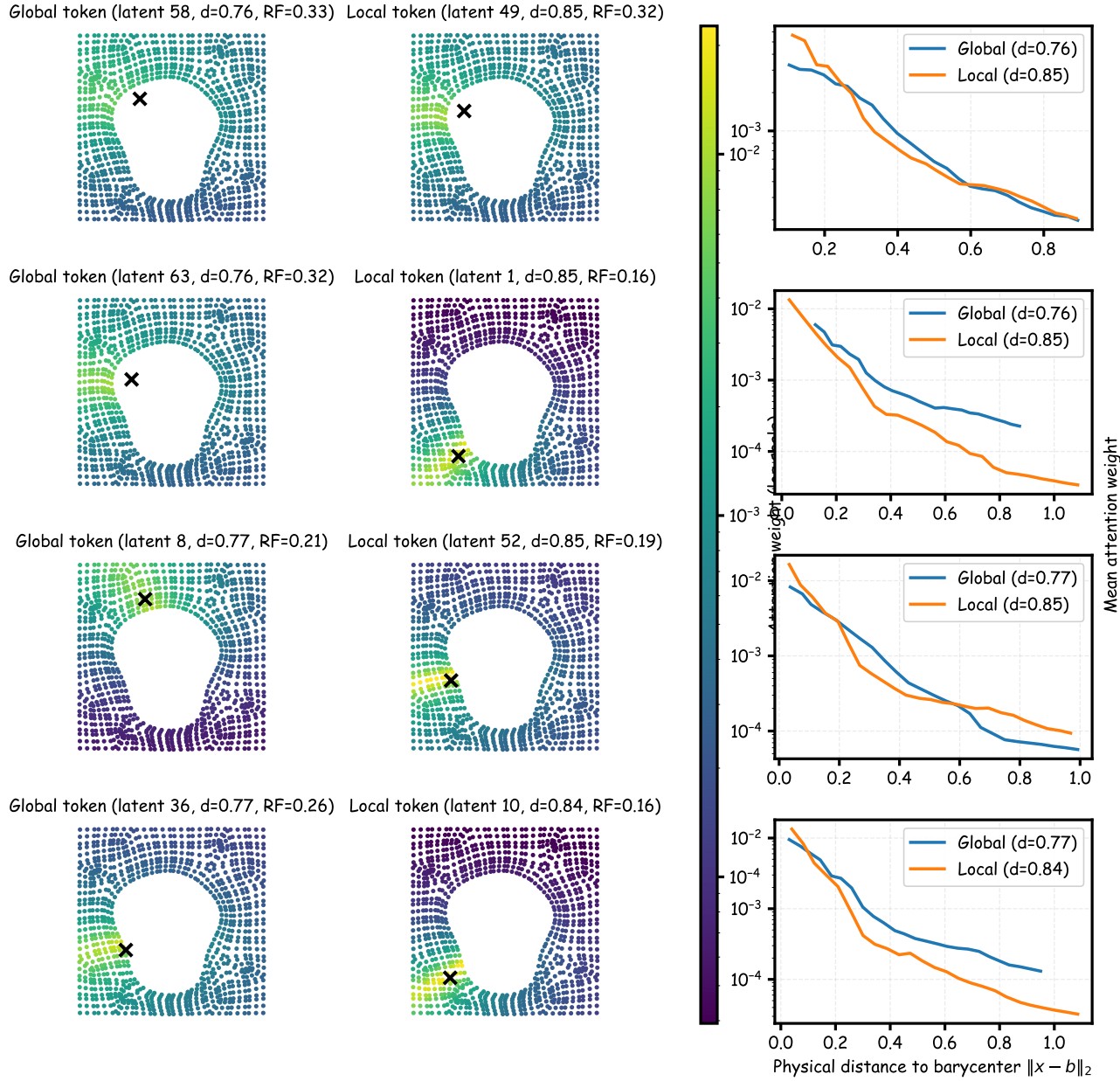

*Figure 28.* Additional attention-map examples on Elasticity, page 5.

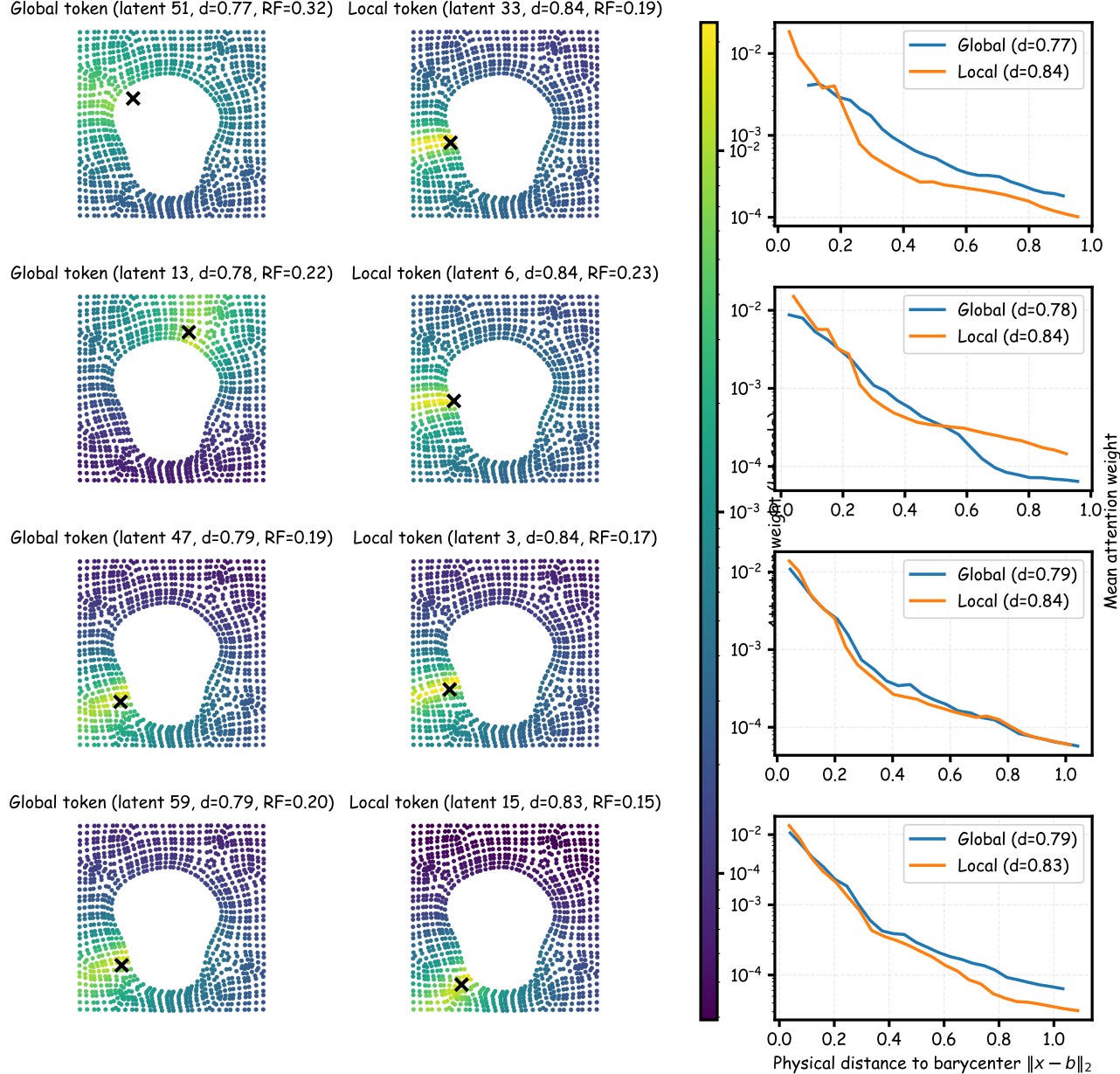

*Figure 29.* Additional attention-map examples on Elasticity, page 6.

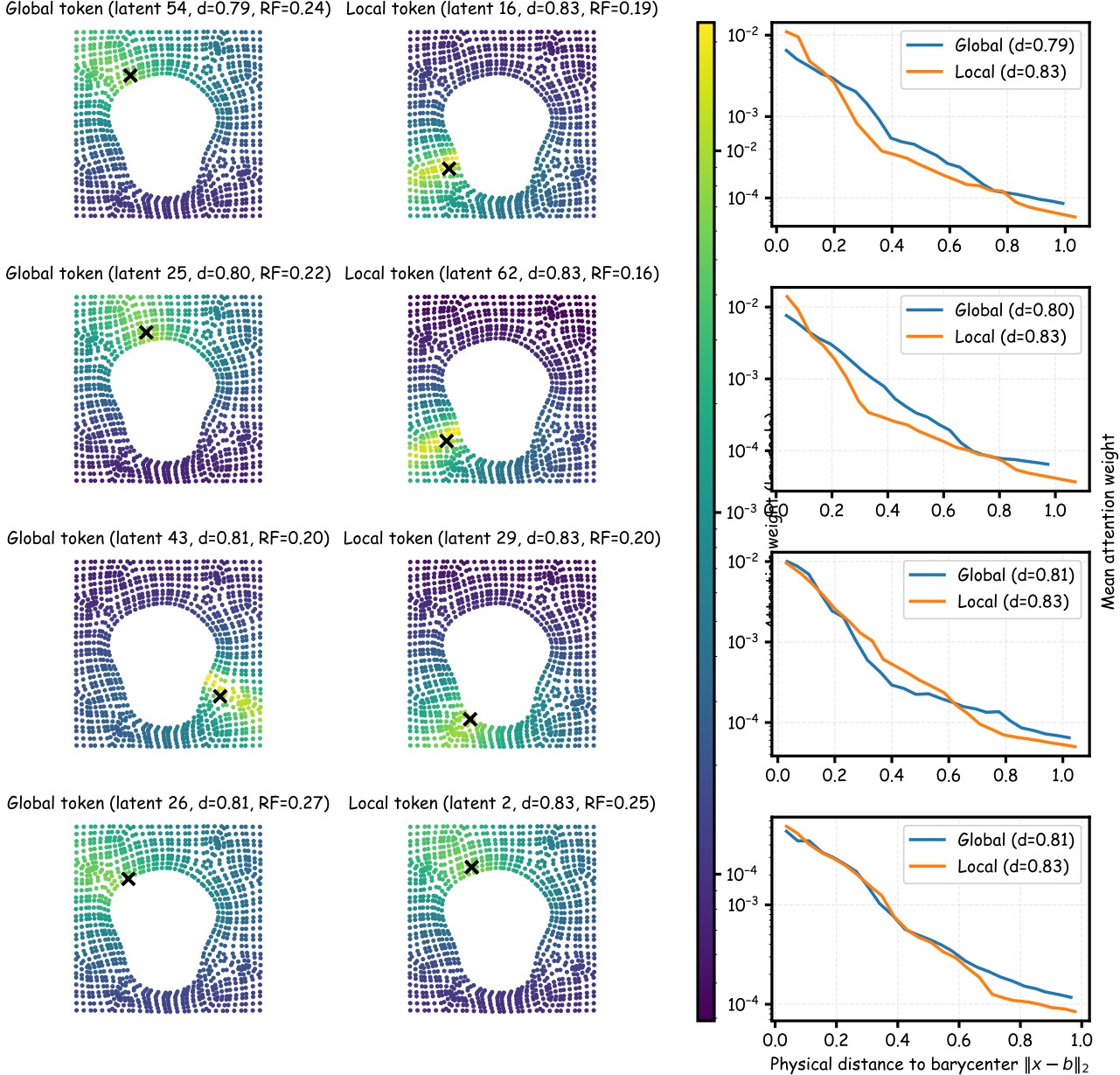

*Figure 30.* Additional attention-map examples on Elasticity, page 7.

**Elasticity attention maps: additional global vs local examples**

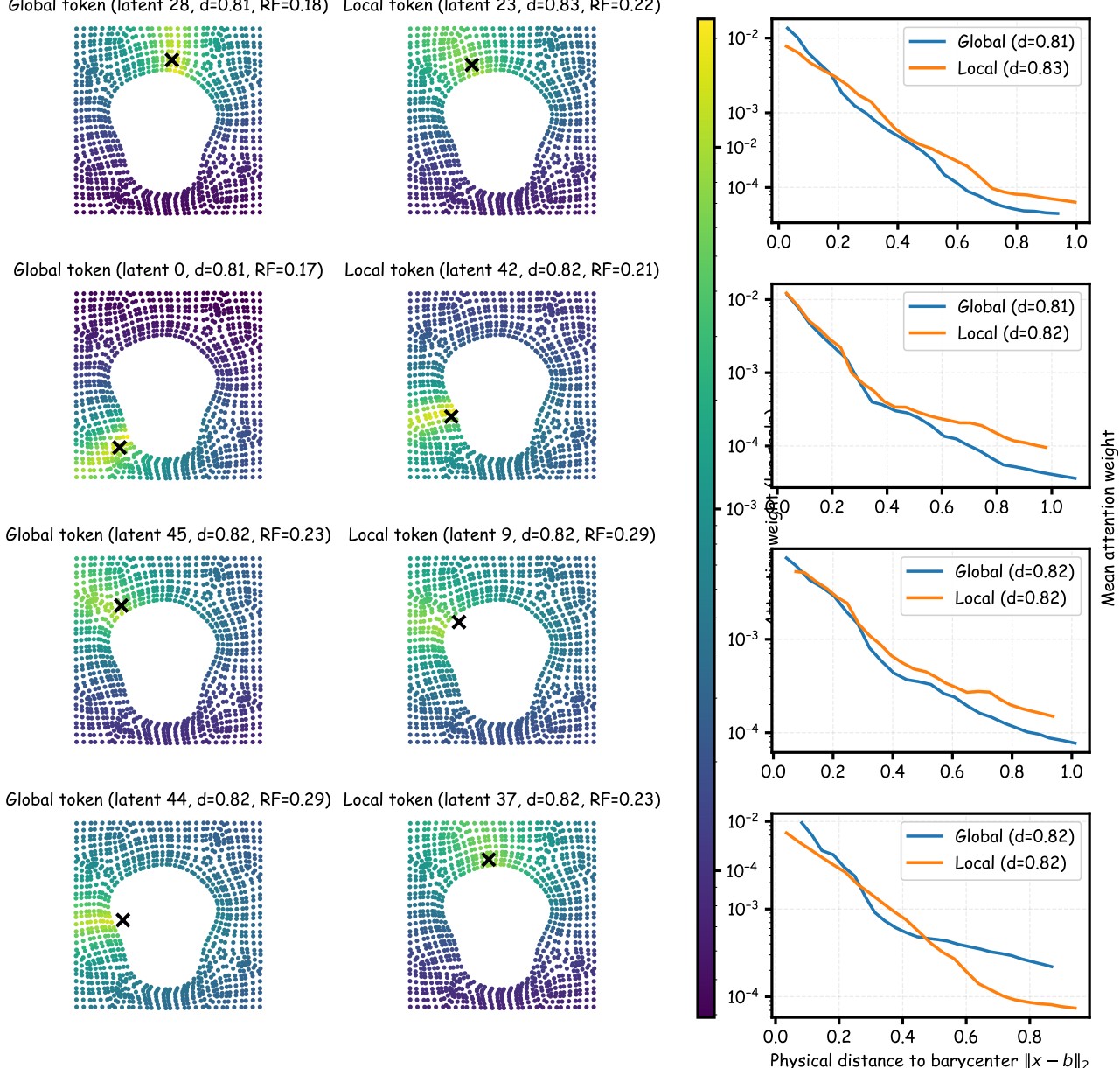

*Figure 31.* Additional attention-map examples on Elasticity, page 8.

### K.3. Additional Mechanism Verification Statistics

We report robust correlations for Fig. 9. **Definitions.** For a latent query token $i$ with attention weights $\{\alpha_{ij}\}_{j=1}^{N}$ over input nodes at physical coordinates $\{x_j\}_{j=1}^{N}$, we define the normalized entropy as $H_i/\log N$, where $H_i := -\sum_{j=1}^{N}\alpha_{ij}\log(\alpha_{ij})$. We define the barycenter $b_i := \sum_{j=1}^{N}\alpha_{ij}x_j$ and the span (receptive field) $\mathrm{RF}_i := \sum_{j=1}^{N}\alpha_{ij}\|x_j - b_i\|_2$, where $\|\cdot\|_2$ is the Euclidean distance in the physical domain. For each test sample, we compute Spearman correlations between radius $r$ and either entropy $H$ or physical span RF, flattening over all heads and latent tokens. We also compute head-wise Spearman correlations within each head (over latent tokens) and report the head-wise median $\mathrm{med}_h\,\rho_h$ per sample. Tab. 17 reports mean±std over 100 test samples, with 95% bootstrap confidence intervals for the mean.

*Table 17.* Robust correlations for mechanism verification on Elasticity. Values are mean±std over 100 test samples; bracketed intervals are 95% bootstrap confidence intervals for the mean.

| SCORE | $\rho(r, H)$ | $\rho(r, \mathrm{RF})$ | $\mathrm{med}_h\,\rho_h$ $(r, H)$ | $\mathrm{med}_h\,\rho_h$ $(r, \mathrm{RF})$ |
|---|---|---|---|---|
| HYPERBOLIC | $-0.614 \pm 0.005$ | $-0.564 \pm 0.013$ | $-0.784 \pm 0.018$ | $-0.770 \pm 0.038$ |
| | $[-0.614, -0.613]$ | $[-0.567, -0.562]$ | $[-0.787, -0.780]$ | $[-0.777, -0.762]$ |
| DOT-PRODUCT | $-0.191 \pm 0.004$ | $-0.073 \pm 0.018$ | $-0.373 \pm 0.009$ | $-0.350 \pm 0.044$ |
| (NORMALIZED) | $[-0.192, -0.191]$ | $[-0.076, -0.069]$ | $[-0.375, -0.371]$ | $[-0.358, -0.341]$ |

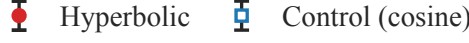

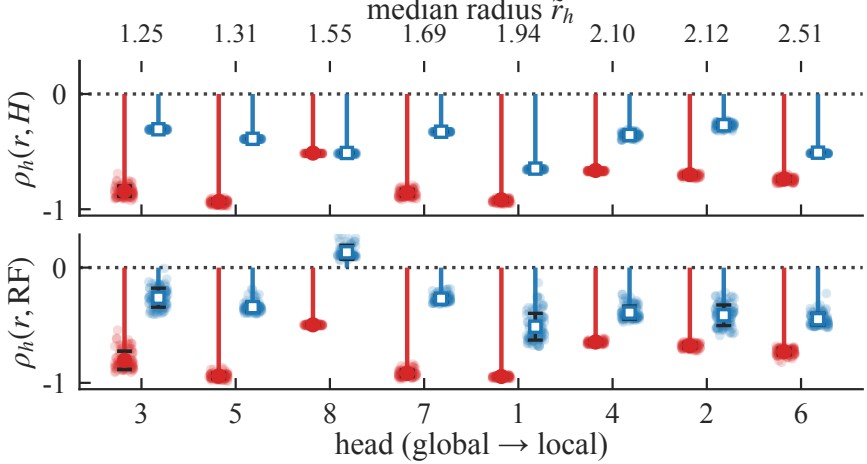

*Figure 32.* Head-wise robustness for Fig. 9. Heads are ordered by median query radius (global→local). Markers show the mean and vertical bars show ±std of per-sample Spearman correlations within each head (over latent tokens); faint points show individual test samples. Hyperbolic attention yields consistently negative correlations, while the cosine-control is weaker and more heterogeneous.

