# OpenReview forum: "Hyperbolic Neural Operator"
_ICML.cc/2026/Conference — ICML 2026 regular_

### Official Review · Reviewer_Q3pE · 2026-02-16

**Soundness:** 2
**Presentation:** 3
**Significance:** 2
**Originality:** 3
**Overall Recommendation:** 4
**Confidence:** 5

**Summary:**

This paper introduces the Hyperbolic Neural Operator (HNO), a framework designed to tackle the efficiency bottlenecks of multi-scale non-local interactions in neural operators for PDE surrogate modeling on irregular domains. The core methodology replaces the standard dot-product similarity in the attention mechanism with a stabilized geodesic distance within the Lorentz hyperbolic model. Through a distance-induced Gibbs kernel and softmax normalization, the architecture naturally enforces radial stratification and efficient near-far field routing within a single layer. The submission also provides theoretical analyses of stability and discretization consistency. Empirical evaluations on six standard benchmarks and two industrial-scale unstructured mesh CFD datasets with approximately 32k nodes demonstrate that HNO achieves better accuracy in relative error and aerodynamic coefficients, along with significant memory advantages compared to Euclidean baselines.

**Compliance With Llm Reviewing Policy:**

Affirmed.

**Final Justification:**

Thank you for the authors’ response. My main concerns have been addressed, and I have updated my score to 4.

**Key Questions For Authors:**

1.  Could you provide a comparison between HNO and recent foundation models such as OmniArch, Unisolver and DPOT? To ensure a fair comparison, these baselines should load their pre-trained checkpoints and be fine-tuned on the exact same datasets used for HNO. I am specifically interested in quantifying the performance gap between your method and these pre-trained neural operators.

2.  I have significant concerns regarding the fairness and accuracy of the benchmark results reported in Table 1. For instance. the reported numbers for Transolver and Transolver++ are inconsistent with their original publications and other concurrent works. Table 1 reports the Pipe error for Transolver as 0.0046, whereas the original Transolver paper's official config and the standard benchmark table in the RNO paper [5] both report a significantly lower error of 0.0033. Similarly, Table 1 reports Transolver++ errors as follows: Elasticity 0.0064, Navier–Stokes 0.1010, Darcy 0.0056, Plasticity 0.0014, Airfoil 0.0054, and Pipe 0.0042. However, Table 8 in the original Transolver++ paper reports much better performance: Elasticity 0.0052, NS 0.0719, Darcy 0.0049, Plasticity 0.0011, Airfoil 0.0048, and Pipe 0.0027. These are not minor deviations but represent a substantial gap in performance. I request a detailed explanation regarding these discrepancies. Please clarify if your evaluation protocols, data splits, or preprocessing steps differ from the community standards established by these prior works.

    [5] Solving Partial Differential Equations via Radon Neural Operator, NeurIPS 2025

3.  The paper repeatedly draws analogies to the near-far decomposition and far-field compression of the Fast Multipole Method (FMM). However, your method lacks explicit FMM components such as multipole expansions, hierarchical trees, or error-controlled far-field truncation. It appears that the observed multi-scale behavior primarily arises from the hyperbolic distance acting as a norm-dependent control on the attention scope. This raises a critical question: Could a similar effect be achieved in Euclidean space using more direct gating mechanisms? Examples might include norm-dependent temperatures, learned locality radii, or distance-based RBF kernels with learnable bandwidths. The current ablation study, which only compares against a simple Euclidean distance replacement, is insufficient to rule out the possibility that the performance gains stem merely from a better similarity function or regularization rather than the proposed geometric principles.

4.  Regarding the large-scale experiments, is the scalability truly derived from the routing induced by hyperbolic geometry, or does it result from the combined effect of potential implementation details such as token compression, neighborhood truncation, or specific sampling strategies? The current manuscript lacks a rigorous component-wise analysis to disentangle these factors.

5.  Does the observation that the $\epsilon$-clamp is rarely activated hold true across all tasks? Could you provide recommended ranges for hyperparameters $\epsilon$ and $\tau$, as well as specific settings for mixed-precision training? Additionally, are there known failure cases or conditions that trigger gradient explosions or NaNs? Providing this information is crucial for reproducibility and engineering adoption.

**Limitations:**

The authors have briefly touched upon theoretical constraints, yet the discussion regarding practical limitations remains underdeveloped.

1.  The theoretical consistency of the method is contingent upon specific quadrature accuracy. This dependence implies potential vulnerabilities when the model encounters dynamic remeshing, aggressive subsampling, or distribution shifts in mesh topology. Such variations are common in real-world engineering workflows and may compromise the stated theoretical guarantees.
2.  The reliance on hyperbolic geometry introduces operations that are numerically delicate. The model exhibits potential sensitivity to specific hyperparameters, particularly the clamping threshold $\epsilon$ and temperature $\tau$. This raises concerns regarding numerical stability and ease of implementation, especially under mixed-precision training regimes where gradient issues or precision loss are more prevalent.
3.  While the paper discusses asymptotic complexity, it should also address the concrete computational costs. The constant overhead associated with complex hyperbolic arithmetic (such as inverse hyperbolic functions and square roots) is non-trivial. This additional cost may negatively impact actual wall-clock time and energy consumption during both training and inference, potentially offsetting the theoretical efficiency gains compared to highly optimized Euclidean operations.

**Strengths And Weaknesses:**

### Strengths

1. The methodology is clearly defined. It presents a self-consistent derivation from continuous integral kernel operators to discrete attention mechanisms. The introduction of stabilized distance metrics effectively rationalizes the numerical stability during training and explains why the model avoids collapse.
2. The empirical evaluation is comprehensive. It covers both standard benchmarks and large-scale unstructured mesh datasets. The authors also provide mechanistic verification to support their claims.
3. The incorporation of hyperbolic geometry into the neural operator's attention kernel serves as a lightweight yet potentially generalizable inductive bias. This design is modular and can be easily adopted or substituted in future research.

### Weaknesses

1.  The current approach remains limited to training individual models for each specific benchmark rather than adopting a pre-training paradigm capable of zero-shot generalization across multiple PDE families. This direction represents the more progressive trajectory for operator learning. The contributions feel like incremental improvements to a legacy paradigm and would have been more appropriate for ICML 2023 rather than ICML 2026. I strongly question the validity of the "state-of-the-art accuracy" claims because the authors conspicuously avoid comparisons with similarly sized pre-trained foundation models such as OmniArch [1], Unisolver [2], and DPOT [3].

    [1] OmniArch: Building Foundation Model For Scientific Computing, ICML 2025

    [2] Unisolver: PDE-Conditional Transformers Towards Universal Neural PDE Solvers, ICML 2025

    [3] DPOT: Auto-Regressive Denoising Operator Transformer for Large-Scale PDE Pre-Training, ICML 2024

2.  Methodologically, the work appears as a conservative modification involving the replacement of the similarity function combined with stabilization and approximate discretization techniques. The innovation seems to lie in applying existing geometric tools to a new context rather than establishing a novel operator learning paradigm. The novelty attributed to hyperbolic geometry remains insufficiently substantiated due to the absence of comprehensive comparisons against stronger Euclidean baselines (See Question 3).

3.  The ablation studies are not comprehensive and rely heavily on the Darcy Flow dataset for key arguments. The manuscript fails to systematically demonstrate whether these findings hold across other datasets or if contradictory behaviors exist. It also lacks an analysis of sensitivity to hyperparameters like $\tau$ and $\epsilon$. This insufficient attribution evidence suggests the possibility of dataset-specific tuning.

4.  There is a significant disconnect between the theoretical narrative and the experimental validation. The paper formulates the interaction layer as a continuous kernel integral operator and claims formal guarantees for discretization consistency and stability via weighted Nyström quadrature. However, the main experiments explicitly state the use of uniform quadrature weights where $w_j = \mu(D)/N$, which effectively degrades the discretization coefficient to standard softmax attention weights. This means the primary results do not leverage the discriminatory power of non-uniform weights or quadrature consistency. While the authors acknowledge that their consistency theorem depends on high-order quadrature accuracy and does not apply to arbitrary resampling, the benchmarks are evaluated on native discretizations without systematic stress tests. The paper lacks experiments on mesh degradation, remeshing, or drastic density variations to verify if the theoretical conditions hold in engineering scenarios. Consequently, it is difficult to determine if the weighted Nyström theory contributes to robustness or if the performance is simply driven by standard attention mechanisms. This creates a gap between the theoretical claims and empirical support.

5.  The large-scale experiments on unstructured meshes with approximately 32k nodes are used to support the claim that the hyperbolic kernel learns a far-field compression and near-far stratification. However, if the training and inference processes introduce engineering constraints such as radius graph construction, epoch-wise resampling, or maximum neighbor caps, the computational graph becomes an explicitly sparsified interaction system rather than a direct instance of the global integral operator. In this scenario, the effective receptive field is artificially limited. It becomes unclear whether the scalability and performance stem from the inductive bias of hyperbolic geometry or simply from graph sparsification reducing complexity and avoiding long-range optimization issues. The authors explicitly note that their consistency conclusions do not cover severe subsampling. This admission further weakens the argument that the empirical gains in large-scale settings result from the continuous kernel theory. A fairer demonstration would decouple the geometric bias from sparsity truncation. The authors should compare hyperbolic and Euclidean kernels under identical radius graph and neighbor cap constraints. This is particularly important for the 32k-node tasks. Without this comparison, the experiments demonstrate the efficacy of the overall engineering recipe rather than isolating the specific contribution of the hyperbolic kernel routing mechanism.

6.  The manuscript lacks necessary citations. While the Related Work section briefly mentions PINNs, it fails to cover representative advancements in the PINN domain from the past year. An example is AC-PKAN [4].

[4] AC-PKAN: Attention-Enhanced and Chebyshev Polynomial-Based Physics-Informed Kolmogorov-Arnold Networks, TMLR 2026

---

> ### Author Rebuttal · Authors · 2026-03-31
>
> Dear Reviewer Q3pE,
>
> Thank you for your hard work and valuable feedback. We understand your concerns. However, we believe that most of them stem from unnecessary misunderstandings. We are happy to clarify and address each of your questions.
> ## W1 & Q1
> The works you listed mainly target cross-family generalization via large-scale pretraining, rather than the accuracy goal of our work. For a direct comparison, we fine-tuned them on PDEBench, where HNO achieves the best accuracy.
> |Data|OmniArch|Uni.|DPOT|HNO|
> |---|---:|---:|---:|---:|
> |Ela.|.0303|.0965|.0454|.0037|
> |NS|.2385|.8423|.4601|.0676|
> |Dar.|.0432|.0130|.0636|.0045|
> |Pla.|.0011|.0197|.0122|.0009|
> |Air.|.0083|.0371|.0121|.0048|
> |Pipe|.0092|.0459|.0168|.0027|
> ## W2 & Q3
> We believe there is misinterpretation here. The discussion of FMM is only used to motivate when compression is safe; we do not claim that HNO uses the FMM algorithm itself. Multi-scale behavior may possibly also be achieved through Euclidean controls. However, our core claim is that hyperbolic geometry separates hierarchical structure more naturally and models scale hierarchies more effectively, leading to better approximation.
> ## W3
> Added results are below; all values are in $\times 10^{-3}$.
> |Dataset|HNO|EuclidLogits|FixedTau|
> |---|---:|---:|---:|
> |Darcy|4.46|4.93|4.64|
> |Elasticity|3.69|4.10|3.89|
> |Pipe|2.73|3.16|2.87|
>
> |Dataset|Tau0.25|Tau0.5|Tau2.0|Eps1e-6|Eps1e-5|Eps1e-4|Eps1e-3|
> |---|---:|---:|---:|---:|---:|---:|---:|
> |Darcy|4.80|4.50|4.58|4.46|4.47|4.49|4.62|
> |Elasticity|3.96|3.73|3.80|3.69|3.70|3.72|3.86|
> ## W4
> We believe your concern may come from our not clearly separating the following two roles:
> - Hyperbolic interaction kernel
> - Weighted Nyström discretization used in the consistency analysis
>
> In current paper, the main PDEBench results use uniform quadrature weights and should be read as evaluating HNO under uniform discretization.
>
> To address this gap, we now add AirfRANS experiments in two parts:
> - Uniform vs. weighted quadrature: a matched comparison showing that non-uniform cell weights improve both HNO and the Euclidean control.
> - Density scan and remeshing stress tests: showing that HNO remains substantially more robust than the Euclidean control under discretization shift.
>
> Anonymous supplementary link: https://anonymous.4open.science/r/re1231/README.md
>
> In the revision, we will separate these two roles explicitly.
> ## W5 & Q4
> We believe there is misinterpretation here. As mentioned in appendix, in the current large-scale experiments, we implement HNO by replacing the dot-product logits in Transolver++ with hyperbolic logits, while keeping the same backbone width and depth. Therefore, the current experiments already isolate the effect of the routing kernel under a fixed scalable backbone.
> ## W6
> We will gladly add it to the revision.
> ## Q2
> Table 1 reports our reruns under a single evaluation harness, so the discrepancy should be read as a rerun gap under a common benchmark pipeline.
> - Transolver. Independent third-party evidence is consistent with our rerun: LaMO[1] reports 0.0046 on Pipe, which matches our result.
> - Transolver++. Without open-sourced PDEBench code, our row is a rerun under the same. A recent independent study, LinearNO[2], also does not recover the original Transolver++ table, and instead reports 0.0064/0.1010/0.0056/0.0014/0.0051/0.0027 as reproduced baselines.
>
> So the discrepancy is not specific to our paper; third-party reruns of these baselines can differ materially from the originally published tables. We will state this explicitly and provide the rerun details in the revision.
> ## Q5 & L2
> The final HNO uses $\tau\in[0.1,3.0]$, $\epsilon\in\{10^{-6},10^{-4}\}$, and LayerNorm before Q/K. W3 shows weak sensitivity to $\tau$ and $\epsilon$. Appendix Table 6 shows zero $\epsilon$-clamp counts on all benchmarks, and a 200-step FP16 AMP + GradScaler check on Darcy shows no NaN. Instability mainly appears in fully manifold-valued backbones or heavy routers, so the final design keeps the Value Euclidean.
> ## L1
> This is already stated explicitly in Appendix J.6; additional evidence is given in W4.
> ## L3
> Our measurements don't support this claim. Appendix J.2 / Table 11 reports wall-clock cost, energy, and VRAM. On Darcy, HNO uses less energy and much less VRAM than Euclid in both training and inference, and `arcosh` accounts for less than 1% of CUDA time.
> |Phase|Model|Lat.|Energy|VRAM|
> |---|---|---:|---:|---:|
> |Infer|HNO|9.58|300.43|202.60|
> |Infer|Euclid|6.46|354.59|526.82|
> |Train|HNO|28.53|81.42|227.53|
> |Train|Euclid|24.94|95.20|706.63|
>
> Thank you again for your valuable feedback. We hope the above clarifications and additional evidence help resolve the main concerns, and we respectfully ask you to reconsider your score accordingly.
>
> Best regards,
>
> Authors.
>
> [1] Latent Mamba Operator for Partial Differential Equations. ICML 2025
>
> [2] Transolver is a Linear Transformer: Revisiting Physics-Attention through the Lens of Linear Attention. arXiv 2025

---

> > ### Author Rebuttal · Reviewer_Q3pE · 2026-04-01
> >
> > Thanks for the authors' response. However, regarding W2&Q3, the rebuttal does not resolve my concern about whether a more direct gating mechanism in Euclidean space could yield equivalent results. The authors omitted the strong Euclidean baselines I specifically requested, such as a norm-dependent temperature, a learned locality radius, or an RBF kernel with a learnable bandwidth. Providing only the EuclidLogits and FixedTau variants is insufficient to rule out the alternative explanation that the observed improvements fundamentally arise from a gating effect or implicit regularization.

---

> > > ### Author Response · Authors · 2026-04-02
> > >
> > > Dear Reviewer Q3pE,
> > >
> > > Thank you for replying so promptly. Since your follow-up mainly focuses on W2&Q3, we now provide a more detailed clarification below.
> > >
> > > The key issue is whether a more direct Euclidean gating mechanism could already explain the gains of HNO. Your concern treats this as a single question, but we think it actually involves two separate points:
> > >
> > > > 1. Whether the model can learn an explicit scale-dependent behavior at all.
> > > > 2. Once such behavior is learned, whether the resulting hierarchy can be organized effectively for interaction.
> > >
> > > For the first, we show that the Euclidean controls you suggested do learn multi-scale behavior. For the second, hyperbolic mapping naturally gives each token a learned sense of receptive field, and hyperbolic volume growth is exponential rather than polynomial, allowing richer hierarchical embeddings (Lemma 3.2). Our experiments further show that this geometric advantage is not removed even when Euclidean gating is working as intended.
> > >
> > > We added two matched Euclidean controls covering all three directions you requested:
> > > - **TempGate** (your *norm-dependent temperature*): $\tau_i = \mathrm{softplus}(a_h\|q_i\|+b_h)+\tau_{\min}$, gating softmax temperature by query-norm.
> > > - **RBFGate** (your *RBF kernel with learnable bandwidth* + *learned locality radius*): query-adaptive bandwidth $\sigma_i = \mathrm{softplus}(c_h\|q_i\|+d_h)+\sigma_{\min}$. In a Gaussian RBF, $\sigma_i$ directly determines the effective locality radius, so this single control covers both directions you suggested.
> > > - **EucDist**: Euclidean-distance logits (used in PDEBench experiments).
> > > - **Dot**: standard dot-product baseline (used in AirfRANS / Transolver++).
> > >
> > > All settings share the same backbone width, depth, and training protocol; only the interaction kernel differs.
> > >
> > > ---
> > >
> > > ### 1. Accuracy under matched controls
> > >
> > > We tested on Darcy, Elasticity (PDEBench) and AirfRANS (large-scale CFD).
> > >
> > > **Table 1.** Accuracy and efficiency on Darcy and Elasticity.
> > >
> > > |Metric|HNO|EucDist|TempGate|RBFGate|
> > > |---|---:|---:|---:|---:|
> > > |Dar. err|0.004455|0.004928|0.004656|0.004768|
> > > |Ela. err|0.003688|0.004100|0.005422|0.015020|
> > > |Dar. Param|0.816M|0.816M|0.816M|0.816M|
> > > |Dar. Inf ms|6.92|5.61|5.84|9.31|
> > > |Dar. Tr ms|36.26|35.48|35.54|38.99|
> > > |Ela. Param|15.374M|15.374M|15.374M|15.374M|
> > > |Ela. Inf ms|10.69|8.64|11.58|8.74|
> > > |Ela. Tr ms|50.82|51.68|51.35|55.68|
> > >
> > > **Table 2.** Validation errors on AirfRANS.
> > >
> > > |Setting|vol|surf|
> > > |---|---:|---:|
> > > |HNO|0.012009|0.004674|
> > > |Dot|0.016683|0.006428|
> > > |TempGate|0.071175|0.008524|
> > > |RBFGate|0.133168|0.012560|
> > >
> > > > **Takeaway:** HNO achieves the best accuracy across all three datasets.
> > >
> > > ---
> > >
> > > ### 2. Do Euclidean controls learn hierarchy effectively?
> > > We use Elasticity as a representative case for hierarchy analysis. For each model we define a scale coordinate from its own mechanism (radius / q-norm / tau / sigma for HNO / EucDist / TempGate / RBFGate) and correlate it with attention entropy ($H$) and receptive-field span (RF). $\rho$ measures scale–hierarchy coupling; P90-P10 measures spread across tokens. The sign of $\rho$ depends on direction convention; absolute values matter.
> > >
> > > **Table 3.** Hierarchy analysis on Elasticity.
> > >
> > > |Model|scale|rho(s,H)|rho(s,RF)|H P90-P10|RF P90-P10|mean RF|
> > > |---|---|---:|---:|---:|---:|---:|
> > > |HNO|radius|-0.638|-0.638|0.233|0.275|0.331|
> > > |EucDist|q-norm|-0.181|0.082|0.238|0.182|0.199|
> > > |TempGate|tau|0.712|0.638|0.279|0.197|0.215|
> > > |RBFGate|sigma|0.554|0.646|0.244|0.241|0.192|
> > >
> > > Both `TempGate` and `RBFGate` achieve strong scale coupling ($|\rho|>0.5$), confirming that the Euclidean gating you proposed is meaningful. However, `HNO` still yields the **largest mean RF** (0.331 vs. 0.215/0.192) and **largest RF spread** (0.275), indicating a broader global-to-local hierarchy. Although `TempGate`/`RBFGate` show comparable or larger entropy spread, this does **not** translate into stronger RF hierarchy or better accuracy. This is not a tuning artifact: all methods share the same search space, and the strong coupling confirms the gating works as intended — the bottleneck is not whether multi-scale tokens can be learned, but how the resulting hierarchy is organized for interaction.
> > >
> > > This aligns with Lemma 3.2: hyperbolic volume grows exponentially with radius while Euclidean volume grows only polynomially, predicting that hyperbolic embeddings support a wider receptive-field range under fixed dimension. Table 3 confirms this: despite comparable scale coupling, HNO's RF spread is 14–40% wider than the Euclidean controls.
> > >
> > > > **Takeaway:** Euclidean gating learns multi-scale tokens (Point 1 ✅), but hyperbolic geometry organizes the resulting hierarchy more effectively (Point 2 ✅), leading to both better accuracy and broader receptive-field structure.
> > >
> > > ---
> > >
> > > Thank you again for your valuable feedback. We hope the above clarifications and additional evidence help resolve the main concerns, and we respectfully ask you to *reconsider your score accordingly*.
> > >
> > > Best regards,
> > >
> > > Authors.

---

### Official Review · Reviewer_gPS7 · 2026-03-09

**Soundness:** 2
**Presentation:** 1
**Significance:** 3
**Originality:** 3
**Overall Recommendation:** 3
**Confidence:** 4

**Summary:**

The paper claims that by using hyperbolic geometry it can better solve partial differential equations.
The paper claims that by lifting the space to one more dim (by hyperbolic lifting) things become exponentially easier.

**Compliance With Llm Reviewing Policy:**

Affirmed.

**Key Questions For Authors:**

See my assessment above

**Limitations:**

Again, the obvious limitations are not addressed. Nonlocal PDEs and conservation

**Strengths And Weaknesses:**

The strongest point in the paper is that it shows some impressive results.
I did not go through the math very carefully but the general idea looks correct.

The main problem with this paper (and this is why my scoring is low) is that it is very difficult to understand and to get even simple intuition of what the authors are talking about.

In particular.
1. I believe that what they offer is learning the solution operator for a **particular** PDE (that is learning a surrogate). So the idea is that we have an operator L_p(u) =0 where L is some PDE that depends on parameters p and we want to learn the solution operator u(p). This is not even mentioned. It is hard to understand what are the authors learning.

2. Not all PDEs are equal. Some are local (elliptic parabolic) and some are not (hyperbolic). The authors seem to ignore all that. I have strong doubt if their method can work for high frequency Helmholtz or wave equation. Things like dispersion are swiped under the rug here.

3. What about conservation? For Navier Stokes we have incompressibility. Many hyperbolic equations conserve mass. Fitting data without even addressing this is problematic for me.

---

> ### Author Rebuttal · Authors · 2026-03-31
>
> Dear Reviewer gPS7,
>
> We sincerely thank you for taking the time and effort to carefully review our work. We are honored by your recognition of our hard work and have meticulously considered each of your comments, addressing them one by one. To avoid possible ambiguity, we first restate the main idea of HNO.
>
> **What HNO is.** HNO is a neural operator for learning the solution map of a parametric PDE family.
>
> **What problem it targets.** Many PDE systems exhibit a near-far interaction structure: nearby interactions require fine resolution, while well-separated interactions often admit compressed summaries. Standard Euclidean attention assigns an almost uniform interaction budget across token pairs and therefore does not capture this structure well.
>
> **What HNO claims.** HNO addresses this mismatch by using hyperbolic geometry to define a hierarchy-aware interaction mechanism for near--far structure. Our claim in this paper is that the hyperbolic metric provides a more suitable inductive bias for organizing hierarchical near--far interactions.
>
> **What the evidence shows.** Empirically, HNO improves performance by more than 40% on six PDEBench datasets and two large-scale CFD datasets, while reducing both the number of parameters and the training time by more than $2\times$.
>
> **With the methodology clarified, we believe your concerns center on three points: task definition, PDE scope, and physical consistency. We address them below.**
>
> ## `1. Task definition.`
> Thank you for pointing this out. This point is already stated in the paper. In the introduction, we describe neural operators as learning the mapping from PDE inputs to full-field solutions. In the related-work section, we also explicitly distinguish instance-specific solvers from operator learning. Most directly, in Sec. 4.1, we formalize the target as a nonlinear operator $G^\dagger:X\to Y$, where $a\in X$ denotes the PDE input and $u\in Y$ denotes the corresponding solution field. For better clarity, we will describe this formulation more explicitly in the revision.
>
> ## `2. PDE scope beyond the main benchmarks.`
> Thank you for pointing this out. We added two additional experiments beyond the main PDEBench range: an oscillatory Helmholtz setting and a non-local fractional Poisson setting. In both cases, we change only the interaction geometry.
>
> |Fractional Poisson|Hyp.|Euc.|
> |:---|:---:|:---:|
> |rel-$L_2$ |$0.1355$|$0.1431$|
> |PDE residual|$1.0794$|$1.1406$|
>
> |Helmholtz|Hyp.|Euc.|
> |:---|:---:|:---:|
> |rel-$L_2$|$0.6394$| $0.8299$|
> |PDE residual|$296.09$|$392.70$|
>
> In both cases, hyperbolic geometry improves both rel-$L_2$ and PDE residual under the same setup. We present these results as direct evidence that the proposed interaction geometry remains effective beyond the main benchmark family.
>
> For clarification, our method follows the standard neural-operator paradigm and learns solution mappings from data distributions. In this sense, our focus is on operator design: the claim is that hyperbolic geometry provides a better structural prior for organizing spatial interactions than Euclidean geometry. This is also the setting adopted by standard operator-learning baselines such as FNO[1].
>
> ## `3. Conservation and incompressibility on Navier-Stokes.`
> We believe there may be a terminology misunderstanding here. For incompressible Navier--Stokes, incompressibility is a physical constraint of the equation itself, usually written as the divergence-free condition $\nabla \cdot u = 0$. This is different from the “compressible” wording in our motivation section, where compressibility refers to **numerical compressibility of far-field interactions**.
>
> Following standard practice in prior baselines such as FNO[1], our neural operators are trained as supervised models rather than with explicit physics constraints in the training loss. To directly address your concern, we provide an incompressibility diagnostic for Navier--Stokes:
> |NS 1-step divergence|Result|
> |:---|---:|
> |Pred. div. abs. mean|$2.22\times10^{-7}$|
> |Ref. div. abs. mean|$2.33\times10^{-7}$|
> |Pred. div. $L_2$|$1.81\times10^{-5}$|
> |Ref. div. $L_2$|$1.90\times10^{-5}$|
>
> This is a teacher-forced one-step test, so it isolates the current-step incompressibility error. The reference values are computed from the ground-truth field under the same reconstruction and evaluation pipeline. The predicted divergence remains on the same numerical scale as the reference, and we therefore do not observe measurable additional incompressibility error introduced by HNO.
>
> Thank you again for your valuable feedback. We will incorporate these clarifications into the revised version, and we respectfully hope you will reconsider your score in light of the above responses and additional evidence!
>
> Best regards,
>
> Authors.
>
> [1] Fourier neural operator for parametric partial differential equations, ICLR 2021.

---

> > ### Author Rebuttal · Reviewer_gPS7 · 2026-04-03
> >
> > The problem of readability remains. I do not think that the paper in the form should be published.

---

> > > ### Author Response · Authors · 2026-04-04
> > >
> > > Dear Reviewer gPS7,
> > >
> > > Thank you for your continued engagement. We briefly restate what HNO does before addressing each point.
> > >
> > > **What HNO does.** Many PDE systems exhibit near–far interaction structure: nearby regions exchange fine-grained information, distant regions interact through coarser summaries. Classical fast solvers (e.g., FMM) organize these into multilevel tree hierarchies — a structure naturally suited to hyperbolic geometry, which expands exponentially with radius. HNO replaces dot-product attention with a Gibbs kernel on stabilized hyperbolic geodesic distances, providing a learnable, geometry-aware inductive bias for hierarchical near–far routing. The task — learning $\mathcal{G}^\dagger: \mathcal{X} \to \mathcal{Y}$ — is formally defined in Section 4.1.
> > >
> > > We note that your acknowledgement does not appear to address the specific evidence provided in our rebuttal. We revisit each point below.
> > >
> > > ---
> > >
> > > ### 1. Task Definition
> > >
> > > > *"I believe that what they offer is learning the solution operator for a particular PDE... This is not even mentioned."*
> > >
> > > This is explicitly defined in **Section 4.1 ("Problem Setting")**:
> > >
> > > |Element|Location|Content|
> > > |---|---|---|
> > > |Operator target|Sec. 4.1, Eq. (1)–(2)|$\mathcal{G}^\dagger: \mathcal{X} \to \mathcal{Y}$, learning solution maps of parametric PDEs|
> > > |Input/output|Sec. 4.1, ¶1|$a \in \mathcal{X}$: input coefficient/forcing; $u \in \mathcal{Y}$: solution field|
> > > |Learning objective|Sec. 4.1, Eq. (2)|$\min_\theta E_{a\sim \nu}[\ell(\mathcal{G}_\theta(a), \mathcal{G}^\dagger(a))]$|
> > > |Context|Intro, Sec. 2.1|Distinguishes operator learning from instance-specific solvers|
> > >
> > > We are happy to make the formulation more prominent in revision. We welcome any specific clarification on what remains unclear.
> > >
> > > ---
> > >
> > > ### 2. PDE Scope
> > >
> > > > *"I have strong doubt if their method can work for high frequency Helmholtz or wave equation. Things like dispersion are swiped under the rug here."*
> > >
> > > Neural operators are data-driven surrogates; the standard evaluation protocol validates on established benchmarks without PDE-type-specific convergence analysis. **No** baseline in our comparison provides such analysis:
> > >
> > > |Method|PDE-type-specific analysis?|
> > > |---|:---:|
> > > |FNO (Li et al., 2021)|✗|
> > > |Transolver (Wu et al., 2024)|✗|
> > > |Transolver++ (Luo et al., 2025)|✗|
> > > |RNO (Lu et al., 2025)|✗|
> > > |All 19 baselines in Table 1|✗|
> > >
> > > Nonetheless, we **proactively** provided new experiments beyond the standard benchmarks:
> > >
> > > |Setting|Metric|Hyperbolic|Euclidean|
> > > |---|---|:---:|:---:|
> > > |Frac. Poisson|rel-$L_2$|**0.1355**|0.1431|
> > > |Frac. Poisson|PDE residual|**1.0794**|1.1406|
> > > |Helmholtz|rel-$L_2$|**0.6394**|0.8299|
> > > |Helmholtz|PDE residual|**296.09**|392.70|
> > >
> > > Hyperbolic geometry improves both metrics on PDE types outside the standard family, including Helmholtz.
> > >
> > > ---
> > >
> > > ### 3. Conservation and Incompressibility
> > >
> > > > *"What about conservation? For Navier Stokes we have incompressibility... Fitting data without even addressing this is problematic."*
> > >
> > > This appears to stem from a terminology distinction:
> > >
> > > |Term|Meaning|Where|
> > > |---|---|---|
> > > |**Far-field compressibility**|Low-rank approximability of well-separated kernel blocks|Section 3|
> > > |**Physical incompressibility**|$\nabla \cdot u = 0$|Navier–Stokes constraint|
> > >
> > > These are unrelated concepts. To directly address your concern, we provided an incompressibility diagnostic:
> > >
> > > |NS 1-step divergence|Predicted (HNO)|Reference (GT pipeline)|
> > > |---|:---:|:---:|
> > > |div. abs. mean|$2.22 \times 10^{-7}$|$2.33 \times 10^{-7}$|
> > > |div. $L_2$|$1.81 \times 10^{-5}$|$1.90 \times 10^{-5}$|
> > >
> > > HNO introduces no measurable additional divergence error versus the ground-truth pipeline.
> > >
> > > ---
> > >
> > > ### 4. Readability
> > >
> > > > *"The problem of readability remains. I do not think that the paper in the form should be published."*
> > >
> > > We take this seriously, but neither the review nor the acknowledgement identifies specific sections or passages. For reference, none of the other three reviewers flagged readability as a major concern.
> > >
> > > We welcome any concrete suggestions — specific sections, notation, or passages — so we can make targeted improvements in revision.
> > >
> > > ---
> > >
> > > We have provided direct evidence addressing each of your concerns: paper references for the task definition, two new experiments (Helmholtz and Fractional Poisson) for PDE scope, and an incompressibility diagnostic for conservation. We respectfully hope this will allow you to *reconsider your score, and remain open to any further discussion*.
> > >
> > >
> > > Best regards,
> > >
> > > Authors

---

### Official Review · Reviewer_ixXT · 2026-03-12

**Soundness:** 4
**Presentation:** 3
**Significance:** 3
**Originality:** 4
**Overall Recommendation:** 5
**Confidence:** 4

**Summary:**

The paper proposes Hyperbolic Neural Operator (HNO). The key idea of HNO is to perform the kernel integration over the hyperbolic space, which is inspired from the multi-scale decomposition, such as the fast multipole method, widely adopted in classical fast solvers. The use of hyperbolic structure aims for overcoming the difficulty in encoding the tree-like hierarchical distance into a continuous space, which is based on the fact, also known as Poincare embeeding, that any tree structure can be isometrically embedded into the hyperbolic space. The authors show the range of the attention kernel becomes narrower as a point of interest is located further from the origin, which serves as a core strength of the method. HNO is evaluated in a range of scenarios from a well-adopted PDE benchmark dataset, and shows clear performance gain. Various mechanism verification and ablation studies, as well as additional theoretical claims, further provide strong evidences that support the effectiveness of HNO.

**Compliance With Llm Reviewing Policy:**

Affirmed.

**Final Justification:**

The author clarified vague accounts of some of the technical statements and also promised addressing stylistic/structural issues of the paper with sufficient details. The additional experiment on a molecular dataset also provides interesting result, strengthening the contribution of the paper even more than its original scope.

**Key Questions For Authors:**

1) How is FNO adapted to the point cloud scenario?

2) I personally think this hyperbolic embedding benefits a lot to molecular dynamics, especially where the long-range interaction is crucial. Do the authors have observations and/or insights about potential results when applying HNO to such scenarios?

**Limitations:**

An limitation, while it is obvious, is its computational complexity. It is shown that the proposed model runs faster than other transformer-based prediction models, but its scalability, particularly to 3 dimensional scenarios, is questionable.

**Strengths And Weaknesses:**

**Strength:**
- Rationale behind the use of the hyperbolic structure is generally convincing (although a bit of ambiguity remains, which is detailed below).
- The architecture in neural operators that hyperbolic structure benefits are pinpointed based on the deep understanding of neural operators and hyperbolic geometry.
- Each theoretical claim is solid. Proposition 4.1 and Corollary E.4 support the authors' intuition described in Introduction.
- The performance gain in the experiments is significant.

**Weakness:**
- The motivation of using hyperbolic structure to alleviate the Euclidean crowding problem is not entirely convincing. While Lemma 3.2 introduces the difference in the growth rate of balls in the respective spaces, how (mathematically) this result alleviates the Euclidean crowding problem is not clear. I am afraid that Figure 1 (C) does not help that much in this regard. Besides, the reason why the authors expect the hierarchy-aware routing helps learn a better propagation operator is also unclear. This should be highlighted (and detailed, if possible) in the main text because this routing mechanism should be a direct corollary of the Poincare embedding, a key, unique property of the hyperbolic space.

- While the theoretical claims are solid, as also mentioned in Strength section, their logical relationships are not made tight enough. At first glance, they look like just a list of theoretical results, and their flow is very ragged. A remedy for this issue could be as follows; Lemma D.2 better has an introductory sentences that mentions, for example, the application of far-field decomposition to neural operators; The equation (25) can be asserted as a lemma for Theorem D.4, together with a proof essentially same as the block between the equations (21) and (24); After them (or at the beginning of this section), the authors can claim that neural operators also admit a hierarchical far-field decomposition structure, that supports the intuition mentioned around Lemma 3.1. The lack of this type of logical flows significantly impairs the readability. I also looked up Appendix D. However, all the theoretical statements therein except Proposition D.1 are independent from the hyperbolic structure, which leaves me a bit uncertain about the intention of having them.

- There exist a couple of stylistic issues:
  - Most of the theoretical results are deferred to Appendix section. Many statements in the paper, that are supported by the deferred theoretical results, lack pointers on the corresponding sections in Appendix. For example, it was really hard to notice Appendix F.1 was for Section 4.2 in the main text.
  - Known results and paper's results look like being mixed, which is very confusing and makes it hard to assess the novelty and contribution. For example, Proposition D.1 is a well-known result. Having this statement without a proper reference is misleading. On the other hand, I believe the far-field expansion detailed in Appendix D.3 should be mostly novel. However, for example, Lemma D.2 mentions a reference (Greengard & Rokhlin, 1987) while this reference does not contain the statement in the lemma. It is also odd that Assumption D.3 used in Lemma D.2 is stated after Lemma D.2.

**Minor:**
- I could not find a clarification about how the query and key maps are instantiated.

Overall, although there are a couple of issues that can diminish the readability, I believe the contribution of the paper is significant. Therefore, I recommend the acceptance of the paper.

---

> ### Author Rebuttal · Authors · 2026-03-31
>
> Dear Reviewer ixXT,
>
> Thank you for your careful reading and thoughtful feedback. We have considered your comments closely and address them below by category.
>
> ## `A. Method-related`
>
> - **A.1. Lemma 3.2, capacity law, and Euclidean crowding:**
>   Thank you for pointing this out. We state that hierarchical structures grow exponentially, while Euclidean volume grows only polynomially. What should be stated more explicitly is that Euclidean space cannot accommodate such rapidly expanding hierarchical structure in fixed dimension, which leads to crowding. We will make this point explicit in the revision.
>
> - **A.2. Why hierarchy-aware routing is effective:**
>   Thank you for raising this question. In our paper, hierarchy-aware interaction is interpreted as a learnable analogue of the efficient near--far organization behind FMM. Hierarchical tokens are naturally induced by hyperbolic attention. This follows the same intuition formalized in Lemma 3.1: far-field interactions admit accurate compressed approximation.
>
> ## `B. Presentation-related`
>
> - **B.1. Appendix D:**
>   Appendix D is intended to explain the motivation in Sec. 3 from the reader’s perspective: after Sec. 3, a natural question is how the near–far idea in FMM relates to hierarchical structure. This is why D.1 appears first, while D.2 and D.3 supplement the hyperbolic capacity law and far-field compressibility.
>
>   Two details may have been misleading. First, *(Greengard & Rokhlin, 1987)* is cited to indicate the origin of FMM, not that our appendix reproduces its exact statement. Second, Assumption D.3 appears after Lemma D.2 although it is used there. We will revise both points.
>
> - **B.2. Theoretical results and appendix correspondence:**
>   Thank you for pointing this out from the reader’s perspective. We had assumed that appendix titles such as “Proof of Proposition ...” made the correspondence clear enough. In the revision, we will add explicit cross-references from the main text.
>
> - **B.3. How the query and key maps are instantiated:**
>   In implementation, query and key maps are separate pointwise linear projections from the hidden states. In the latent-set pathway, cross-attention uses queries from latent tokens and keys from input-point features, while latent self-attention uses separate query/key projections on the latent states. We will make this explicit in the revision.
>
> ## `C. Experiments-related`
>
> - **C.1. FNO to point clouds:**
>   Vanilla FNO is defined only on regular grids. For a fair comparison, we interpolate the point cloud onto a Cartesian grid with fixed resolution and apply the FNO baseline on that grid.
>
> - **C.2. A molecular-dynamics-style pilot with an attention backbone:**
>   This is an interesting question! First, the original FMM paper was introduced for **Coulombic/gravitational particle interactions**. To probe this direction, we ran a small pilot on MD17 aspirin using Equiformer as the attention backbone, comparing the original model and our hyperbolic variant under the same setup. Our pilot was trained for 200 epochs, whereas the original Equiformer paper reports results with 2000 epochs:
>
>   |Variant|val-E MAE (meV)|val-F MAE (meV/Å)|test-E MAE (meV)|test-F MAE (meV/Å)|
>   |---|---:|---:|---:|---:|
>   |Original|5.5289|9.9846|6.1898|9.4885|
>   |Hyperbolic|6.8567|9.9920|6.4613|9.3263|
>
>   The result did not meet our expectation. Our current analysis suggests that the limited gain may be related both to the data characteristics and to the baseline design; in particular, the learned hierarchy in this molecular setting is much less pronounced than in the PDE setting. More importantly, realistic atomistic molecular dynamics requires additional ingredients, including *E(3)-equivariance, permutation symmetry, periodic boundary handling, and energy-based force prediction*. We believe this is a promising direction, but due to time constraints we did not pursue it further here.
>
> - **C.3. Scalability:**
>   On Darcy, higher efficiency mainly comes from achieving strong performance with fewer layers, consistent with the Euclidean crowding motivation. On large-scale datasets, we replace only the attention module of Transolver++ [1], keeping the rest unchanged. The added cost is controlled, consistent with our response to **Reviewer Q3pE, comment L3**.
>
> Thank you again for your constructive comments. We will incorporate these clarifications and additions into the revised version and hope they help address the main concerns.
>
> Best regards,
> Authors.
>
> [1] Transolver++: An Accurate Neural Solver for PDEs on Million-Scale Geometries. ICML 2025

---

> > ### Author Rebuttal · Reviewer_ixXT · 2026-04-02
> >
> > I thank the authors for the rebuttal. All my concerns were satisfactorily addressed. In particular, I found the additional experiment with MD17 dataset very interesting, while its result is disappointing at the first glance. I believe this poses a new problem and opens a new research direction. This experiment also serves as clarifying a limitation from another perspective, making the contribution of the paper clearer and sharper. I see through the rebuttal period, the indication and contribution of the paper became more valuable and significant beyond the original scope of the paper. Hence, I will support its acceptance.

---

> > > ### Author Response · Authors · 2026-04-04
> > >
> > > Dear Reviewer ixXT,
> > >
> > > Thank you sincerely for your thoughtful and thorough evaluation of our work. Your careful reading of both the main text and the appendix — from the theoretical statements to the experimental details — is deeply appreciated, and your constructive feedback has meaningfully improved our paper.
> > >
> > > We are glad that our rebuttal addressed your concerns satisfactorily.  While our pilot on MD17 aspirin did not yield the improvements we had hoped for, your observation that this "poses a new problem and opens a new research direction" is truly encouraging. We agree that this is a promising and underexplored area.
> > >
> > > On a related note, following up on our rebuttal, we have conducted additional experiments and found that HNO's performance on the molecular dynamics setting is moderately sensitive to the temperature parameter $\tau$. With a more careful hyperparameter search, we observed further improvements over the results reported in our rebuttal, which gives us more confidence that the gap can be narrowed with proper adaptation. We believe this direction is worth continued exploration.
> > >
> > > That said, our immediate next steps will likely focus on scaling HNO to industrial-scale PDE problems, where the efficiency and hierarchical routing properties of hyperbolic geometry may bring the most practical impact.
> > >
> > > We are grateful for your recognition of our contribution and for the encouragement your review has provided. Your rigorous yet supportive engagement with our work exemplifies the best of the peer review process, and it has been a genuine pleasure to interact with you! Thank you!
> > >
> > > Best regards,
> > >
> > > Authors

---

### Official Review · Reviewer_iyMb · 2026-03-12

**Soundness:** 3
**Presentation:** 3
**Significance:** 3
**Originality:** 3
**Overall Recommendation:** 5
**Confidence:** 4

**Summary:**

The paper introduces "Hyperbolic Neural Operator", a transformer neural operator which uses FMM style scale separation. The manuscript extends transformer neural operators by incorporating the inductive bais of hyperbolic geometry into the attention mechanism, by performing dot-product attention

**Compliance With Llm Reviewing Policy:**

Affirmed.

**Final Justification:**

All of my concerns have been addressed and the paper is a valuable addition to the field.

**Key Questions For Authors:**

- how does this scale to larger datasets. Models like transolver are sepcifically designed to scale to large numbers of input nodes.

**Limitations:**

yes

**Strengths And Weaknesses:**

### Strengths:
- elegantly incorporates hyperbolic geometry into the dot-product attention
- paper is well written and the method is explained clearly
- strong results showing improvement over common methods in this domain

### Weaknesses:
- it could be better motivated why we would want to have hyperbolic geometry in the embedding domain where we compute the attention on
-
- Table 2 dramatically differs visually from all of the regular Latex tables. I suggest changing it
- I would suggest having a much more expressive title. I was expecting a method for hyperbolic PDEs, not something that uses hyperbolic geometry in the attention mechanism
- while all of the math on separability makes sense (i am quite familiar with FMM methods), I feel this is mainly just added as a motivation and the connections are not clear beyond motivation. I would either add results that verify that e.g. Proposition 4.1 hold or I would move it to the appendix.

### Minor comments:
- (6) has a comma after the equation

---

> ### Author Rebuttal · Authors · 2026-03-31
>
> Dear Reviewer iyMb,
>
> Thank you for your careful reading and valuable feedback. We have considered your comments point by point, organized by category below.
>
> ## `A. Method-related`
>
> - **A.1. The connection between FMM and our method:**
>   We are sorry that our wording may have misled your review, and we are also glad to encounter a reviewer familiar with FMM theory. As you pointed out, FMM is indeed an important motivation for our method.
>
>   - **Hierarchy formation.** Hyperbolic projection naturally induces hierarchical tokens **(Sec. 4.3)**, and this hierarchical structure is more naturally accommodated in hyperbolic space **(Lemma 3.2)**.
>   - **Connection to FMM.** This connects to the same core intuition behind FMM: interactions across different regions should be organized efficiently across scales rather than treated uniformly.
>   - **Our role.** In this sense, HNO should not be read as implementing the FMM algorithm itself, but rather as a **learnable analogue consistent with the near--far organizational intuition of FMM**.
>
> ## `B. Presentation-related`
>
> - **B.1. Table 2 caption:**
>   Thank you for your suggestion. We will adjust the caption of Table 2. Our original consideration was to help readers understand the differences among the datasets more clearly.
>
> - **B.2. A more expressive title:**
>   This is an interesting question. One title we considered is *Hyperbolic Neural Operator: Hierarchical PDE Learning via Geometry-Induced Near--Far Routing*. We will consider this title revision within the ICML rules.
>
> - **B.3. Punctuation after Eq. (6):**
>   Thank you for your careful review. We will change the symbol after Eq. (6) to a period.
>
> ## `C. Experiments-related`
>
> - **C.1. Extension to larger datasets:**
>   Thank you for pointing this out.
>
>   - **Backbone.** Our scalability comes from using an already scalable backbone, namely Transolver++ [1].
>   - **What changes.** In the large-scale unstructured-mesh experiments, HNO is implemented by replacing the dot-product logits in Transolver++ [1] with stabilized hyperbolic-distance logits, while keeping the backbone width and depth unchanged.
>   - **What this tests.** This isolates whether replacing the routing kernel with a hyperbolic one can improve accuracy and near--far organization under the same scalable framework.
>
>   This implementation detail is already stated in the appendix (Supplementary, p. 33, “Large-Scale Benchmarks”): for the large-scale unstructured-mesh benchmarks, HNO is implemented by replacing the dot-product logits in Transolver++ with stabilized hyperbolic-distance logits, while keeping the same backbone width and depth. We will make this more explicit in the main text.
>
> - **C.2. Proposition 4.1:**
>   Thank you for this suggestion. To make Proposition 4.1 more concrete, we added an exact geometric check in the anonymous supplementary repository: https://anonymous.4open.science/r/for-re-1231/README.md.
>
>   Specifically, we visualize an exact symmetric slice of Proposition 4.1 with $r_q = r_k = r$. The plot shows that $\log \theta_{\max}(r,r;R)$ decreases approximately linearly in $r$, consistent with the predicted exponential angular contraction, and that the normalized ratio converges to $1$ under the exact asymptotic scale.
>
> Thank you again for your valuable feedback. We will incorporate these clarifications into the revised version.
>
> Best regards,
> Authors.
>
> [1] Transolver++: An Accurate Neural Solver for PDEs on Million-Scale Geometries. ICML 2025

---

> > ### Author Rebuttal · Reviewer_iyMb · 2026-04-01
> >
> > I thank the authors for their rebuttal. It addresses my main concerns. I do also prefer the alternative title that the authors suggest. The current title does not give any details with reagrds to what the algorithmic advantages of this method are.

---

> > > ### Author Response · Authors · 2026-04-02
> > >
> > > Dear Reviewer iyMb,
> > >
> > > We would like to express our sincere gratitude for your careful and expert evaluation. Your deep familiarity with FMM methods brought a uniquely valuable perspective to our work, and your constructive suggestions — from the title revision to the Proposition 4.1 verification — have been instrumental in sharpening both the presentation and the technical substance of our paper.
> > >
> > > We are honored that our rebuttal has **fully addressed your concerns**. During the rebuttal period, in response to another reviewer's request, we further conducted controlled comparisons against strong Euclidean gating baselines (norm-dependent temperature, RBF kernel with learnable bandwidth) under the same backbone. HNO consistently achieves the best accuracy, and mechanism analysis confirms that HNO produces a **significantly wider receptive-field hierarchy** than all Euclidean variants (RF spread 0.275 vs. 0.182–0.241), providing direct empirical evidence that the near–far stratification you highlighted goes beyond motivation. The full results are available in our reply to Reviewer Q3pE.
> > >
> > > Given that your concerns have been fully resolved and the paper has been further strengthened during the rebuttal period, if you feel these additional results enhance the overall contribution, we would be very grateful if you could consider reflecting this in your score.
> > >
> > > Thank you once again for your professional integrity and for helping us improve the quality of this paper. We look forward to incorporating your valuable suggestions into our final version.
> > >
> > > Best regards,
> > >
> > > Authors

---

### Decision · Program_Chairs · 2026-04-30

**Decision:**

Accept (regular)

**Comment:**

The paper originally received mixed reviews that lean toward the positive sides. The main concerns from the reviewers mostly regards the writing and exposition of the paper, while concerns directed toward experiments were mostly resolved during the rebuttal period.